# Uridine-derived ribose fuels glucose-restricted pancreatic cancer

Zeribe C. Nwosu[1,13], Matthew H. Ward[1,2,3,4,13], Peter Sajjakulnukit[1,13], Pawan Poudel[5,13], Chanthirika Ragulan[5], Steven Kasperek[1], Megan Radyk[1], Damien Sutton[1], Rosa E. Menjivar[6], Anthony Andren[1], Juan J.Apiz-Saab[7], Zachary Tolstyka[1], Kristee Brown[8], Ho-Joon Lee[1], Lindsey N. Dzierozynski[7], Xi He[8], Hari PS[5], Julia Ugras[1], Gift Nyamundanda[5], Li Zhang[1], Christopher J. Halbrook[1], Eileen S. Carpenter[9], Jiaqi Shi[10], Leah P. Shriver[2,3,4], Gary J. Patti[2,3,4], Alexander Muir[7], Marina Pasca di Magliano[8,11], Anguraj Sadanandam[5,12✉] & Costas A. Lyssiotis[1,9,11✉]

Pancreatic ductal adenocarcinoma (PDA) is a lethal disease notoriously resistant to therapy[1,2]. This is mediated in part by a complex tumour microenvironment[3], low vascularity[4], and metabolic aberrations[5,6]. Although altered metabolism drives tumour progression, the spectrum of metabolites used as nutrients by PDA remains largely unknown. Here we identified uridine as a fuel for PDA in glucose-deprived conditions by assessing how more than 175 metabolites impacted metabolic activity in 21 pancreatic cell lines under nutrient restriction. Uridine utilization strongly correlated with the expression of uridine phosphorylase 1 (UPP1), which we demonstrate liberates uridine-derived ribose to fuel central carbon metabolism and thereby support redox balance, survival and proliferation in glucose-restricted PDA cells. In PDA, *UPP1* is regulated by KRAS–MAPK signalling and is augmented by nutrient restriction. Consistently, tumours expressed high *UPP1* compared with non-tumoural tissues, and *UPP1* expression correlated with poor survival in cohorts of patients with PDA. Uridine is available in the tumour microenvironment, and we demonstrated that uridine-derived ribose is actively catabolized in tumours. Finally, UPP1 deletion restricted the ability of PDA cells to use uridine and blunted tumour growth in immunocompetent mouse models. Our data identify uridine utilization as an important compensatory metabolic process in nutrient-deprived PDA cells, suggesting a novel metabolic axis for PDA therapy.

PDA remains one of the deadliest cancers[1,2]. The PDA tumour microenvironment (TME) is a major contributor to this lethality, and is characterized by abundant immune cell infiltration, expansion of stromal fibroblasts and the associated deposition of extracellular matrix. This leads to an increase in interstitial fluid pressure and the collapse of arterioles and capillaries[3,4,7]. These phenomena collectively contribute to low oxygen saturation, therapeutic resistance, metabolic alterations, and heterogeneity within the tumour at the cellular level[5,8,9]. PDA cells surviving in such nutrient and oxygen deregulated TME exhibit metabolic adaptations that increase their scavenging and catabolic capabilities[10–13]. In addition, recent studies have defined tumour-extrinsic nutrient sources for PDA, including extracellular matrix, immune, and stromal-derived metabolites[14–16]. While these studies uncovered discrete nutrient inputs, comprehensive screens with the power to identify many such nutrient drivers and mechanisms have not been performed previously.

## Nutrient-deprived PDA consumes uridine

To screen for metabolites that fuel metabolism in nutrient-deprived PDA cells, we applied the Biolog phenotypic screening platform on 19 human PDA cell lines and 2 immortalized, non-malignant pancreas cell lines (human pancreatic stellate cells and human pancreatic nestin-expressing cells) (Fig. 1a). We used the screen to assess cellular ability to capture and metabolize more than 175 nutrients in a 96-well arrayed format under nutrient-limiting conditions (0 mM glucose, 0.3 mM glutamine and 5% dialysed fetal bovine serum (FBS)). The nutrient panel included carbon energy and nitrogen substrates (Supplementary Table 1). Metabolic activity was assessed by monitoring the reduction of a tetrazolium-based dye, a readout of cellular reducing potential, every 15 min for approximately 3 days (Fig. 1a and Extended Data Fig. 1a). Analyses of nutrient consumption profiles revealed

[1]Department of Molecular and Integrative Physiology, University of Michigan, Ann Arbor, MI, USA. [2]Department of Chemistry, Washington University in St Louis, St Louis, MO, USA. [3]Department of Medicine, Washington University in St Louis, St Louis, MO, USA. [4]Center for Metabolomics and Isotope Tracing, Washington University in St Louis, St Louis, MO, USA. [5]Division of Molecular Pathology, The Institute of Cancer Research, London, UK. [6]Cellular and Molecular Biology Program, University of Michigan, Ann Arbor, MI, USA. [7]Ben May Department for Cancer Research, University of Chicago, Chicago, IL, USA. [8]Department of Surgery, University of Michigan, Ann Arbor, MI, USA. [9]Department of Internal Medicine, Division of Gastroenterology, University of Michigan, Ann Arbor, MI, USA. [10]Department of Pathology and Clinical Labs, Rogel Cancer Center, University of Michigan, Ann Arbor, MI, USA. [11]Rogel Cancer Center, University of Michigan, Ann Arbor, MI, USA. [12]Centre for Global Oncology, Division of Molecular Pathology, The Institute of Cancer Research, London, UK. [13]These authors contributed equally: Zeribe C. Nwosu, Matthew H. Ward, Peter Sajjakulnukit, Pawan Poudel. ✉e-mail: anguraj.sadanandam@icr.ac.uk; clyssiot@med.umich.edu

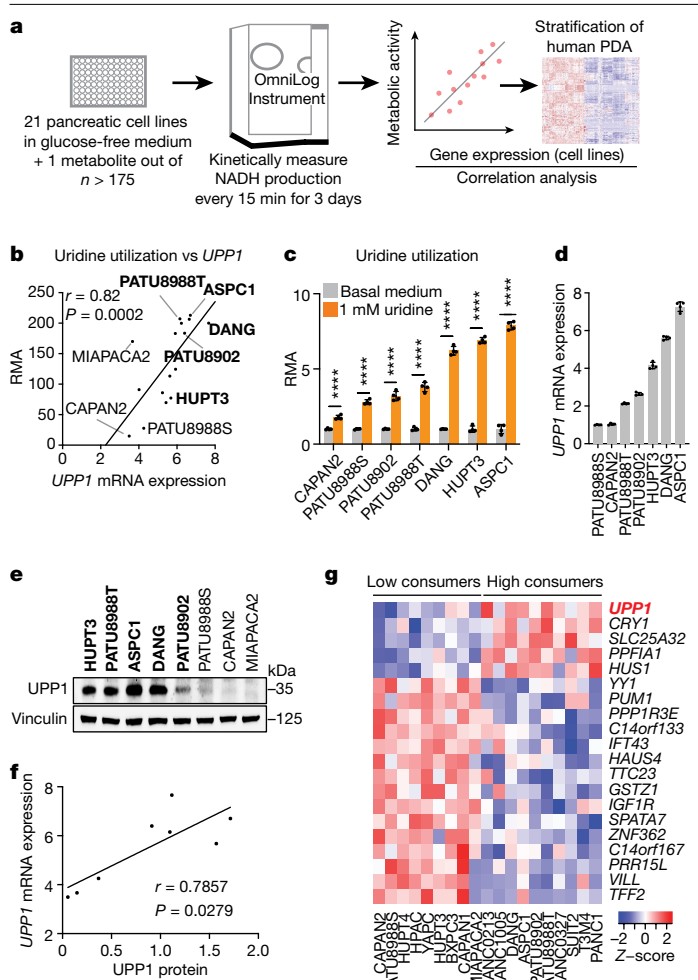

**Fig. 1 | Profiling of metabolite utilization in PDA cells identifies uridine.**
**a**, Scheme of the nutrient metabolism screening assay and the correlation with gene expression in PDA cell lines and tumours. **b**, Spearman correlation (*r*) between the normalized relative metabolic activity (RMA) for uridine catabolism in the screening data and *UPP1* mRNA expression data from an independent dataset[17] (16 PDA cell lines). *UPP1*-high cell lines are shown in bold. **c**, The RMA in a subset of PDA cell lines following supplementation with 1 mM uridine for 3 days in glucose-free condition. **d**, Quantitative PCR (qPCR) validation of *UPP1* mRNA expression in a subset of PDA cell lines. **e**, Immunoblot showing basal UPP1 expression in PDA cell lines. Blots are representative of three technical replicates with similar results. **f**, Spearman correlation (*r*) between protein densitometry analysis of the blot in **e** and *UPP1* mRNA expression in the eight PDA cell lines highlighted in **e**. **g**, Top 20 genes differentially expressed by the PDA cell lines that were identified as uridine-high consumers/metabolizers compared with uridine-low consumers from the nutrient metabolism screen. Data source: Cancer Cell Line Encyclopedia (CCLE). Data in **c**,**d** are mean ± s.d. See Methods 'Statistics and reproducibility' section for additional information.

several metabolites that, in the absence of glucose, were utilized at similar levels as the glucose positive control (Extended Data Fig. 1b). For example, adenosine, uridine and several sugars were utilized by most of the cell lines.

## Uridine consumption correlates with UPP1

To select lead metabolites for investigation, we correlated metabolite utilization patterns to the expression of metabolism-associated genes using a public dataset[17,18]. From the top metabolite and gene correlation pairs (Extended Data Fig. 1c), we pursued uridine and uridine phosphorylase 1 (UPP1) (*r* = 0.82, *P* = 0.0002; Fig. 1b) for the following

reasons. First, *UPP1* expression correlated positively with the metabolic activity from its known substrate, uridine. Second, our in vitro validation showed that all tested PDA cells utilized uridine, although to varying degrees (Fig. 1c and Extended Data Fig. 1b), suggesting that it is a broadly used metabolic fuel. Third, contrary to our expectation that nutrients used in the absence of glucose would be carbohydrates, uridine was unusual in that it is a nucleoside. Finally, to our knowledge, UPP1-mediated uridine metabolism is unexplored in the context of PDA.

We further confirmed the correlation of uridine catabolism and UPP1 expression by mRNA and protein analyses (Fig. 1d–f). To determine the specificity of this association, we assessed the correlation of *UPP1* expression to other nucleosides in the Biolog screen. Although both inosine and adenosine were readily catabolized, their utilization was not correlated with *UPP1* expression. Thymidine was neither actively metabolized nor correlated, when compared with negative controls (Extended Data Fig. 1d). These results indicate that the association between *UPP1* expression and uridine catabolism is robust and specific.

Next, we rank-sorted the PDA cell lines into high and low uridine metabolizers on the basis of the Biolog data, and identified more than 700 differentially expressed genes (*P* < 0.05) between these groups using the Cancer Cell Line Encyclopedia (CCLE) data[19] (Supplementary Table 2). Consistent with our previous correlation analysis in a different dataset (Extended Data Fig. 1c), UPP1 was the top gene in uridine-high consumers/metabolizers in the CCLE dataset (Fig. 1g). Pathway analysis of the upregulated genes in the uridine-high metabolizers (that is, *UPP1*-high cell lines) showed upregulation of endocytosis and several inflammation and/or immune-related pathways, notably NFκB signalling (Supplementary Table 3). We also observed that *UPP1*-high tumours exhibit higher expression of glycolysis genes (Extended Data Fig. 1e), indicating a potential link between the UPP1–uridine axis and energy metabolism. By contrast, *UPP1*-high cell lines and *UPP1*-high PDA tumours from patients displayed a profound downregulation of other metabolic pathways (Extended Data Fig. 1f,g), notably amino acid, fatty acid and glutathione metabolism.

## Uridine-derived ribose fuels metabolism

Our screen (Fig. 1a) was performed in glucose-free medium to reveal both metabolites whose use would otherwise be overshadowed by glucose, and the carbon sources that could act in place of glucose. Thus, we next directly assessed the metabolic activity of equimolar glucose and uridine. Across four PDA cell lines, uridine and glucose fuelled metabolism to a similar degree (Fig. 2a). Previous reports have also documented that uridine can substitute for glucose by supporting nucleotide metabolism[20–26]. However, our screening data illustrate that uridine supplementation increases the cellular reducing potential. Together with the observed connection with UPP1, which catalyses the cleavage of uridine to ribose-1-phosphate and uracil, we hypothesized that the UPP1-liberated ribose is recycled into central carbon metabolism to support cellular reducing potential. To test this hypothesis, we supplemented glucose-deprived cells with ribose, a cell-permeable substitute for ribose 1-phosphate. Indeed, similar to exogenous uridine, ribose supplementation fuelled the reducing potential (Fig. 2a).

In our initial screen, glutamine concentration was also intentionally low (0.3 mM), as it is an important anaplerotic substrate that fuels tricarboxylic acid (TCA) cycle and proliferation in PDA[27]. Uridine potentiated reducing potential with or without glutamine and had a greater effect when glutamine was present (Extended Data Fig. 2a). Together, these data suggest that uridine and glucose similarly fuel central carbon metabolism distinctly from glutamine.

Next, we provided uridine to the UPP1-low PATU8988S and the UPP1-high DANG cell lines under glucose deprivation and applied liquid chromatography–mass spectrometry (LC–MS)-based metabolomics[28]. In both cell lines, uridine supplementation led to increased levels of glycolytic intermediates and lactate secretion (suggesting glycolytic

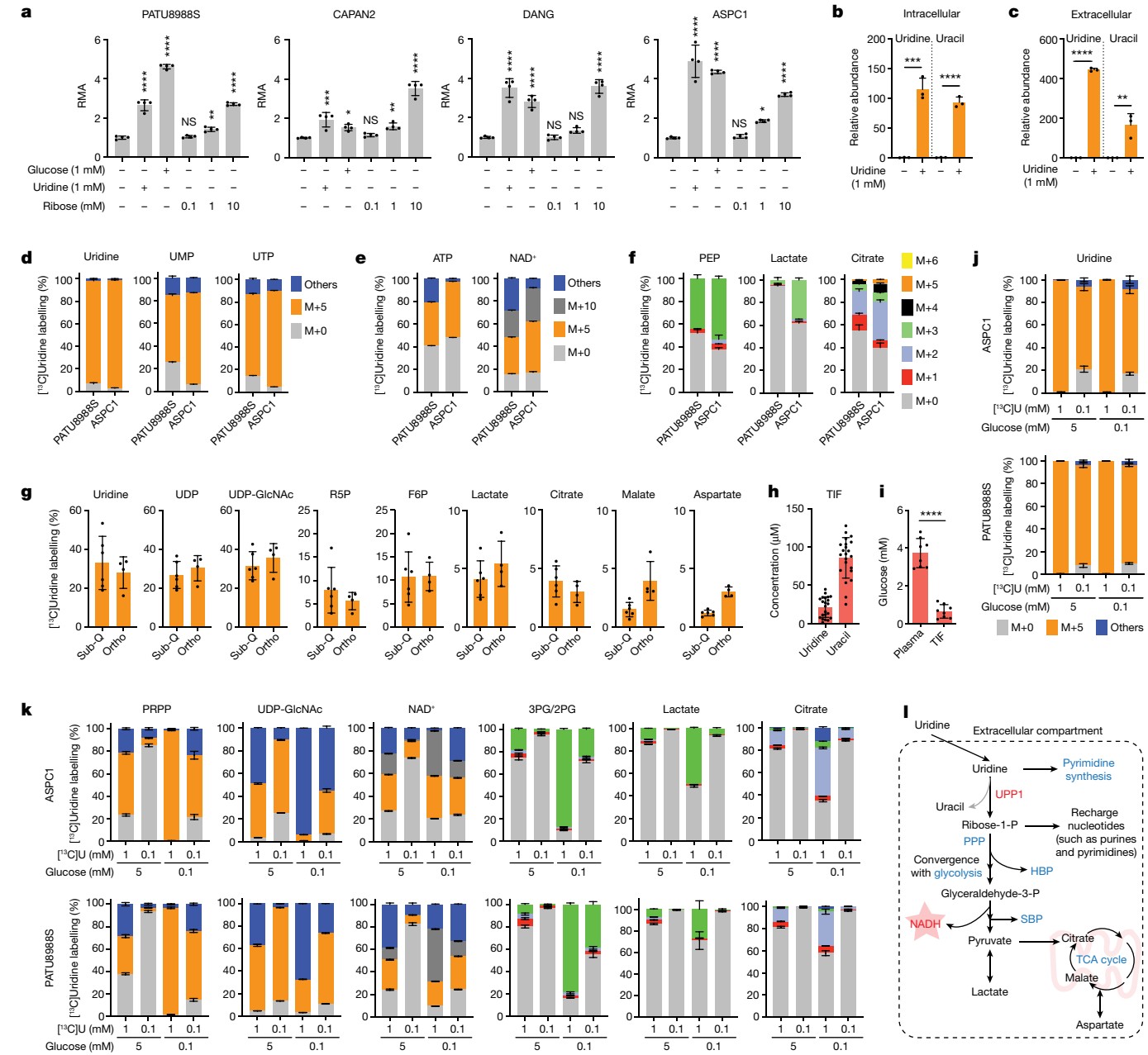

**Fig. 2 | Uridine-derived ribose supports nutrient-restricted PDA. a**, RMA of four PDA cell lines supplemented with glucose, uridine or ribose under nutrient-limiting culture conditions (0 mM glucose, 0.3 mM glutamine and 5% dialysed FBS). **b,c**, Intracellular and extracellular uridine and uracil after 24 h culture of PATU8988S cells in medium with no glucose and 10% dialysed FBS with or without 1 mM uridine as measured by LC–MS. **d–f**, Mass isotopologue distribution of carbon derived from $[^{13}C_5]$uridine in uridine, UMP and UTP (**d**), ATP and NAD+ (**e**) and phosphoenolpyruvate (PEP), lactate and citrate (**f**) in the indicated cell lines after 24 h culture with 1 mM uridine. **g**, Isotope tracing showing $[^{13}C_5]$uridine-derived carbon labelling in subcutaneous (sub-Q) or orthotopically (ortho) implanted KPC 7940b tumours collected 1 h after injecting the mice with 0.2 M $[^{13}C_5]$uridine. F6P, fructose-6-phosphate; R5P, ribose-5-phosphate; UDP-GlcNAc, uridine diphosphate *N*-acetylglucosamine; **h**, Absolute quantification via metabolomics of uridine and uracil concentration in the TIF of orthotopic PDA tumours from syngeneic mouse KPC cells. **i**, Absolute quantification via metabolomics of glucose concentration in the pancreatic TIF and plasma of mice orthotopically implanted with KPC 7490b syngeneic tumours. **j,k**, Mass isotopologue distribution of $[^{13}C_5]$uridine ribose-derived carbon after 24 h culture of ASPC1 and PATU8988S cells in medium supplemented with 1 mM or 0.1 mM uridine and each with 5 mM or 0.1 mM glucose. **j**, Uridine. **k**, PRPP, UDP-GlcNAc, NAD+, 3-phosphoglycerate/ 2-phosphoglycerate (3PG/2PG), lactate and citrate. PRPP, phosphoribosyl pyrophosphate. **l**, Schematic depicting the fate of uridine-derived ribose carbon in PDA cells actively catabolizing uridine. Glyceraldehyde-3-P, glyceraldehyde-3-phosphate; HBP, hexosamine biosynthetic pathway; PPP, pentose phosphate pathway; ribose-1-P, ribose-1-phosphate; SBP, serine biosynthesis pathway. See Methods 'Statistics and reproducibility' section for additional information.

flux), uridine derivatives (suggesting overflow metabolism), amino acids (indicative of increased anabolism) and TCA cycle intermediates (suggesting more mitochondrial activity) (Extended Data Fig. 2b–e). Moreover, supplementation with uridine led to a marked accumulation

of intracellular uridine and over 100-fold increase in uracil content in the medium (Fig. 2b,c and Extended Data Fig. 2f,g), consistent with uridine capture, ribose extraction and uracil release. Notably, the intracellular uracil concentration increased by a similar amount of

uridine, reflective of direct conversion of substrate to product (Fig. 2b and Extended Data Fig. 2f). Collectively, these profiling efforts support a model in which uridine is catabolized to broadly fuel PDA cell metabolism.

To precisely delineate how uridine is metabolized, we used LC–MS to trace the metabolic fate of isotopically labelled uridine ([$^{13}C_5$]uridine with uniformly labelled ribose carbon)[29] in PATU8988S (UPP1-low) and ASPC1 (UPP1-high) cell lines. Both cell lines demonstrated high uridine, UMP and UTP labelling (over 90%), as indicated by M+5 from ribose (Fig. 2d). Other nucleotides, such as ATP, AMP and ADP (all M+5), as well as NAD$^+$ (M+5, M+10) were also labelled (Fig. 2e and Extended Data Fig. 2h), demonstrating the use of uridine-derived ribose for ribosylation of the adenine base. Also labelled were glycolytic (PEP, pyruvate and lactate), PPP (X5P and ribose-5-phosphate), hexosamine biosynthetic pathway (UDP-GlcNAc) and TCA cycle intermediates (malate and citrate), as well as non-essential amino acids (aspartate, glutamate and serine) and oxidized glutathione (Fig. 2f and Extended Data Fig. 2h).

To determine the relevance of uridine metabolism for pancreatic tumours in vivo, we implanted mouse syngeneic pancreatic cancer cells into the pancreas of immunocompetent hosts to establish tumours. Then, we injected these mice with [$^{13}C_5$]uridine, and collected tumour tissues after 1 h for LC–MS analysis. Indeed, there was a robust uptake of uridine by the tumours, with almost 30% of the uridine pool being labelled. We observed the M+5 label in pyrimidine and purine species (that is, ribose salvage) as well as in glycolytic and TCA cycle intermediates (Fig. 2g and Extended Data Fig. 3a). $^{13}C$ incorporation into the TCA cycle was low in vivo, presumably owing to the short duration of labelling. Nearly identical results to those from orthotopic studies were observed in subcutaneous tumours from immune-competent mice. These results confirm that PDA catabolizes uridine in vivo.

In parallel, we collected tumour interstitial fluid (TIF) from independent orthotopic allograft tumours and quantified bulk uridine and glucose concentrations by LC–MS. Uridine and glucose were present in the low and high micromolar concentration range, respectively (Fig. 2h,i), similar to previous findings[30]. To determine how physiological concentrations of glucose and uridine affected uridine metabolism, we grew two PDA lines in 5 mM or 0.1 mM glucose and 0.1 mM or 1 mM [$^{13}C_5$] uridine and analysed labelling patterns. First, the labelling of uridine was nearly 100% (Fig. 2j). At the low equimolar concentration (0.1 mM uridine and glucose), uridine carbon contributed to several metabolites in the PPP, PRPP (involved in nucleotide biosynthesis), NAD$^+$, glycolysis, and TCA cycle, exceeding 50% enrichment in some cases (Fig. 2k and Extended Data Fig. 3b). When glucose was 50-fold higher (5 mM), uridine carbon contributed to a much lower level to metabolite labelling, whereas at lower glucose concentrations, uridine carbon dominated, consistent with competition for these two carbon sources into the same pathways. Further isotope tracing using the exact TIF concentrations of uridine and glucose showed that both human (ASPC1) and mouse (MT3-2D) cells incorporate uridine into central carbon metabolism (Extended Data Fig. 3c,d). We confirmed these results in four human PDA lines with the tetrazolium assay: uridine supported bioenergetics at physiological (Extended Data Fig. 4a,b) but not at elevated glucose levels (Extended Data Fig. 4c).

Our data suggest that uridine yields ribose-1-phosphate via UPP1 to fuel both catabolic and biosynthetic metabolism. Ribose-1-phosphate can be converted to the PPP product ribose-5-phosphate by phosphoglucomutase 2 (PGM2) to enter nucleotide biosynthesis. Alternately, cells can convert uridine to UMP via uridine-cytidine kinase (UCK1/2) in the pyrimidine salvage pathway (Extended Data Fig. 4d). We found that *PGM2* and *UCK2* are high in PDA and *UCK1* is low, but these genes were largely uncorrelated with *UPP1* (Extended Data Fig. 4e–h). Being the most upregulated, we tested PGM2 by western blot and found it to be expressed in most PDA cells but uncorrelated with UPP1 (Extended Data Fig. 4i). Inhibition of the three genes using short interfering RNA (siRNA) showed that only *PGM2* knockdown suppressed the uridine-mediated

rescue of metabolic activity following glucose deprivation (Extended Data Fig. 4j–l). Together, these data support our model in which uridine catabolism converges with central carbon metabolism, and they also reveal that exogenous uridine fuels PDA metabolism in a similar way to glucose, supplying carbon for redox, nucleotide, amino acid and glycosylation metabolite biosynthesis (Fig. 2l).

## UPP1 provides uridine-derived ribose

To confirm the role of UPP1 in uridine catabolism, we knocked out UPP1 (UPP1-KO) using CRISPR–Cas9 in the PATU8988S (UPP1-low) and ASPC1 (UPP1-high) human PDA cell lines and validated two independent clones per cell line (Fig. 3a). In these knockout lines, the ability of uridine to rescue NADH production in the absence of glucose (Fig. 3b) or cellular bioenergetics, read out by ATP-based viability (Fig. 3c), was abolished. Consistent with the blockade of uridine catabolism, metabolomics showed that UPP1-KO cell lines displayed an increase in intracellular and extracellular uridine (Extended Data Fig. 5a,b) accompanied by a marked drop in intracellular and extracellular uracil (Fig. 3d and Extended Data Fig. 5b). Furthermore, UPP1-KO broadly altered the intracellular metabolome of both cell lines (Extended Data Fig. 5c). Notably, ASPC1 cells are highly sensitive to glucose deprivation in combination with UPP1-KO, thus metabolomics in this cell was performed after 6 h when assessing the knockout effect.

We next used our isotope tracing metabolomics platform to determine the effect of UPP1-KO on uridine catabolism in PATU8988S and ASPC1 cells (Fig. 3e and Extended Data Fig. 5d–k). The ribose-labelled [$^{13}C_5$]uridine tracing showed similar fractional enrichment of the intracellular uridine pool in the UPP1-KO and control cells (Extended Data Fig. 5d,e), indicative of unchanged, steady-state uridine uptake. By contrast, and consistent with our model, flux of uridine ribose-derived carbon into glycolysis (Fig. 3e and Extended Data Fig. 5f), TCA cycle-associated metabolites (Fig. 3e and Extended Data Fig. 5g), non-essential amino acids, oxidized glutathione and UDP-GlcNAc (in glycosylation), was holistically blocked or suppressed (Extended Data Fig. 5h,i). Recycling of uridine-derived ribose was also completely blocked in the UPP1-KO cells, as evidenced by the absence of carbon labelling in NAD$^+$ and the bioenergetic metabolites AMP, ADP and ATP (Extended Data Fig. 5j,k). Together, these results reveal that the UPP1-mediated catabolism of uridine is indispensable for the utilization of uridine to support reducing potential, bioenergetics and cell proliferation, and provide a detailed molecular confirmation that UPP1 directly controls the utilization of uridine-derived ribose in PDA cells.

## High UPP1 in PDA predicts poor survival

To further assess the relevance of UPP1 in PDA tumours, we next analysed its expression in publicly available human PDA datasets. We found that *UPP1* is highly expressed in PDA tumours compared with non-tumoural samples, and also in liver metastasis compared with primary tumours (Fig. 3f). Its paralogue *UPP2* is not expressed in this tumour type (Extended Data Fig. 6a), as observed in The Cancer Genome Atlas (TCGA) data. We also observed from the Human Protein Atlas public database that *UPP1* gene expression is extremely low in normal pancreas and UPP1 protein expression is high in PDA (Extended Data Fig. 6b,c). Consistently, *UPP1* expression is also high in several other non-PDA cancers from TCGA, with colon and prostate cancers being the notable exceptions (Extended Data Fig. 6d). High *UPP1* also predicted poor survival outcome in lung, stomach, liver and renal cancers (Extended Data Fig. 6e). Indeed, we tested uridine utilization in several non-PDA cancer cell lines and observed a modest uridine-derived increase in metabolic activity (Extended Data Fig. 6f), supporting the potential relevance of this metabolite in other cancers.

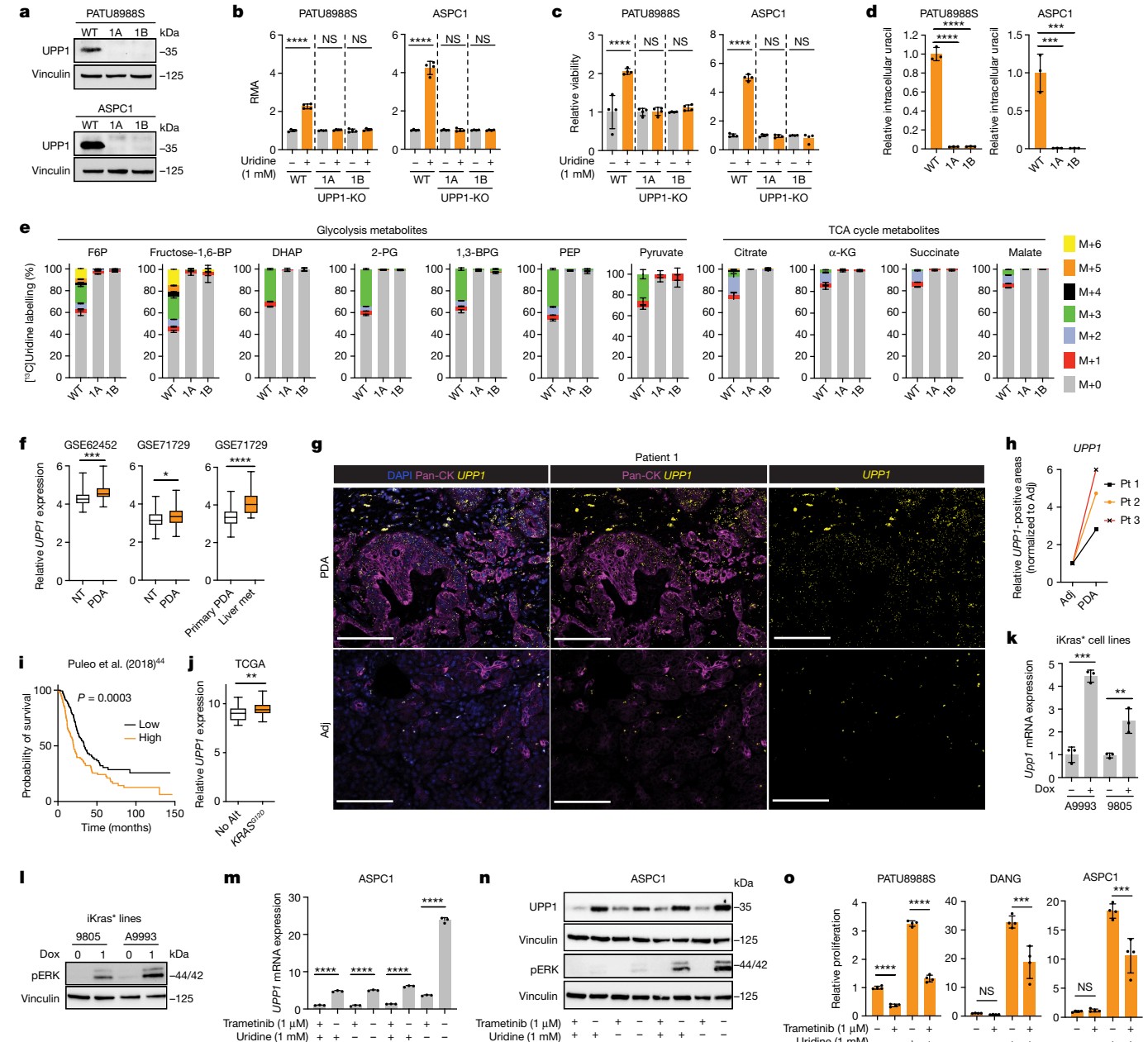

**Fig. 3 | KRAS-driven UPP1 liberates ribose and is increased in PDA.**
**a**, Western blot validation of UPP1-KO in human PDA cell lines. 1A and 1B denote UPP1-KO cells. WT, wild type. **b,c**, RMA from tetrazolium assay showing uridine-derived reducing potential (**b**) and CellTiter Glo showing ATP production (**c**) in UPP1-KO and wild-type PATU8988S and ASPC1 cells cultured with or without 1 mM uridine for 48 h. **d**, Relative intracellular uracil as determined by LC–MS in wild-type and UPP1-KO human PDA clones cultured with 1 mM uridine for 24 h (PATU8988S) or 6 h (ASPC1). **e**, Mass isotopologue distribution of 1 mM [$^{13}C_5$]uridine-derived carbon in glycolysis and TCA cycle metabolites in wild-type or UPP1-KO ASPC1 cells after 6 h. α-KG, α-ketoglutarate; 1,3-BPG, 1,3-bisphosphoglycerate; DHAP, dihydroxyacetone phosphate; fructose-1,6-BP, fructose-1,6-bisphosphate. **f**, *UPP1* mRNA expression in PDA tumours and non-tumoural pancreas tissues in microarray datasets. Liver met, liver metastasis; NT, non-tumour tissue. **g,h**, RNAscope showing representative *UPP1* mRNA expression in tumour and adjacent normal tissue (adj) sections (**g**) and quantification from three patients (Pt 1–3) with PDA (**h**). Scale bars, 100 μm. **i**, Kaplan–Meier overall survival analysis (log-rank test) based on ranked *UPP1* expression in the PDA dataset published previously[44]. **j**, Comparison of *UPP1* mRNA expression in human PDA tumours annotated as *KRAS^G12D* or with no alteration (No Alt) in *KRAS* from the TCGA dataset. **k**, qPCR data showing *Upp1* expression in mouse cell lines (A9993 and 9805) with doxycycline-inducible oncogenic *Kras* (iKras*). **l**, Western blot validation of MAPK pathway induction as indicated by phosphorylated ERK (pERK) in the iKras* cell lines. **m,n**, qPCR for *UPP1* mRNA (**m**) and western blot for pERK and UPP1 (**n**) in ASPC1 cells cultured with or without 5 mM glucose, 1 mM uridine and 1 μM trametinib for 48 h. **o**, MTT assay showing relative proliferation of PDA cell lines with 1.25 μM trametinib and 1 mM uridine in the absence of glucose. See Methods 'Statistics and reproducibility' section for additional information.

To independently validate UPP1 expression in human PDA, we used patient samples to assay tumoural *UPP1* expression by RNAscope (Fig. 3g,h and Extended Data Fig. 7a), the cellular distribution of *UPP1* expression by single cell RNA sequencing (Extended Data Fig. 7b,c), and UPP1 expression by immunohistochemistry (Extended Data Fig. 7d). Collectively, these data were consistent with public databases, illustrating UPP1 upregulation in human PDA relative to normal pancreas tissue. In addition, our histological assessment of

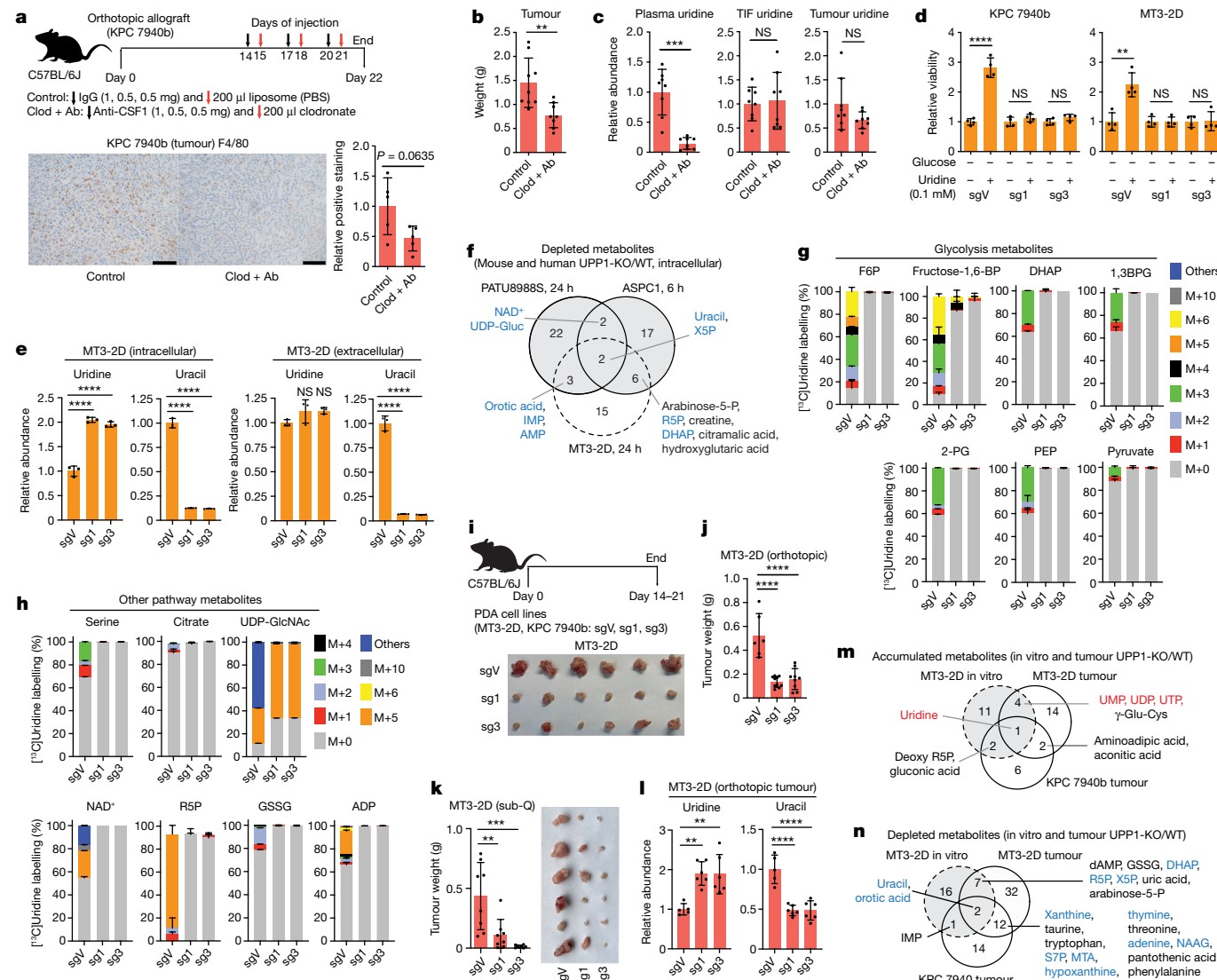

**Fig. 4 | UPP1-KO impairs the growth of orthotopic pancreatic tumour allografts. a**, Schematic of macrophage depletion (top) and validation by immunohistochemistry with F4/80 monoclonal antibody (bottom). Scale bars, 100 μm. Clod + Ab denotes anti-mouse CSF1 antibody and clodronate liposome. **b**, Tumour weight from control and macrophage-depleted mice at end point. **c**, Relative plasma, TIF and whole-tumour uridine levels in samples from the control and macrophage-depleted groups, measured by LC–MS. **d**, CellTiter Glo assay indicating relative viability (via ATP) of UPP1-KO (sg1 and sg3) and non-targeting control vector (sgV) mouse PDA cells. **e**, Relative intracellular and extracellular uridine and uracil levels in the control and UPP1-KO MT3-2D mouse PDA cell lines after 1 mM uridine supplementation, as determined by LC–MS. **f**, Venn diagram showing metabolites depleted in vitro upon UPP1-KO in the human (PATU8988S and ASPC1) and mouse (MT3-2D) PDA cell lines, determined by LC–MS. Metabolites in blue are those from the glycolysis, nucleotide biosynthesis or pentose phosphate pathways. Arabinose-5-P, arabinose-5-phosphate; IMP, inosine monophosphate; UDP-Gluc, uridine diphosphate-glucose; X5P, xylulose 5-phosphate. **g**,**h**, Stable isotope tracing showing mass isotopologue distribution of 1 mM [$^{13}C_5$]uridine-derived carbon in metabolites

from glycolysis (**g**) and other pathways (**h**) in the MT3-2D cells after 6 h culture. 1,3BPG, 1,3-bisphosphoglycerate; GSSG, oxidized glutathione. **i**, Top, schematic of tumour studies. Bottom, representative photograph of tumours collected from mice orthotopically implanted with the control vector (sgV) or UPP1-KO (sg1, sg3) MT3-2D cells. **j**, Tumour weight at end point of C57BL/6J mice orthotopically implanted with the MT3-2D cells. **k**, Weight and photograph of tumours collected at the end point after subcutaneous implantation of MT3-2D cells into the left and right flanks of C57BL/6J mice. **l**, Relative tumour uridine and uracil abundance in the orthotopic tumour samples described in **i**,**j**, as determined by LC–MS. **m**,**n**, Venn diagram showing metabolites commonly accumulated (**m**) and depleted (**n**) in UPP1-KO tumours implanted in mice and UPP1-KO cultured cells versus the wild type, as determined by LC–MS. **m**, Metabolites in red are associated with nucleotide biosynthesis. **n**, Metabolites in blue are associated with glycolysis, nucleotide biosynthesis or PPP. MTA, 5′-methylthioadenosine; NAAG, N-acetylaspartylglutamic acid; S7P, sedoheptulose 7-phosphate. See Methods 'Statistics and reproducibility' section for additional information.

injured acinar cells indicated that UPP1 is upregulated upon pancreatic injury (Extended Data Fig. 7d), suggesting a potential role in PDA formation. Finally, stratification of tumour datasets also showed that the high expression of *UPP1* predicted poor overall survival outcome in three out of the four PDA patient cohorts we analysed (Fig. 3i and Extended Data Fig. 7e). Together, these data support a key role for UPP1 in PDA.

## KRAS–MAPK pathway regulates UPP1

*KRAS* mutations are the signature transforming event observed in the majority of PDAs[31]. Using data from TCGA, we determined that PDA with the *KRAS*[G12D] mutation express higher levels of *UPP1* than those with no *KRAS* alteration (Fig. 3j). We also analysed the CCLE protein expression data of *KRAS* wild-type versus mutant cell lines across cancers

and from lung or colorectal cancer. A targeted analysis of PDA lines was not performed because *KRAS* mutations are observed in all but one line in the CCLE. From the pan-analysis (*n* = 374), we observed a borderline, non-significant, association between *KRAS* status and UPP1 expression (*P* = 0.09) (Extended Data Fig. 8a), but lung cancer cell lines (*n* = 79) showed a significant association (*P* = 0.003). Colorectal cancer lines (*n* = 30) showed no difference or slightly reduced UPP1 in mutant *KRAS* lines (Extended Data Fig. 8a), consistent with the data in Extended Data Fig. 6d. The colon cancer results may be owing to the timing of *KRAS* mutation in tumour evolution and differences in its tissue-specific function relative to that of lung and PDA[32].

To experimentally test the role of mutant KRAS on UPP1 expression, we first queried published microarray data from our doxycycline-inducible KRAS (iKras) mouse model of PDA[33,34]. Mutant KRAS promoted *Upp1* expression in a subcutaneous xenograft model in vivo and iKras PDA cell lines in vitro (Extended Data Fig. 8b). Consistently, in vitro validation experiments in two additional, independent iKras cell lines confirmed doxycycline (via KRAS) induction of *Upp1* (Fig. 3k,l).

We previously showed that KRAS-mediated regulation of anabolic glucose metabolism in PDA occurs via mitogen-activated protein kinase (MAPK) signalling and MYC-dependent transcription[34]. In human and mouse cell lines, the pharmacological inhibition of MAPK reduced UPP1 transcript and protein, concurrent with the suppression of pERK (Fig. 3m,n and Extended Data Fig. 8c–j). MAPK inhibition also blocked the catabolism of uridine, as reflected by intracellular uridine accumulation, changes in a spectrum of other metabolites (Extended Data Fig. 8k), and a suppressed uridine-fuelled proliferation (Fig. 3o and Extended Data Fig. 8l). In contrast to the previous KRAS mechanism[34], MYC inhibition did not alter *UPP1* expression or appear among the transcription factors binding to the *UPP1* promoter (Extended Data Fig. 9a–c), suggesting that MYC does not mediate KRAS regulation of UPP1 in PDA.

## Nutrient availability modulates UPP1

Given that glucose availability influences the use of uridine-derived ribose, we hypothesized that a glucose-depleted microenvironment triggers PDA to upregulate UPP1 as a compensatory response. Indeed, the removal or reduction of glucose in the medium induced a strong increase in UPP1 expression (Fig. 3m–n and Extended Data Figs. 8c–e,h–j and 9d), which was attenuated in uridine-supplemented medium. Consistently, we found that *Upp1* was strongly induced in a mouse tumour-derived cell line from the KPC model (*p48-cre;LSL-Kras^G12D^;LSL-Trp53^R172H^*) when cultured with TIF medium[30] or implanted as orthotopic allografts, relative to cells cultured in routine medium with high glucose (Extended Data Fig. 9e). Thus, KRAS–MAPK signalling and a nutrient-deprived TME may both be responsible for high UPP1 expression.

## UPP1-KO blunts PDA tumour growth

Uridine concentration is reported to be around twofold higher in TIF than in plasma[30]. In looking for a cellular source supplying uridine to tumours, we observed from our previously generated dataset[14] that macrophages release uridine and uracil in vitro when differentiated and polarized to a tumour-educated fate with PDA-conditioned medium (Extended Data Fig. 10a). Thus, we tested the role of tumour-associated macrophages (TAMs) in supplying intratumoural uridine by depleting macrophages from mouse allograft tumours. To this end, we treated mice with a CSF1 antibody and clodronate liposome combination (Fig. 4a), which depletes TAMs and suppressed orthotopic tumour growth[35,36] (Fig. 4a,b). We observed a reduction in the plasma uridine level by around eightfold upon macrophage depletion, concomitant with an increased plasma uracil level. However, uridine and uracil levels in the TIF and tumour were not altered (Fig. 4c and Extended Data Fig. 10b). This marked effect on plasma uridine levels following

macrophage depletion indicates that macrophages may be important mediators of uridine production and/or release.

To determine whether UPP1 supports tumour growth in vivo, we generated two independent models of UPP1-KO in the syngeneic mouse pancreatic cancer lines MT3-2D and KPC 7940b. Two single guide RNA (sgRNA) constructs targeting *Upp1* (sg1 and sg3) were compared to a vector control (sgV). We confirmed that glucose-deprived mouse UPP1-KO cells were not rescued by uridine in vitro (Fig. 4d). Furthermore, metabolomics profiling confirmed intracellular uridine accumulation and a reduced intracellular and extracellular uracil (Fig. 4e)−consistent with a blocked uridine catabolism. These were accompanied by a change in a spectrum of other metabolites that were also altered in the human UPP1-KO cell lines (Fig. 4f and Extended Data Fig. 10c). Finally, isotope tracing of uridine ribose-derived carbon illustrated conclusively that UPP1-KO in the mouse lines restricted the use of uridine to fuel central carbon metabolism (Fig. 4g,h). This battery of metabolic assays confirmed a successful UPP1-KO in lieu of a mouse UPP1 antibody. Notably, the UPP1-KO cells did not differ from controls in terms of proliferation under optimal culture where glucose is present (Extended Data Fig. 10d).

We orthotopically implanted these two mouse lines and the corresponding UPP1-KO lines into the pancreas of syngeneic hosts and assessed tumour weight at end point (Fig. 4i). In both lines, contrary to the lack of a proliferative defect in vitro, we observed a markedly reduced tumour growth following UPP1-KO (Fig. 4j and Extended Data Fig. 10e). Similar results were also reproduced in an immunocompetent subcutaneous model using the MT3-2D cell lines (Fig. 4k). Metabolomic profiling of the orthotopic tumours revealed an increase in tumoural uridine and a drop in uracil in the UPP1-KO tumours as well as a profound change in the metabolome relative to vector controls (Fig. 4l–n and Extended Data Fig. 10f,g). In addition, compared with in vitro data, we observed the accumulation of a similar collection of metabolites (Fig. 4m) and a strong depletion of uracil, components of PPP, glycolysis and nucleotide metabolism (Fig. 4n).

The marked anti-tumour effect of UPP1-KO prompted us to look for changes in the in vivo microenvironment. Histological analysis of F4/80 staining revealed no differences in macrophage content between UPP1-KO and vector control tumours. However, the UPP1-KO tumours had lower vessel density (CD31) and more anti-tumour T cell infiltration (CD8 T cells) (Extended Data Fig. 10h). Together, our data indicate that UPP1 and uridine are important in PDA growth. The contrast between the in vitro and in vivo growth phenotype further highlights the role of nutrient availability, as well as the potential involvement of the TME and immune cell subsets in influencing the necessity of UPP1 in vivo.

## Discussion

The metabolic features of PDA drive disease aggression and therapeutic resistance and present new opportunities for therapy[2,6]. Despite this, the range of nutrients used by PDA cells is poorly understood. We addressed this problem by applying high-throughput in vitro nutrient screening and found that under glucose-restricted conditions and KRAS–MAPK signalling activation, uridine serves as a nutrient source for PDA cells. This aligns with previous studies, where uridine rescued glucose deprivation-induced stress in astrocytes and neurons[20,21,23]. Indeed, others have shown that the uridine-mediated rescue is UPP1-dependent and induced by glucose availability, and it was proposed that UPP1 functions in this capacity to support bioenergetics by providing nucleotides[20]. We show that the uridine ribose ring fuels both energetic and anabolic metabolism in PDA cells. We also found that in addition to KRAS–MAPK, uridine utilization axis is regulated by a yet unknown rheostat sensing the upstream availability of glucose and/or uridine. These are newly identified regulators of UPP1, adding to p53 regulation[37]. Exploration of these regulatory pathways in PDA and other cancers may hold translational promise.

Nutrients in the TME can be derived from serum or the various cell types that make up the tumour. Although discovery methods involving conditioned medium and metabolomics are high-throughput, they tend not to capture the complex metabolic interactions of the TME. Biolog assays provide an unbiased approach to assess metabolic fuel utilization. Here we used this system to first obtain source-agnostic information about the nutrients utilized by PDA cells, before conducting a targeted analysis of potential sources providing exogenous uridine to tumours. We observed evidence of uridine enrichment in the TIF of mouse pancreatic tumours, tumour consumption of plasma-derived uridine by in vivo isotope tracing, reduction of the plasma uridine pool upon whole-body macrophage depletion, and in vitro micromolar release of uridine from naive macrophages and TAMs. Together, these findings illustrate the complexity of nutrient availability in the TME and suggest a model in which cells inside and outside PDA tumours fuel cancer metabolism with uridine.

There is a growing appreciation for the importance of nucleosides in cancer, including inosine, thymidine and deoxycytidine[6,14,38,39]. Inosine is consumed in melanomas by both cancer and CD8[+] effector T cells[40]. The upregulation of nucleoside usage under nutrient deprivation[40–42], especially in immune and PDA cells, supports the idea that metabolic competition contributes to immunosuppression and tumour progression. Along these lines, we show that UPP1-KO in an orthotopic syngeneic model of PDA severely blunts tumour growth, thus the UPP1–uridine scavenging axis is important for PDA cells. RNA is another important source of uridine for glucose-starved cells and may be relevant for PDA cells, which readily scavenge intracellular (that is, autophagy) and extracellular biomolecules to fuel metabolism[5]. There are also several non-tumoural cell types that utilize uridine for various purposes[20,21,23,43], and this complexity is a promising area for future study. Collectively, our data identify the uridine–UPP1 axis as a driver of compensatory metabolism and support the therapeutic tractability of nucleoside metabolism in solid tumours.

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

# Methods

## Cell culture

The PDA cell lines A549, HT1080, HCT116 and U2OS and human pancreatic nestin-expressing cells were purchased from the American Type Culture Collection (ATCC) or the German Collection of Microorganisms (DSMZ). The human pancreatic stellate cells (hPSC) and mouse PDA cell lines KPC 7940b and MT3-2D were provided under a material transfer agreement by R. Hwang, G. Beatty and D. Tuveson, respectively. iKras cell lines A9993 and iKRAS 9805 were derived as described[33]. The identity of cell lines was confirmed by STR profiling, and lines were routinely tested for mycoplasma using MycoAlert (Lonza, LT07-318). For routine propagation, unless otherwise indicated, all cell lines were cultured in high-glucose DMEM (Gibco, 11965092) supplemented with 10% FBS (Corning, 35-010-CV) at 37 °C and 5% $CO_2$. PBS (Gibco, 10010023) was used for cell washing steps unless otherwise indicated. For treatments, the following inhibitors were used: MYC, fedratinib (MedChemExpress, HY-10409) and 10058-F4 (Cayman Chemical, 15929); MEK1, trametinib (Selleckchem, S2673).

## Biolog metabolic assay

In the initial phenotypic screen, the 22 cell lines were grown in 96-well PM-M1 and PM-M2 plates (Biolog, 13101 and 13102). The assay was set up such that one well was used per test metabolite substrate, accompanied by three replicates of positive (glucose) and negative (blank) control wells. The RMA from substrate catabolism in the cells was measured using Biolog Redox Dye Mix MB. In brief, the cell lines were counted, and their viability assessed using Trypan Blue Dye (Invitrogen, T10282). The cells were then washed two times with Biolog Inoculating fluid IF-M1 (Biolog, 72301) to remove residual culture medium. Then, a cell suspension containing 20,000 cells per 50 µl was prepared in Biolog IF-M1 containing 0.3 mM glutamine and 5% dialysed FBS (dFBS) (Hyclone GE Life Sciences, SH30079.01) and plated into PM-M1 and PM-M2 96-well plates at 50 µl per well. Plates were incubated for 24 h at 37 °C and 5% $CO_2$, after which 10 µl Biolog Redox Dye Mix MB (Biolog, 74352) was added to each well. Plates were sealed to prevent the leakage of $CO_2$. The reduction of the dye over time was measured as absorbance (A590-A750) using the OmniLog PM-M instrument (Biolog, 93171) for 74.5 h at 15 min intervals. To account for proliferation or cell number in the Biolog screening assay, CyQUANT was used for normalization.

The data were processed and normalized using the opm package[45] version 1.3.77 in the R statistical programming tool. After removing CFPAC1 (atypically high signal across the plate), the maximum metabolic activity per cell line was taken as its main readout for substrate avidity (Extended Data Fig. 1a) and normalized by subtracting the median negative control signal for a given cell line from all other values for that cell line. Heat map visualization of the data was plotted using heatmap2 and ComplexHeatmap packages in R.

## Correlation of Biolog metabolites to gene expression of enzymes

High-confidence metabolites from the Biolog screening assay were correlated (Spearman Correlation) to gene expression data for enzymes associated with metabolite usage. Genes with high correlation co-efficient to a given metabolite were chosen for further analysis.

## NADH assay

Cells were seeded in 96-well plates at 10,000 cells per well directly into the indicated medium conditions. Following the incubation period at 37 °C and 5% $CO_2$ (24, 48 or 72 h), MTT (Thermo Scientific, L1193903) was added directly to the wells containing medium. Cells were incubated at 37 °C and 5% $CO_2$ for 1 h, after which the medium and MTT reagent were carefully removed. Next, to each well, 50 µl of DMSO (Sigma-Aldrich, D2650) was added followed by 5 min incubation at room temperature before measuring absorbance at 570 nm.

## CyQUANT proliferation assay

Cells were seeded at 20,000 cells per well for the screening study or 2,000 cells per well for proliferation assays in growth medium in 96-well plates (Corning, 3603). For proliferation assays, the culture medium was removed the next day, followed by a gentle 1× wash with PBS. Treatment medium was then applied, and the cells incubated at 37 °C and 5% $CO_2$ until they reached ~70% confluence. Medium was then carefully aspirated, and the plate with cells attached was moved to −80 °C for at least 24 h to ensure complete cell lysis. To prepare the lysis buffer and DNA dye, CyQUANT Cell Lysis Buffer and CyQUANT GR dye (Invitrogen, C7026) were diluted in water at 1:20 and 1:400, respectively. The frozen cells were then thawed and 100 µl of the lysis buffer was added to each well. Thereafter the plate was covered to prevent light from inactivating the GR dye and was placed on an orbital shaker for 5 min before measurement. Fluorescence from each well, indicating GR dye binding to DNA, was then measured utilizing a SpectraMax M3 Microplate Reader with SoftMax Pro 5.4.2 software at an excitation wavelength of 480 nm and an emission wavelength of 520 nm.

## ATP-based viability assay

Cells were seeded in quadruplicates at a density of 2,500–5000 cells in 100 µl DMEM per well of the white walled 96-well plates (Corning/Costar, 3917). Next day, the medium was aspirated, each well was washed with 200 µl PBS after which treatment medium was introduced. At the end of the experiment duration (48 or 72 h), relative proliferation was determined with CellTiter-Glo 2.0 Cell Viability Assay Kit (Promega, G9243) and the luminescence quantified using a SpectraMax M3 Microplate Reader.

## Live-cell proliferation assay

Cells were seeded in 96-well plates at 1,000 cells per well in 100 µl of growth medium and incubated overnight at 37 °C, 5% CO2. After 24 h, medium was changed, and the cells were incubated for a further 72 h during which cell proliferation was determined by live-cell imaging on a BioSpa Cytation.

## UPP1 CRISPR–Cas9 knockout

The expression vector pspCas9(BB)-2A-Puro (PX459) used to generate the UPP1 CRISPR–Cas9 constructs was obtained from Addgene (Plasmid 48139). The plasmid was cut using the restriction enzyme BbsI followed by the insertion of human or mouse uridine phosphorylase 1 sgRNA sequences (Supplementary Table 4), as previously described[46]. The human and mouse sequences were obtained from the Genome-Scale CRISPR Knock-Out (GeCKO) library. For transfection, the human or mouse PDA cells were seeded at 150,000 cells per well in a 6-well plate a day earlier. The cells were transfected with 1 µg of plasmid pSpCas9-UPP1 using Lipofectamine 3000 Reagent (Invitrogen, L3000001) according to manufacturer's instruction. After 24 h, the selection of successfully transfected cells was commenced by culturing the cells with 0.3 mg ml$^{-1}$ puromycin in DMEM. The puromycin-containing medium was replaced every two days until selection was complete, as indicated by the death and detachment of all non-transfected cells. Thereafter, the successfully transfected cell lines were expanded and clonally selected after serial dilution.

## siRNA experiments

Approximately $5 \times 10^5$ ASPC1 cells were seeded per 6-cm dish for 24 h in the growth medium. On day 2, medium was changed and the respective SMARTpool siRNA-containing medium was added. siRNA transfection was performed using Lipofectamine RNAiMAX (ThermoFischer Scientific, 13778075) according to the manufacturer's instructions. For the transfection, Opti-MEM Reduced Serum Medium (31985-062) was used and siRNAs were added at a concentration of 20 nM. Cells were transfected for 48 h after which the cells were trypsinized,

counted, and plated for MTT assay. The remaining cells were pelleted, and RNA was extracted for qPCR. The siRNAs used were as follows: non-targeting control (D-001810-01-05), SMARTpool ON-TARGETplus Human PGM2 (55276) (L-020785-01-0005), ON-TARGETplus Human UCK1 (83549) (L-004062-00-0005) and ON-TARGETplus Human UCK2 (7371) (L-005077-00-0005).

## Reverse transcription with qPCR

Cells were seeded at a density of $5 \times 10^5$–$8 \times 10^5$ cells per well, allowed to attach overnight, treated where applicable and pelleted after 24 h. RNA samples were isolated using the RNEasy Plus Mini Kit (QIAGEN, 74134) according to the manufacturer's instructions. RNA purity was assessed using a NanoDrop One (ThermoFisher Scientific, ND-ONE-W). Thereafter, 1 µg of the RNA samples were reverse transcribed to cDNA using the iScript cDNA Synthesis Kit (Bio-Rad, 1708890) according to the accompanying instructions. qPCR was performed on the Quant-Studio 3 Real-Time PCR System (ThermoFisher Scientific, A28131) or Applied Biosystems StepOne Plus instrument (software version 2.3) using Power SYBR Green PCR Master Mix (ThermoFisher Scientific, 4367659). The reactions were run at 10 µl total volume consisting of 5 µl SYBR, 2 µl nuclease free water, 2 µl of cDNA after diluting 1:4 in water, 0.5 µl of 10 µM forward (F) primer, and 0.5 µl 10 µM reverse (R) primer. Primer sequences are listed in Supplementary Table 5. The gene expression was calculated as $\Delta C_t$ and $RPS21$ or $ACTB$ was used as a housekeeping gene.

## Western blotting

Following culture, medium was aspirated, and the wells washed one time with PBS. Thereafter, 100 µl of radioimmunoprecipitation assay buffer (Sigma-Aldrich, R0278) to which phosphatase and protease inhibitors were added, as transferred to each well to lyse the cells. Lysis and the collection of the lysates were completed on ice. Following a 5- to 10-minute incubation on ice, lysates were collected into 1.5 ml Eppendorf tubes and centrifuged at 4 °C for 10 min at 18,000$g$ to extract the sample supernatant. Protein concentration of the samples for western blot analysis were measured using Pierce BCA Protein Assay Kit (ThermoFisher, 23227) according to the manufacturer's instructions. For the running step, samples were loaded at 20–25 µg protein per lane along with the SeeBlue Plus2 protein ladder (Invitrogen, LC5925) and run at 120 V on an Invitrogen NuPAGE 4–12% Bis-Tris gel (ThermoFisher, NP0336BOX). Thereafter, the separated proteins were transferred to methanol-activated PVDF membranes (Millipore) at 25 V for 1 h. Following this, membranes were immersed in block-ing buffer (5% blotting-grade blocker (Bio-Rad, 1706404) in TBS-T solution: tris-buffered saline (Bio-Rad, 1706435) with 0.1% Tween-20 (Sigma-Aldrich, 9005-64-5)) for ~1 h on a plate rocker at room tempera-ture. Next, membranes were washed 3 times with TBS-T at 10 min per wash, immersed in the indicated primary antibodies, and incubated overnight at 4 °C on a plate rocker. The antibodies used were diluted in blocking buffer at dilutions recommended by the manufacturer. The following day, the primary antibody was removed, and the membrane was washed 3 times with TBS-T and on a plate rocker for 5 min per wash. Immediately after, the membrane was incubated for 1 h and with gentle rocking at room temperature in the appropriate secondary antibody diluted 1:10,000 in TBS-T. Lastly, the membrane was washed 3 times in TBS-T at 10 min per wash and incubated in chemiluminescence rea-gent (Clarity Max Western ECL Substrate, 705062) according to the manufacturer's instructions. Subsequently, blot images were acquired on a Bio-Rad ChemiDoc Imaging System (Image Lab Touch Software version 2.4.0.03). The following primary antibodies were used in this study and at 1:1,000 dilution: anti-UPP1 (Sigma-Aldrich, HPA055394), anti-c-MYC (Cell Signaling, 5605S), anti-pERK (Cell Signaling, 9106L), anti-ERK (Cell Signaling, 9102S), anti-PGM2 (Invitrogen, PA5-31378), and anti-Vinculin (Cell Signaling, 13901S). The following secondary antibodies were used: anti-rabbit-HRP (Cell Signaling, 7074S), and anti-mouse-HRP (Cell Signaling, 7076P2). The uncropped, unprocessed images of the western blots are presented in Supplementary Figs. 1–9.

## Mouse tumour studies

Animal studies were performed at the University of Michigan, the Insti-tute of Cancer Research (ICR), and the University of Chicago according to approved protocols. Specifically, for studies at the University of Michigan, the Institutional Animal Care and Use Committee (IACUC) PRO00010606 was followed; Institute of Cancer Research studies conformed to UK Home Office Regulations under the Animals Sci-entific Procedures Act 1986 and national guidelines (project licence P0A54750A protocol 5); University of Chicago IACUC protocol 72587 was followed. Mice were housed in a pathogen-free animal facility at a maximum of five mice per cage with a 12 h light/12 h dark cycle, 30–70% humidity and 20–23 °C temperatures. Mice were provided water and fed ad libitum with chow (5L0D – PicoLab Laboratory Rodent Diet).

## Pancreatic tumour models

For mouse studies at the University of Michigan, male and female 6- to 8-week-old C57BL/6J mice were obtained from The Jackson Labora-tory (strain 000664) and maintained in the facilities of the Unit for Laboratory Animal Medicine (ULAM) under specific pathogen-free conditions. Prior to tumour cell injection, wild-type or UPP1-KO mouse cell lines (either derived from MT3-2D or KPC 7940b) were collected from culture plates according to standard cell culture procedures. The cells were counted, washed once with PBS and resuspended in 1:1 solution of serum-free DMEM and Matrigel (Corning, 354234). For the orthotopic surgical procedure, mice were anaesthetized using inha-lation anaesthesia. The surgical site was sterilized by swapping with iodine (Povidine-Iodine Prep Pad, PDI, B40600). This was followed by incision on the left flank using sterilized instruments. Thereafter, the cell lines were injected into the pancreas and the incision sutured. Cell injection was as follows: 50,000 or 100,000 cells in 50 µl final volume for orthotopic implantation or ~1 × 10$^6$ cells in 100 µl final volume for subcutaneous implantation. Animals were monitored regularly and all orthotopic experiments were concluded ~3–4 weeks after injection.

For the studies at the Institute of Cancer Research, female ~6-week-old C57BL/6NCrl mice were purchased from Charles River Laboratories (strain 027). Prior to tumour cell injection, MT3-2D sgVector (sgV) and the UPP1-KO cells (sg1 and sg3) were trypsinized according to standard cell culture protocols. Cells were washed with PBS and resuspended in 1:1 Hank's balanced salt solution (HBSS; Gibco, 14025092) and Matrigel (Corning, 354234). Following surgical incision, 50,000 cells in 20 µl final volume were injected into the pancreas. Tumour growth was monitored by palpating three times per week. Studies were terminated when the animals injected with the vector reached a high tumour burden based on palpation.

For the studies at the University of Chicago, C57BL/6J mice 8–12 weeks of age were purchased from Jackson Laboratories (strain 000664). About $2.5 \times 10^5$ cells per tumour were resuspended in 20 µl of 5.6 mg ml$^{-1}$ Cultrex Reduced Growth Factor Basement Membrane Extract (RGF BME; R&D Biosystems, 3433-010-01) and serum-free RPMI (SF-RPMI) solu-tion. The basement membrane extract–cell mixture was injected into the splenic lobe of the pancreas of the mice, as previously described[43]. After implantation, end point was determined by abdominal palpation and daily monitoring of body weight.

## Macrophage depletion

The KPC 7940b cell line was orthotopically implanted into approxi-mately 8-week-old male and female C57BL/6J mice, as above. Two weeks after injection, mice were randomized into two groups: control or mac-rophage depletion. The control group mice were treated on day 1 with 1 mg IgG (InVivoMAb rat IgG1 Isotype control, anti-trinitrophenol, BE020, Bio X Cell) and on day 2 with 200 µl Control Liposome (PBS) (CP-005-005, Liposoma). The macrophage depletion mice were treated

on day 1 with InVivoMAb anti-mouse CSF1 (anti-CSF1, BE0204, Bio X Cell) and day 2 with 200 µl clodronate liposomes (CP-005-005, Liposoma). Two subsequent treatment sequences were administered at 2-day intervals as follows: 0.5 mg IgG or anti-CSF1 followed the next day by 200 µl control liposome or clodronate liposomes for the control and depletion groups, respectively. The experiment was terminated after one week. From each mouse, blood samples were collected into EDTA BD Vacutainer K2 EDTA 3.6 mg (36784) and centrifuged at 200g for 5 min for plasma collection. In addition, tumours were collected, weighed and used for the extraction of TIF, as below.

### TIF collection

TIF was isolated from tumours as described[30]. In brief, tumours were rapidly dissected after euthanizing the animals. Tumours were weighed and rinsed in blood bank saline solution (150 mM NaCl) and blotted on filter paper (VWR, 28298–020) until dry. Tumour isolation was done in less than 3 min to minimize the time the tumour was ischaemic prior to TIF isolation. Tumours were cut in half and put onto 20-µm nylon mesh filters (Spectrum Labs, 148134) on top of 50 ml conical tubes, and centrifuged for 10 min at 4 °C at 400g. TIF was then processed for metabolomics in a similar manner as plasma, as described below.

### In vivo delivery of isotopically labelled uridine

Uridine-derived ribose carbon was traced in vivo using [$^{13}C_5$] ribose-labelled uridine (Cambridge Isotope Laboratories, CLM-3680-PK). Specifically, mice bearing orthotopic or subcutaneous tumours were generated, as described above, using KPC 7940b cell lines. For the orthotopic tumours, after the tumours became palpable (3 weeks after implantation), mice were injected intraperitoneally with 200 µl of 0.2 M [$^{13}C_5$]uridine. For the subcutaneous models, 50 µl of 0.2 M [$^{13}C_5$]uridine was injected directly into the tumours 3 weeks after implantation. Tumours were collected 1 h after uridine injection and processed for isotope tracing, as detailed below.

### Mass spectrometry-based metabolomics

**Metabolomics sample preparation.** For in vitro extracellular (medium) and intracellular metabolomic profiling, PDA cells were seeded in triplicates in a 6-well plate at $4–6 × 10^6$ cells per well in growth medium. A parallel plate for protein estimation and sample normalization was also set up. After overnight incubation, the culture medium was aspirated and replaced with medium containing treatments or supplemented metabolites of interest. The cells were then cultured for a further 24 h. Thereafter, for extracellular metabolites, 200 µl of medium was collected from each well into a 1.5 ml Eppendorf tube and to that 800 µl ice-cold methanol was added. For intracellular metabolites, the remaining medium was aspirated, and samples washed once with 1 ml PBS before incubation with 1 ml ice-cold 80% methanol on dry ice for 10 min. Thereafter, cell lysates were collected from each well and transferred into separate 1.5 ml Eppendorf tubes. The samples were then centrifuged at 12,000g. For each experimental condition, the volume of supernatant to collect for drying with SpeedVac Vacuum Concentrator (model: SPD1030) was determined based on the protein concentration of the parallel plate.

For tumours, the samples were flash frozen in liquid nitrogen upon collection. Tumours of approximately equal weight (<100 mg) were collected per sample per experimental group. The tumours were then put into 2 ml Eppendorf tubes to which 1 ml of ice-cold 80% methanol (diluted in 20% $H_2O$). Metallic beads were added to each tube and samples were shaken and homogenized on an Retsch TissueLyser II (129251128) in intervals of 30 s until fully homogenized. Samples were then centrifuged at 12,000g and supernatant collected for further processing.

**Targeted metabolomics.** The collected supernatants were dried using SpeedVac Vacuum Concentrator, reconstituted with 50% v/v methanol in water, and analysed by LC–MS, as described in detail previously[47]. Data were analysed with Agilent Masshunter Workstation Quantitative Analysis for QQQ version 10.1, build 10.1.733.0.

**Stable isotope tracing.** For stable isotope tracing in cells, [$^{13}C_5$]uridine (Cambridge Isotope Laboratories, CLM-3680-PK) was supplemented at 0.1 mM or 1 mM for in vitro assays. In brief, wild-type or UPP1-KO cells were cultured overnight in regular medium. Next day, cells were washed once followed by the introduction of medium containing the indicated amounts of glucose, dialysed FBS and supplemented with labelled uridine. In parallel, a similar experiment was set up for unlabelled uridine. The cell lines were then cultured in the uridine-supplemented medium for 24 h or as otherwise indicated, followed by sample collection, as detailed for unlabelled intracellular metabolomics above. Labelled tumours were similarly collected as detailed above. Samples were prepared for time-of-flight mass spectrometry, as described in detail previously[47], and analysed with Agilent MassHunter Workstation Profinder version 10.0, build 10.0.10062.0.

For the experiments were glucose and uridine concentrations were varied (5 and 0.1 mM and 1 and 0.1 mM, respectively) followed by stable isotope tracing, the cells were seeded at a density of 500,000 and treated with [$^{13}C_5$]uridine or unlabelled uridine in DMEM supplemented with dialysed FBS and the indicated concentration of glucose for 24 h. Then cells were washed with 1 ml cold PBS followed by the addition of 1 ml 2:2:1 methanol:acetonitrile:water at −20 °C to the wells on dry ice for 10 min. Cells were scraped from the dish. Next, samples were subjected to 3 cycles of 30 s vortex, 1 min liquid $N_2$, and 10 min 25 °C bath sonicate. Samples were then stored at −20 °C overnight and centrifuged at 14,000g at 4 °C for 10 min. A volume of 860 µl of the supernatant was transferred to a new tube and dried by SpeedVac Vacuum Concentrator. Protein pellets were also dried similarly to remove excess supernatant, resuspended in 400 µl 100 mM NaOH through repeated vortexing, 5 min incubation at 95 °C, and protein quantified by BCA assay (ThermoFisher, 23227). Dried supernatant pellets were resuspended in 2:1 acetonitrile:water at 1 µl per 2.5 µg protein and subjected to 2 cycles of: 5 min 25 °C bath sonicate, 1 min vortex. Samples were incubated at 4 °C overnight, then centrifuged at 14,000g at 4 °C for 10 min and the supernatant transferred to liquid chromatography vials and stored at −80 °C until analysis. For sample analysis, 4 µl of metabolite extracts were run on an Agilent 6545 Q-TOF Mass Spectrometer and an Agilent 1290 Infinity II LC system using a iHILIC-(P) Classic 2.1 mm × 100 mm, 5 µm column (HILICON, 160.102.0520) with iHILIC-(P) Classic Guard column (HILICON, 160.122.0520) attached. A column temperature of 45 °C and a flow rate of 250 µl min$^{-1}$ was used. Mobile phases were A: 95% water, 5% acetonitrile, 20 mM ammonium bicarbonate, 0.1% ammonium hydroxide solution (25% in water), 2.5 µM medronic acid and B: 85% acetonitrile, 5% water, 2.5 µM medronic acid. Each sample was subjected to a linear gradient: 0–1 min 90% B, 1–12 min 35% B, 12–12.5 min 25% B, 12.5–14.5 min 25% B, 14.5–15 min 90% B, which was then followed by 4 min at 400 µl min$^{-1}$ and 2 min at 250 µl min$^{-1}$ at 90% B for re-equilibration. Chromatograms for selected metabolites were extracted in Skyline Daily (software version 22.2.1.256) and manually integrated according to an in-house list of standard $m/z$ and retention times. Natural isotope abundance correction was performed, and peak areas plotted.

**Quantification of TIF metabolite levels.** For quantification of uridine and glucose in TIF, quantitative metabolite profiling of fluid samples was performed as previously described[30]. In brief, chemical standards were prepared and serially diluted in high-performance liquid chromatography grade water in a dilution series from 5 mM to 1 µM. Using the external standard library dilutions, we created a standard curve based on the linear relationship of the normalized peak area and the concentration of the metabolite. This standard curve was then used to interpolate the concentration of the metabolite in the TIF sample.

## Clinical samples

Patients with pancreas resections for PDA from 2021 to 2022 at the University of Michigan Health System were included in the study. All haematoxylin and eosin (H&E)-stained slides were reviewed and diagnoses confirmed, and corresponding areas were carefully selected and marked. The collection of patient-derived tissues for histological analyses was approved by the Institutional Review Board at the University of Michigan (IRB number: HUM00098128). Tissues were fixed in 10% neutral buffered formalin and paraffin embedded using standard protocols before sectioning and staining.

## Tissue microarrays

All specimens are from patients with pancreas resections for pancreatitis, cystic neoplasms, or PDA from 2002 to 2015 at the University of Michigan Health System. After fixation in 10% neutral buffered formalin (hours to a couple of days depending on the size of the tissue), samples were embedded in paraffin. All tissues were H&E stained, reviewed, and diagnoses confirmed. Corresponding areas were carefully selected and marked. Duplicate 1 mm diameter tissue cores from a total of 213 patient tissue samples were selectively punched and transferred to recipient tissue array blocks. Five tissue microarrays (TMAs) were set up and H&E and immunohistochemistry staining was performed on each TMA block using standard protocols. The TMA was previously published[48].

## RNAscope

RNAscope was performed as previously described[49], and according to the manufacturer's protocol (ACD: 323100-USM). In brief, paraffin wax was removed with xylene and slides were rehydrated. Samples underwent antigen retrieval for 15 min. Samples were blocked for 30 min at room temperature with CoDetection Antibody Diluent and then incubated overnight at 4 °C with a primary antibody for panCK (Mouse anti-cytokeratin, pan reactive; 1:100; BioLegend; 628602) diluted in CoDetection Antibody Diluent. Protease digestion was performed for 13 min at room temperature. Human *UPP1* RNAscope probe (ACD; 509279; Lot: 21272A) was added to slides for 2 h at 40 °C. Samples were incubated with TSA-Cy3 fluorophore (1:2,000; Akoya Biosciences; NEL704A001KT) diluted in CoDetection Antibody Diluent. Following HRP blocking, slides were rinsed in PBS + 0.1% Tween-20 (PBST). Slides were stained with DAPI (1:30,000; Millipore Sigma; 10236276001) diluted in PBS for 15 min at room temperature. After rinsing in PBST, slides were incubated for 45 min at room temperature with secondary antibodies diluted 1:500 in CoDetection Antibody Diluent. Slides were rinsed with PBST and mounted in ProLong Gold Antifade Mountant (Invitrogen, P36930). Sections were visualized on a Leica SP5X upright confocal. For quantitation, 20× fields of view were imaged and analysed using FIJI/Image J (version 1.53c). For analysis, images were converted to 16 bit, the threshold was adjusted, and the area of *UPP1* expression was measured per 20× image.

## Immunohistochemistry.

Patients tissue slides were deparaffinized and rehydrated with graded Histo-Clear (National Diagnostics), ethanol, and water. Slides were quenched for 15 min in a methanol solution containing 1.5% hydrogen peroxide before antigen retrieval. Samples underwent antigen retrieval with sodium citrate buffer (2.94 g l⁻¹ sodium citrate, 0.05% Tween-20, pH 6). Samples were blocked using blocking buffer (5% bovine serum albumin, 0.2% Triton X-100, in PBS) for 1 h at room temperature. After blocking, slides were incubated overnight at 4 °C with primary antibody (rabbit anti-UPP1; 1:200; Sigma-Aldrich, HPA055394) diluted in blocking buffer. Slides were rinsed in PBS and incubated for 1 h at room temperature with a biotinylated secondary antibody (horse anti-rabbit; 1:500; Vector Labs, BA-1100). After rinsing, slides were prepared for a colour reaction by incubating with Vectastain Elite ABC Reagent (Vectastain Elite ABC-HRP Kit; Vector Labs, PK-6100) for 30 min at room temperature. Sections

were developed with DAB (DAB Substrate Kit; Vector Labs, SK-4100) for 2 min, rinsed, and counterstained with haematoxylin. Slides were mounted in Permount Mounting Medium (Fisher). After drying, slides were imaged using an Olympus BX53F microscope, Olympus CP80 digital camera, and CellSens standard software.

Mouse tumours were fixed in 10% neutral buffered formalin for 48 h and embedded in paraffin as formalin-fixed paraffin-embedded (FFPE) blocks. Serial sections of 4 μm thickness were cut from FFPE blocks, deparaffinized in xylene, processed in graded alcohol, and rehydrated in water. One section was stained with H&E for histological analysis. The Dako Autostainer Link 48 automated immunostaining platform was used for all the below immunostainings. Anti-Cd31 monoclonal antibody (SZ31, DIA-310, Dianova) was used at a 1:75 dilution, followed by heat-induced epitope retrieval for 20 mins at 97 °C using a PT Link module (Agilent). Anti-Cd8 monoclonal antibody (clone 4SM15, 14-0808, eBioscience) was used at a 1:200 dilution, and anti-F4/80 monoclonal antibody (Clone-A3-1, MCA497G, Bio-Rad) was used at a 1:100 dilution. For these antibodies, EnVision FLEX Target Retrieval Solution (high pH; K800421-2, Agilent) and Nichirei anti-rat Histofine polymer reagent (41491F, Nichirei Biosciences) primary antibody detection kits were used. For CD3, polyclonal antibody (ab5690, Abcam) was used at a 1:400 dilution with EnVision FLEX Target Retrieval Solution (low pH of 6) for 20 mins in PT link module and detected with Vector Rabbit ImmPRESS HRP horse anti-rabbit IgG polymer kit (Vector Laboratories, MP-7401-50). Appropriate positive and negative controls were used in all runs. The Nanozoomer-XR C12000 (Hamamatsu) was used to scan whole stained sections. Antigen expression was scored using Definiens Test Studio Software (Definiens). F4/80 was quantified using Image J.

Immunohistochemistry of UPP1 expression in human normal and PDA tissues was also accessed from the Human Protein Atlas portal[50].

## PDA dataset analysis

The human PDA microarray datasets with accession numbers GSE71729 (*n* = 46 normal pancreas vs 145 tumour tissues) and GSE62452 (*n* = 61 non-tumoural vs 69 tumour tissues) were obtained from NCBI GEO[51]. Differential gene expression between PDA and non-tumours were performed in R using the limma package (version 3.38.3). Kaplan–Meier overall survival (log-rank test) was performed after splitting the tumour samples per dataset into *UPP1*-high and *UPP1*-low subsets. For Kaplan–Meier analysis, human PDA tumour datasets and the accompanying clinical data from the following sources were used: GSE71729 (*n* = 145), TCGA data (*n* = 146), International Cancer Genome Consortium (ICGC, *n* = 267), and Puleo et al. (2018) (*n* = 288)[44]. The iKras mice data were obtained from NCBI GEO under the accession number GSE32277. TCGA expression data of tumours with *KRAS* wild type (*n* = 43) and *KRAS*^G12D^ mutation (*n* = 42) were used to determine the relative expression of *UPP1* in *KRAS*^G12D^ mutated tumours.

## Pan-cancer dataset analysis

TCGA pan-cancer datasets including bladder, colon, oesophageal, lung, head and neck, prostate cancer and glioblastoma, were downloaded from Xena Platform from University of California Santa Cruz[52]. An additional colorectal dataset (GSE44076) was also used. For the comparisons, the normal or adjacent matched and unmatched normal samples were used. In total, 2,828 cancer tissue samples and 379 non-tumoural control tissue samples were analysed. These datasets were used to compare *UPP1* expression between cancer and non-cancer tissues.

## CCLE gene analysis and PDA tumour data stratification

Gene expression data for uridine high and uridine low metabolizers were extracted from the CCLE (GSE36133). The subsets were then compared using the limma package in R to determine the differentially expressed genes in uridine-high metabolizers and/or consumers relative to the lower metabolizers and/or consumers. For the tumour stratification, samples in the dataset GSE71729 (*n* = 145) were ranked into

*UPP1*-high and *UPP1*-low groups and compared as above to determine the genes differentially expressed in *UPP1*-high tumours. UPP1 protein expression analysis was performed in *KRAS* mutant and wild-type cell lines using data from DepMap[53].

## Pathway analyses

Pathway analyses were performed using DAVID functional annotation platform (https://david.ncifcrf.gov/, version 6.8) or gene set enrichment analysis (GSEA, version 4.0.3) with GSEAPreranked option. Ranking of genes was based on the product of the logFC and −log(*P* value). GSEA was run with default parameters, except gene set size filter set at min = 10. Gene ontology analyses were performed with DAVID.

## Promoter analysis of *UPP1*

CiiDER[54] was used for predicting *UPP1* gene transcription factor sites. DNA sequence flanking the *UPP1* transcription start site (1,500 bases upstream, 500 bases downstream) was used to compare to JASPAR2020_CORE_vertebrates position frequency matrix model to generate a score of similarity. As transcription factor binding sites are variable and binding sites rarely match the model perfectly, a default deficit score of 0.15 was used, where deficit score of 0 represents prefect match. Top 10 transcription factors were obtained using the predicted UPP1-binding sites with respect to sequences from the human genome (GRCh38.94) and mouse genome (GRCm38.94).

## Statistical analysis

Statistics were performed either with GraphPad Prism 8 (GraphPad Software Inc.) or using R version 3.5.2. Data from experimental groups were compared using the two-tailed *t*-test or analysis of variance (ANOVA) with post hoc corrections where applicable, and between biological (or in vitro) replicates. Data in all graphs represent the mean ± s.d. Statistical significance was accepted if *P* < 0.05. For data analysis and visualization in R, packages (with versions) used include dplyr (0.8.3), ggplot2 (3.3.5), gplots (3.0.1, heatmap.2 function), ComplexHeatmap (2.3.5), tidyverse (1.3.0) and VennDiagram (1.6.20).

## Statistics and reproducibility

Figure 1. a, The use of >175 metabolites by 19 PDA cell lines and 2 non-PDA pancreatic cell lines was measured every 15 min for ~3 days (74.5 h) using the Biolog OmniLog device. The assay readout, RMA, was correlated with the expression level of metabolic genes in cell lines; human PDA data were used for subsequent analyses. Nutrient-deficient medium, no glucose, 0.3 mM glutamine and 5% dialysed FBS. c, *n* = 4 biologically independent samples per group. Statistical significance was measured by multiple unpaired two-tailed *t*-tests (two-stage step-up method) comparing RMA from cells in basal medium vs 1 mM uridine medium (both glucose-free), ****P < 0.0001. The experiments were performed twice with similar results. d, *n* = 4 biologically independent samples per cell line. The experiment was performed once.

Figure 2. a, *n* = 4 biologically independent samples per group per cell line. Statistical significance was measured using one-way ANOVA with Dunnett's multiple comparisons test. CAPAN2 (comparison between no glucose/uridine and no glucose + 1 mM uridine, ***P = 0.0001; no glucose/uridine and 1 mM glucose/no uridine, *P = 0.021; no glucose/uridine and 0.1 mM ribose, *P = 0.86; no glucose/uridine and 1 mM ribose, **P = 0.0093; no glucose/uridine and 10 mM ribose, ****P < 0.0001). PATU8988S (comparison between no glucose/uridine and no glucose + 1 mM uridine, ****P < 0.0001; no glucose/uridine and 1 mM glucose/no uridine, ****P < 0.0001; no glucose/uridine and 0.1 mM ribose, *P = 0.9817; no glucose/uridine and 1 mM ribose, **P = 0.0019; no glucose/uridine and 10 mM ribose, ****P < 0.0001). DANG (comparison between no glucose/uridine and no glucose + 1 mM uridine, ****P < 0.0001; no glucose/uridine and 1 mM glucose/no uridine, ****P < 0.0001; no glucose/uridine and 0.1 mM ribose, *P > 0.9999; no glucose/uridine and 1 mM ribose, *P = 0.3025; no glucose/uridine and

10 mM ribose, ****P < 0.0001). ASPC1 (comparison between no glucose/uridine and no glucose +1 mM uridine, ****P < 0.0001; no glucose/uridine and 1 mM glucose/no uridine, ****P < 0.0001; no glucose/uridine and 0.1 mM ribose, *P = 0.9974; no glucose/uridine and 1 mM ribose, *P = 0.0103; no glucose/uridine and 10 mM ribose, ****P < 0.0001). The experiment was performed once. b,c, *n* = 3 biologically independent samples. Statistical significance was measured using two-tailed unpaired *t*-test. Intracellular: comparison between no uridine and 1 mM uridine: ***P = 0.0005 (uridine), ****P < 0.0001 (uracil); extracellular: comparison between no uridine and 1 mM uridine: ****P < 0.0001 (uridine), **P = 0.008 (uracil). d–f, *n* = 3 biologically independent samples per cell line. 'Others' indicates M other than M+0 or M+5, where applicable. Bars shown for PATU8988S are same as the WT bars (where applicable) for that cell line in the Extended Data Fig. 5. Tracing experiments were performed twice in these cells with similar results. g, Number of samples: sub-Q, tumours from 3 mice injected on the left and right flanks; ortho, tumours from 4 mice. Mode of uridine injection is intratumoural for sub-Q and intraperitoneal for ortho. h, Median concentration of uridine = 24.1 μM; median concentration of uracil = 90.2 μM; *n* = 22 biologically independent TIF samples. i, Median concentration of glucose = 3.71 mM (plasma) and 0.63 mM (TIF). *n* = 8 biologically independent plasma samples and 8 TIF samples extracted from 8 tumour samples from same mice. These samples are from the control group of the study in Fig. 4a. Statistical significance was measured with two-tailed unpaired *t*-test with Welch's correction, ****P < 0.0001. j,k, j shows the mass isotopologue distribution in uridine and k shows in the indicated metabolites. *n* = 4 biologically independent samples per group per cell line. 'Others' indicates M other than M+0 or M+5, where applicable. Data in a–k are shown as mean ± s.d. The metabolomics experiments (b–k) were performed once.

Figure 3. a, The experiment was performed once. b, *n* = 4 biologically independent samples per group per cell line. Statistical significance was measured using one-way ANOVA with Tukey's multiple comparisons test. PATU8988S, comparison between no uridine (−) and 1 mM uridine (+): ****P < 0.0001, *P* = 0.9703 and *P* = 0.9089 for WT, 1A and 1B groups, respectively. ASPC1, comparison between no uridine (−) and 1 mM uridine (+): ****P < 0.0001, *P* > 0.9999 and *P* > 0.9999 for WT, 1A and 1B groups, respectively. The experiments were performed three times with similar results. c, *n* = 4 biologically independent samples per group per cell line. Statistical significance was measured using one-way ANOVA with Tukey's multiple comparisons test. PATU8988S, comparison between no uridine (−) and 1 mM uridine (+): ****P < 0.0001, *P* > 0.9999 and *P* = 0.9599 for WT, 1A and 1B groups, respectively. ASPC1, comparison between no uridine (−) and 1 mM uridine (+): ****P < 0.0001, *P* = 0.9977 and *P* = 0.6537 for WT, 1A and 1B groups, respectively. The experiments were performed twice with similar results. d, *n* = 3 biologically independent samples per group. Statistical significance was measured using one-way ANOVA with Dunnett's multiple comparisons test. Comparison between WT and clonal cells 1A or 1B: ****P < 0.0001 (PATU8988S) and ***P = 0.0003 (ASPC1). Data are part of the metabolomics experiments shown in Extended Data Fig. 5a–c. The metabolomics experiment was performed once. e, *n* = 3 biologically independent samples per group. 'Others' indicates M other than M+0 or M+5, where applicable. Data are part of the metabolomics experiments shown in Extended Data Fig. 5e,h,j for ASPC1. The metabolomics experiment was performed once. f, Statistical significance was measured using two-tailed unpaired *t*-test with Welch's correction. Number of samples and statistical comparison: GSE62452 (NT, 61 vs PDA, 69, ***P = 0001), GSE71729 (middle: NT, 46 vs PDA, 145, *P = 0.0466), GSE71729 (right: primary, 145 vs liver met, PDA, 25, ****P < 0.0001). Box plot statistics: GSE42452 (NT: minimum = 3.582, maximum = 5.633, 25th percentile = 4.036, 75th percentile = 4.504, median = 4.262; PDA: minimum = 3.853, maximum = 5.989, 25th percentile = 4.37, 75th percentile = 4.843, median = 4.535); GSE71729 (NT: minimum = 2.18, maximum = 4.402, 25th percentile = 2.901, 75th percentile = 3.469,

median = 3.139; PDA: minimum = 2.293, maximum = 4.725, 25th percentile = 3, 75th percentile = 3.657, median = 3.339); GSE71729 (primary: minimum = 2.293, maximum = 4.725, 25th percentile = 3, 75th percentile = 3.657, median = 3.339; liver metastasis: minimum = 3.306, maximum = 5.768, 25th percentile = 3.564, 75th percentile = 4.498, median = 4.023). g, Representative images from patient 1 of 3 tumour tissues. PanCK, pan-cytokeratin, stain indicates tumour cells. i, Number of samples: *UPP1*-low, 144; *UPP1*-high, 144. j, Number of samples: no alteration, 43; G12D, 42. Statistical significance was measured using two-tailed unpaired *t*-test with Welch's correction, **$P = 0.0029$. Box plot statistics: no alteration: minimum = 7.797, maximum = 10.66, median = 9.019, 25th percentile = 8.307, 75th percentile = 9.53; $KRAS^{G12D}$: minimum = 8.154, maximum = 11.3, median = 9.385, 25th percentile = 9.019, 75th percentile = 9.905. k, $n = 3$ biologically independent samples per cell line. Statistical significance was measured using two-tailed unpaired *t*-test. Comparison between Dox (−) and (+) in iKras* cell A9993: ***$P = 0.0002$; in iKras cell 8905: **$P = 0088$. The experiment was performed once. l, Vinculin is used as a loading control. The experiment was performed once. m, 3 biologically independent samples per group. Statistical significance was measured using two-tailed unpaired *t*-test. Comparison between cells cultured in uridine/glucose-containing medium with and without trametinib treatment: ****$P < 0.0001$; comparison between cells treated with and without trametinib in the presence of glucose but no uridine: ****$P < 0.0001$; comparison between cells treated with and without trametinib in the presence of uridine and no glucose: ****$P < 0.0001$; comparison between cells cultured with no uridine/glucose with and without trametinib treatment: ****$P < 0.0001$. The experiment was performed once. n, Vinculin is used as a loading control. The experiments were performed twice with similar results. o, Statistical significance was measured using one-way ANOVA with Tukey's multiple comparisons test. $n = 4$ biologically independent samples per group per cell line. PATU8988S (comparison between cells cultured with and without trametinib in the absence of uridine: ****$P < 0.0001$, and with uridine supplementation: ****$P < 0.0001$); DANG (comparison between cells cultured with and without trametinib in the absence of uridine: $P = 0.9967$, and with uridine supplementation: ****$P = 0.0001$); ASPC1 (comparison between cells cultured with and without trametinib in the absence of uridine: $P = 0.9987$, and with uridine supplementation: ***$P = 0.0001$. The experiment was performed once. Data in b–e,k,m,o are mean ± s.d.

Figure 4. a. The experiment involved a sequential treatment of mice with control IgG followed the next day by liposome PBS (control group) or anti-CSF1 followed the next day by clodronate (clod + Ab (macrophage depletion) group), after the establishment of palpable pancreatic orthotopic tumours with the KPC 7940b cell line. Data represent the average of quantification from three histological slides obtained per tumour. Sample size used for histology, $n = 5$ tumours from the same number of mice per group. Statistical significance was measured using two-tailed unpaired *t*-test with Welch's correction, $P = 0.0635$. The experiment was performed once. b. Sample size: control, 9; clod + Ab, 8 tumours from the corresponding number of mice. Statistical significance was measured using two-tailed unpaired *t*-test with Welch's correction, **$P = 0.0046$. c, Plasma, $n = 8$; TIF, $n = 8$ control and macrophage-depleted ($n = 8$); tumours, $n = 8$ control and $n = 8$ macrophage-depleted. Statistical significance was measured using two-tailed unpaired *t*-test with Welch's correction. Comparison between control and clod + Ab (plasma uridine): ***$P = 0.0003$; control and clod + Ab (TIF uridine): $P = 0.7923$; control and clod + Ab (tumour uridine): $P = 0.1244$. d, The cells were cultured ±1 mM uridine in glucose-free medium supplemented with 2.5% dialysed FBS. Sample size = 4 biologically independent samples per group. Statistical significance was measured using two-tailed unpaired *t*-test. KPC 7940b, comparison of cell culture without and with 0.1 mM uridine: ****$P < 0.0001$, $P = 0.2577$ and $P = 0.1118$ for sgV, sg1 and sg3 groups,

respectively; MT3-2D, comparison of cell culture without and with 0.1 mM uridine: **$P = 0.0026$, $P = 0.9574$ and $P = 0.927$ for sgV, sg1 and sg3 groups, respectively. The experiments were performed twice with similar results. e, Statistical significance was measured using one-way ANOVA with Dunnett's multiple comparisons test, $n = 3$ biologically independent samples per group. Comparison between sgV and sg1 or sg3: ****$P < 0.0001$ (for both intracellular uridine and uracil). Extracellular uridine, comparison between sgV and sg1: $P = 0.1758$; comparison between sgV and sg3: $P = 0.1503$. Extracellular uracil, comparison between sgV and sg1 or sg3: ****$P < 0.0001$. g,h, $n = 3$ biologically independent samples per group. The metabolomics experiment was performed once. i, Tumour weight data are shown in j. j, Number of mice and tumour samples: sgV, $n = 6$; sg1, $n = 10$; sg3, $n = 8$. Statistical significance was measured using one-way ANOVA with Dunnett's multiple comparisons test. Comparison between sgV and sg1 or sg3: ****$P < 0.0001$. Experiment performed once. k, Number of samples: sgV, 8; sg1, 8; sg3, 8 tumours, corresponding to four mice per group. Statistical significance was measured using one-way ANOVA with Dunnett's multiple comparisons test. Comparison between sgV and sg1: **$P = 0.003$; comparison between sgV and sg3: ***$P = 0.0002$. Experiment performed once. l, Samples used for metabolomics per group: sgV, 5; sg1, 6; sg3, 6. Statistical significance was measured using one-way ANOVA with Dunnett's multiple comparisons test. Uridine, comparison between sgV and sg1: **$P = 0.0016$; sgV and sg3: **$P = 0.0019$; uracil, comparison between sgV and sg1: ****$P < 0.0001$; sgV and sg3: ****$P < 0.0001$. Experiment performed once. Data in a–e,g,h,j–l are shown as mean ± s.d. Metabolites used for Venn diagrams (f,m,n) are significantly changed ($P < 0.05$) in the metabolomics profile of UPP1-KO compared with controls per cell line and were derived from LC–MS experiments (Fig. 3d and Extended Data Fig. 5a,c (PATU8988S and ASPC1, intracellular); Fig. 4e and Extended Data Fig. 10c (MT3-2D cell line, in vitro, intracellular); Fig. 4l, Extended Data Fig. 10g (MT3-2D tumours, in vivo) and Extended Data Fig. 10e,f (KPC 7940b tumours, in vivo). Statistical significance was determined using the limma package version 3.38.3 in R. The mouse schematic (a,i) was drawn with Adobe Illustrator 2021 version 25.4.3.

## Reporting summary

Further information on research design is available in the Nature Portfolio Reporting Summary linked to this article.

## Data availability

Human PDA data used in this study are publicly accessible under the accession numbers indicated in the 'PDA dataset analysis' and 'Pan-cancer data analysis' Methods sections, and GSE36133 (for CCLE data). The gene microarrays were obtained from NCBI Gene Expression Omnibus. The ICGC-AU microarray data (release_28) were downloaded from https://dcc.icgc.org/projects/ along with the associated clinical data and had no embargo (March 2020) and the TCGA data were downloaded from CBioPortal (https://www.cbioportal.org/). CCLE protein data were accessed via DepMap portal (https://depmap.org/portal/). Human Protein Atlas data are available from https://www.proteinatlas.org/ENSG00000183696-UPP1/tissue/pancreas (normal) and https://www.proteinatlas.org/ENSG00000183696-UPP1/pathology/pancreatic+cancer#img (PDA). The accompanying source data are provided as supplementary tables. All other data that support the findings of this study are available from the corresponding authors upon request.

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

**Acknowledgements** We thank B. Bochner for providing the OmniLog instrument, helping with the data analysis, countless insightful discussions, and support; S. Chan for support with Biolog data collection and analysis; D. DeNardo and the DeNardo laboratory for support designing the macrophage-depletion studies; N. J. Guppy for processing and immunohistochemistry staining of mouse tumour samples; L. Howell for helping with the automated image analysis of immunohistochemistry staining; and members of the Sadanandam and Lyssiotis laboratories and the entire Pancreatic Disease Initiative at the Rogel Cancer Center, University of Michigan, for their insightful comments and discussions. Z.C.N. was supported by the Michigan Postdoctoral Pioneer Program at the University of Michigan Medical School, NIH/NCI grant K99CA267176 and NIH/NIGMS grant R25GM143298. M.H.W. was supported by the CABRI Undergraduate Research Grant and NIH/NIGMS grant 5T32GM139774-02. P.S by NIH/NCI grants 5T32CA140044-12 and 5T32CA140044-13. M.R. was supported by NIH/HD grant T32-HD007505. C.J.H. was supported by NIH/NCI grant K99/R00CA241357. E.S.C. was supported by the Department of Veteran Affairs Career Development Award IK2BX005875. J.S. was supported by NIH/NCI grants K08CA234222 and R37CA262209. A.M. was supported by American Cancer Society (IRG-16-222-56), the University of Chicago Cancer Center Support Grant (P30 CA14599), the Pancreatic Cancer Action Network (2020 Career Development Award), the Brinson Foundation, the Cancer Research Foundation and the Ludwig Center for Metastasis Research. M.P. was supported by NIH/NCI grants R01CA151588 and R01CA198074. A.S. was supported by two grants from Pancreatic Cancer UK, including Future Leaders Fund (FLF2015_05_ICR) and Research Innovation Fund (RIF2014_06_Sadanandam) and funding from Ian Harty Charitable Trust. C.A.L. was supported by NIH/NCI grants R37CA237421, R01CA248160 and R01CA244931, and M.P.d.M. and C.A.L. were supported by UMCCC Core Grant P30CA046592. The funders had no role in study design, data collection and analysis, or the content and publication of this manuscript.

**Author contributions** Z.C.N., M.H.W., A.S. and C.A.L. designed the study. Z.C.N., M.H.W. and C.A.L. wrote the manuscript. Z.C.N., M.H.W. and C.A.L. designed experiments, and Z.C.N., M.H.W. and P.S. collected data for the bulk of the experimental studies. Z.C.N. prepared manuscript figures. S.K. and Z.T. carried out the initial cell line screen, and P.P. conducted the analysis of the screening data that revealed lead metabolite–enzyme interactions. C.R., M.R., D.S., R.E.M., A.A., J.J.A.-S., K.B., H.-J.L., L.N.D., X.H., H.P., J.U., G.N., L.Z. and C.J.H. carried out experiments. A.A. and E.S.C. contributed to the data analysis. J.S., L.P.S., G.J.P., A.M., M.P.d.M., A.S. and C.A.L. provided resources, funding and conceptual input for experiments and supervised the research. All authors reviewed and approved the final manuscript.

**Competing interests** C.A.L. has received consulting fees from Astellas Pharmaceuticals, Odyssey Therapeutics, and T-Knife Therapeutics, and is an inventor on patents pertaining to KRAS-regulated metabolic pathways, redox control pathways in pancreatic cancer, and targeting the GOT1-pathway as a therapeutic approach (US patent 2015126580-A1, 05/07/2015; US patent 20190136238, 05/09/2019; international patent WO2013177426-A2, 04/23/2015). A.S. received grants from Merck, Pierre Fabre and Bristol Myers Squibb and is an inventor on patents on colorectal cancer classification with differential prognosis and personalized therapeutic responses (PCT/IB2013/060416); prognostic and treatment response prediction in gastric cancer (priority patent CSC/BP7295892); patient classification and prognostic method (GEP-NET) (priority patent EP18425009.0); and molecular predictors of therapeutic response to specific anti-cancer agents (US9506926B2).

**Additional information**
**Correspondence and requests for materials** should be addressed to Anguraj Sadanandam or Costas A. Lyssiotis.

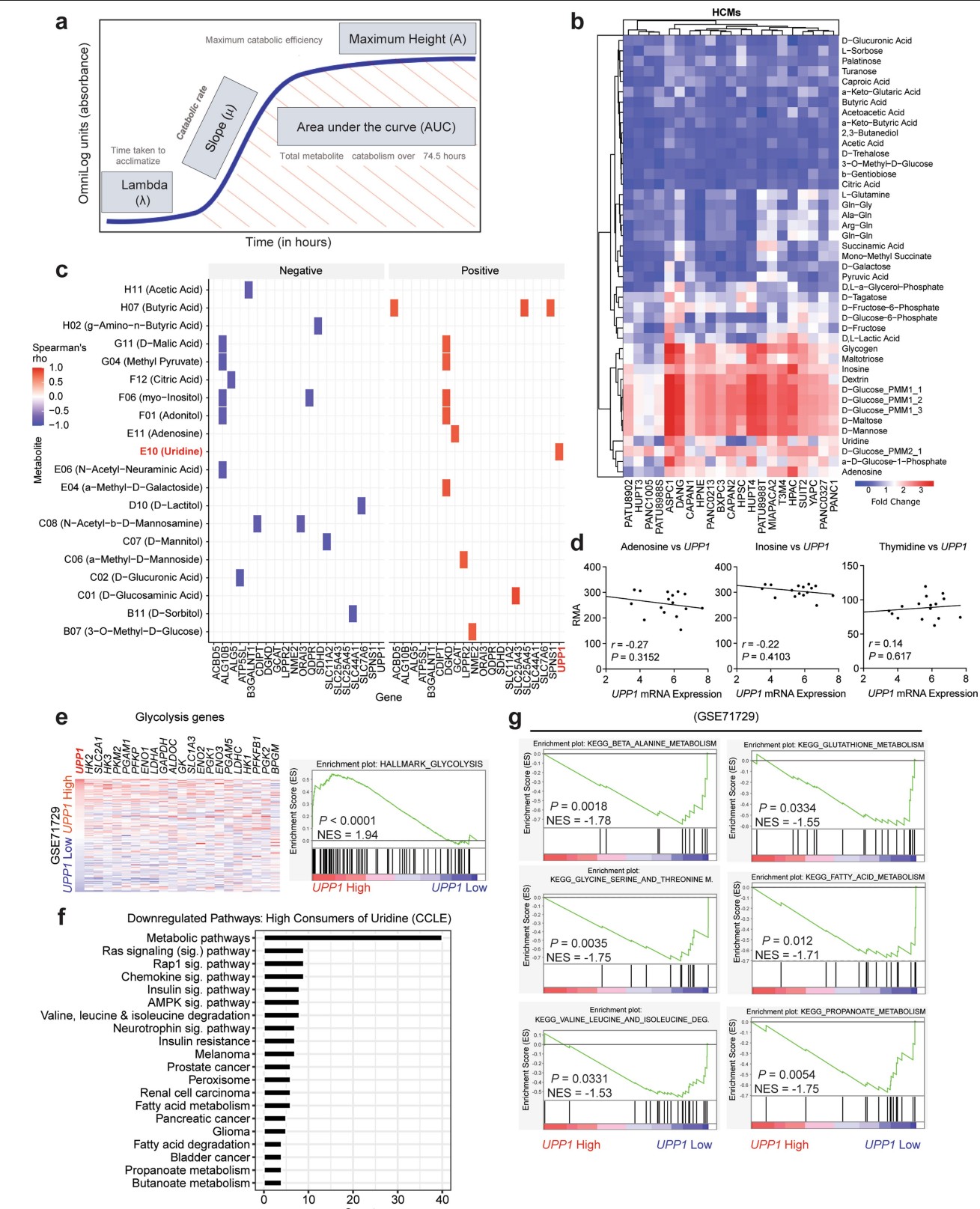

**Extended Data Fig. 1** | See next page for caption.

**Extended Data Fig. 1 | Correlation of nutrient utilization with gene expression identifies uridine and UPP1. a**. Schematic overview of the parameters measured by the Biolog Phenotype Microarray. **b**. Heatmap showing the high confidence metabolites (HCMs), namely, the metabolites that were utilized above or below the median of negative controls as determined by one-tailed Wilcoxon rank sum test. Legend denotes fold change relative to median negative control signal, where red shows high utilization and blue shows low utilization. **c**. Spearman correlation plot, indicating the genes that showed positive or negative correlation with metabolites' RMA in the Biolog screen. **d**. Spearman correlations, r, between *UPP1* expression in cell lines (ref. 17) and RMA of nucleosides that were included in the Biolog screen. n = 16 PDA cell lines. **e**. Heatmap showing the expression of glycolysis genes in human PDA tumours ranked based on *UPP1* expression (dataset: GSE71729, *UPP1* low, n = 72; high, n = 73). On the right: GSEA plot indicating the enrichment of glycolysis hallmark in the *UPP1* high relative to the low tumours. NES, normalized enrichment score.

**f**. Downregulated pathways in PDA cell lines that metabolize uridine at a high level, as revealed by gene ontology (GO) analysis of the differentially expressed genes (P < 0.05). GO analysis was performed with DAVID (https://david.ncifcrf.gov/tools.jsp). Analysis was based on the differential genes derived from CCLE data and part of the data shown in Fig. 1g. **g**. GSEA plots of significantly enriched KEGG pathways in *UPP1*-high PDA tumours relative to *UPP1* low tumours. Plots are part of the data (e) from the analysis of GSE71729 human PDA dataset. **Statistics and reproducibility: a**, The kinetic measurement evaluated several parameters, including the time taken for cells to adapt to and catabolize a nutrient (lambda), the rate of uptake and catabolism (mu or slope), the total metabolic activity (area under the curve; AUC), and the maximum metabolic activity. The values from the maximum catabolic efficiency (maximum height, A) of the respective metabolites were used to determine relative metabolic activity (RMA).

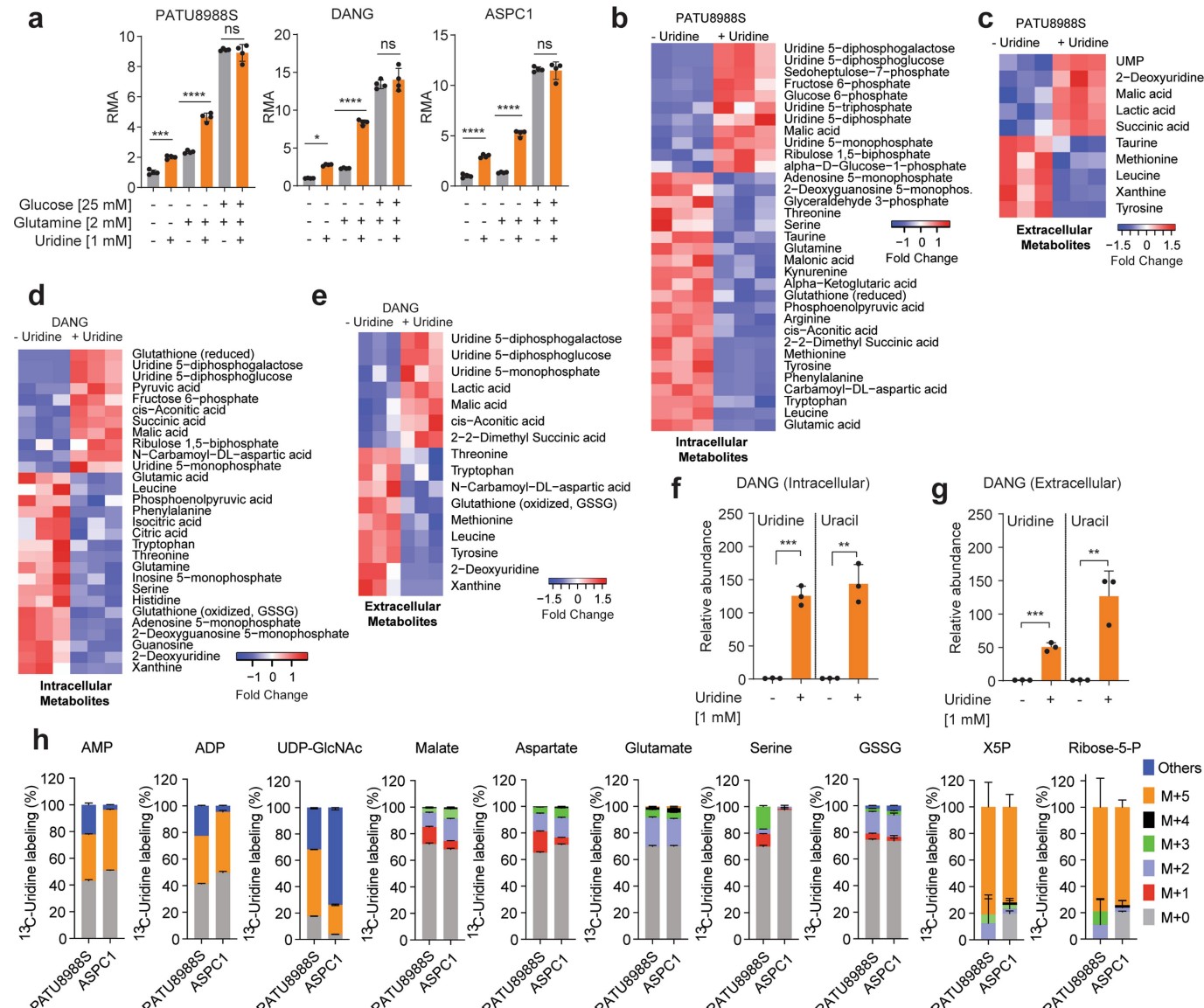

**Extended Data Fig. 2 | Nutrient-deprived PDA use uridine to support metabolism. a.** Relative RMA upon uridine supplementation with or without glucose and glutamine. n = 4 biologically independent samples per group per cell line. **b.** Differential changed (*P* < 0.05) intracellular and **c)** extracellular metabolites from PATU8988S cells supplemented with 1 mM uridine in glucose-free medium for 24 h, as determined by LC-MS metabolomics. n = 3 biologically independent samples per group. **d.** Differentially changed (*P* < 0.05) intracellular and **e)** extracellular metabolites from DANG cells supplemented with 1 mM uridine in glucose-free medium for 24 h, as determined by LC-MS metabolomics. n = 3 biologically independent samples per group. **f.** Intracellular and **g)** extracellular uridine and uracil from DANG cells supplemented with 1 mM uridine in glucose-free medium for 24 h, as determined by LC-MS. Plots in f-g are from the same experiment as d-e. n = 3 biologically independent samples per group. Statistical significance was measured using two-tailed unpaired t-test. Intracellular – comparison between no uridine and 1 mM uridine: *** P = 0.0001 for uridine, ** P = 0.0011 for uracil. Extracellular – comparison between no uridine and 1 mM uridine: *** P = 0.0002 for uridine, ** P = 0.0044 for uracil. **h.** Mass isotopologue distribution of $^{13}C_5$-uridine ribose-derived carbon in the displayed metabolites after 24 h culture in a glucose-free medium supplemented with 1 mM uridine. n = 3 biologically independent samples per cell line. Tracing experiments were performed twice in these cells with similar results. Data (a, f, g, h) are shown as mean ± s.d. ADP, adenosine diphosphate; AMP, adenosine monophosphate; GSSG, oxidized glutathione; NADH, nicotinamide adenine dinucleotide; UDP-GlcNAc, uridine diphosphate N-acetylglucosamine; X5P, xylulose 5-phosphate. **Statistics and Reproducibility: a**, n = 4 biologically independent samples per group per cell line. Statistical significance was measured using one-way ANOVA with Tukey's multiple comparisons test. PATU8988S (comparison between cells cultured in (−) glucose/glutamine/uridine and (−) glucose/glutamine/+uridine: *** P = 0.0007, comparison between (−) and (+) uridine in the presence of glutamine and without glucose: **** P < 0.00001, comparison between (−) and (+) uridine in the presence of glutamine and glucose: P = ns (0.8856)). DANG (comparison between cells cultured in (−) glucose/glutamine/uridine and (−) glucose/glutamine/+uridine: * P = 0.0165, comparison between (−) and (+) uridine in the presence of glutamine and without glucose: **** P < 0.0001, comparison between (−) and (+) uridine in the presence of glutamine and glucose: P = ns (0.7971)). ASPC1 (comparison between cells cultured in (−) glucose/glutamine/uridine and (−) glucose/glutamine/+uridine: **** P < 0.0001, comparison between (−) and (+) uridine in the presence of glutamine and without glucose: **** P < 0.00001, comparison between (−) and (+) uridine in the presence of glutamine and glucose: P = ns (0.9968)).

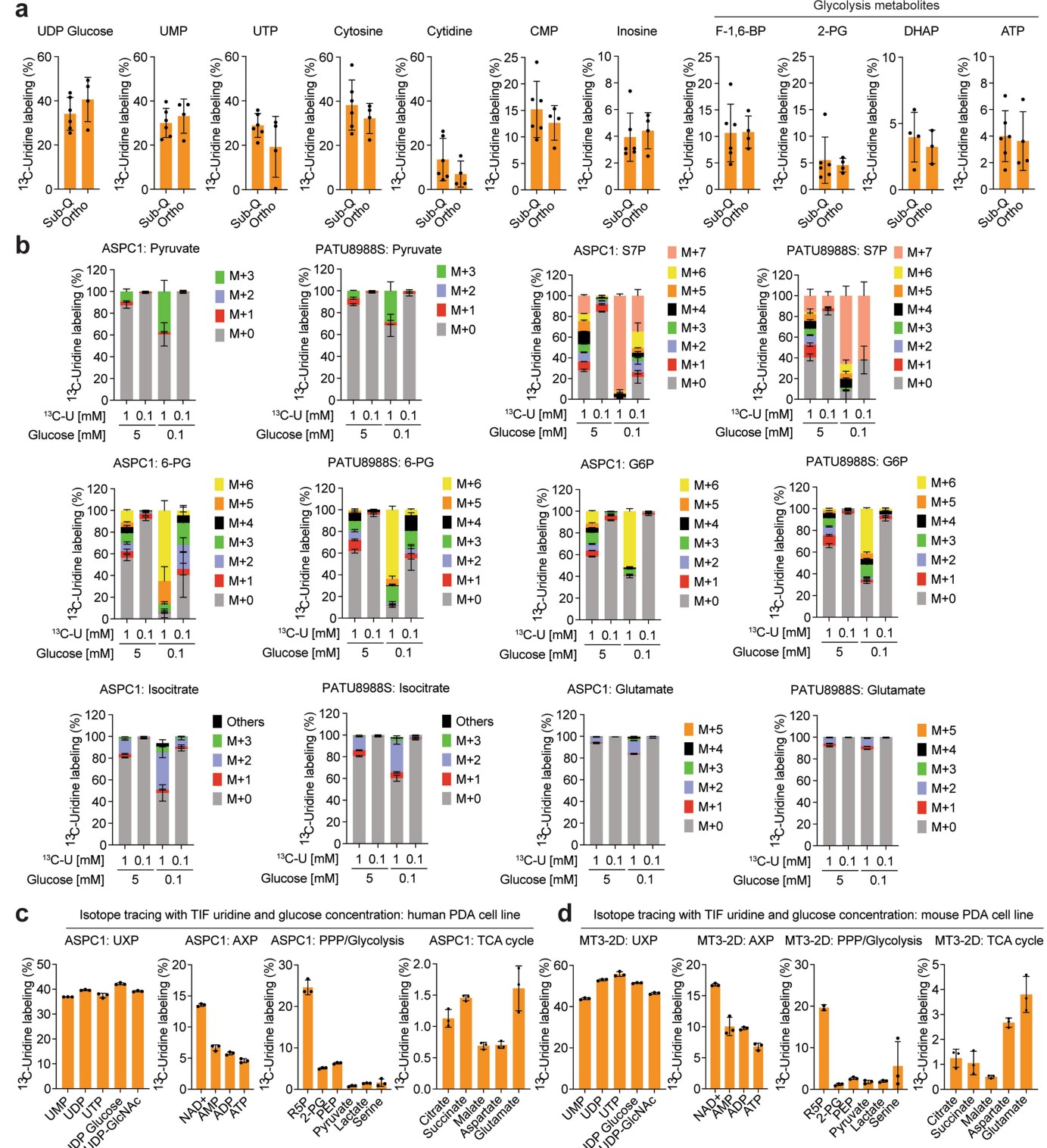

**Extended Data Fig. 3** | See next page for caption.

**Extended Data Fig. 3 | PDA metabolize uridine via central carbon metabolism** *in vitro* **and** *in vivo*. **a**. Isotope tracing showing $^{13}C_5$-uridine ribose-derived carbon labelling in subcutaneous (Sub-Q) or orthotopically (Ortho) implanted KPC 7940b tumours collected 1 h after injecting the mice with 200 µL or 50 µL (Sub-Q) 0.2 M $^{13}C_5$-uridine. Number of samples: Sub-Q = 6 tumours from 3 mice injected on the left and right flanks; Ortho = 4 tumours from 4 mice. Mode of uridine injection is intratumoural for Sub-Q and intraperitoneal for Ortho. **b**. Mass isotopologue distribution of $^{13}C_5$-uridine ribose-derived carbon after 24 h culture of ASPC1 and PATU8988S cells in medium supplemented with 1 mM or 0.1 mM uridine, each with 5 mM or 0.1 mM glucose concentration. n = 4 biologically independent samples per group. M – mass; 'Others' – indicate M other than M+0 or M+5, where applicable. **c**–**d**. Isotope tracing showing metabolite labelling upon supplementation with $^{13}C_5$-uridine at the TIF uridine and glucose concentrations shown in Fig. 2h,i, after 12 h of culturing **c**) human PDA cell line ASPC1 and **d**) murine PDA cell line MT3-2D. The cell lines were cultured in medium supplemented with 25 µM $^{13}C_5$-uridine and 0.65 mM glucose. n = 3 biologically independent samples per cell line. AXP – AMP, ADP, ATP, and related metabolites; UXP – UMP, UDP, UTP and related metabolites. The experiments (a-d) were performed once. Data (a-d) are shown as mean ± s.d, where applicable. 2-PG, 2-phosphoglycerate; 6-PG, 6-phosphogluconate; ADP, adenosine diphosphate; AMP, adenosine monophosphate; ATP, adenosine triphosphate; DHAP, dihydroxyacetone phosphate; F1,6-BP, fructose 1,6-bisphosphate; CMP, cytidine monophosphate; G6P, glucose 6-phosphate; NAD+, nicotinamide adenine dinucleotide; R5P, ribose 5-phosphate; PEP, phosphoenolpyruvate; PPP, pentose phosphate pathway; TCA, tricarboxylic acid; S7P, sedoheptulose 7-phosphate; UDP, uridine diphosphate; UMP, uridine monophosphate; UTP, uridine triphosphate; UDP-GlcNAc, uridine diphosphate N-acetylglucosamine.

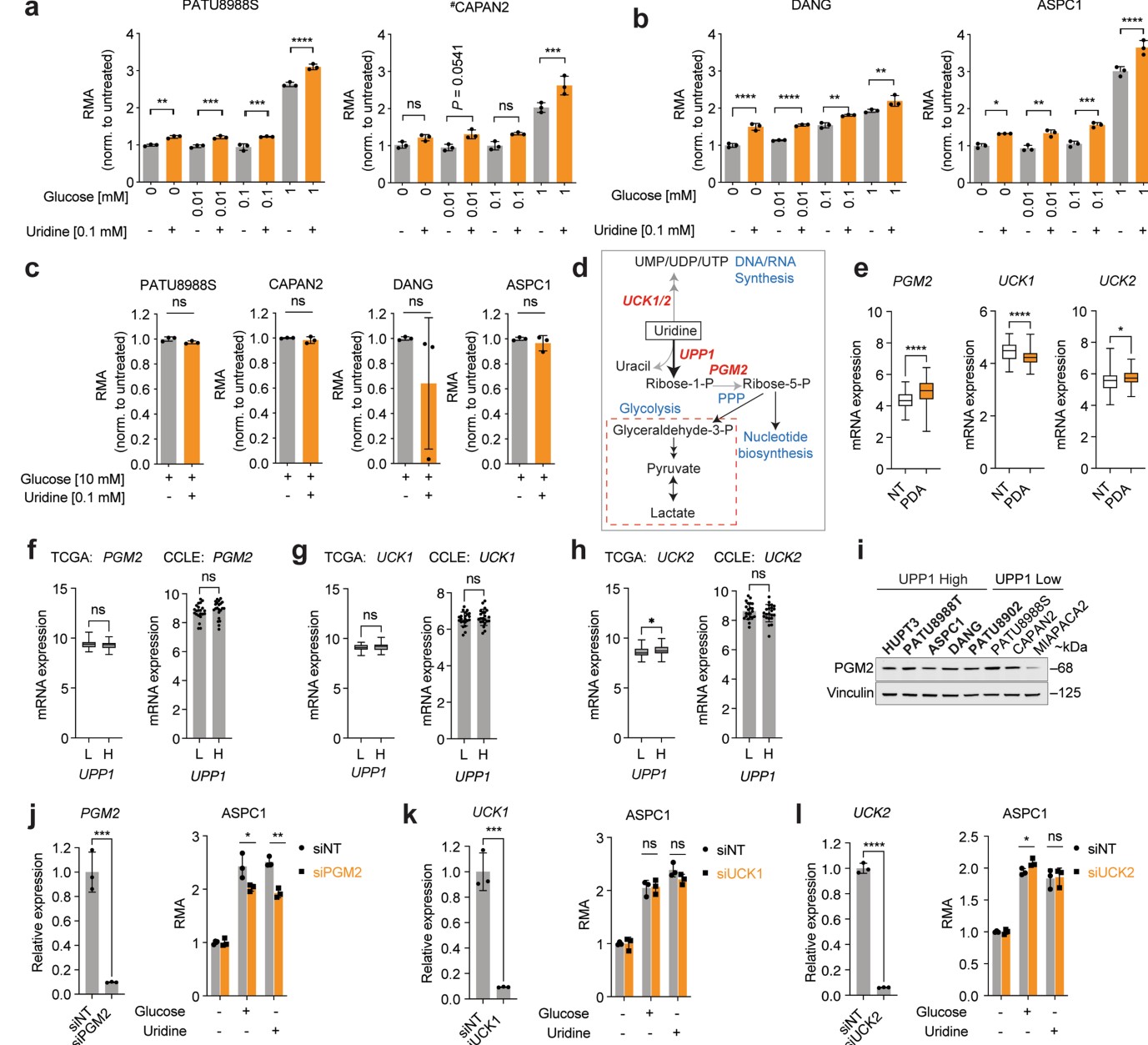

**Extended Data Fig. 4** | See next page for caption.

**Extended Data Fig. 4 | Cellular pathways for ribose salvage from uridine.**
**a–c.** Relative metabolic activity (RMA) of PDA cell lines depicting the preferential use of uridine at (**a,b**) low glucose concentration (0-1 mM) but not at **c**) the high glucose concentration (10 mM), over a 96 h culture. **d.** Schematic depicting metabolic pathways for uridine utilization. **e.** Expression of *PGM2*, *UCK1*, and *UCK2* in non-tumour (NT) and PDA tissue samples from the GSE71729 dataset. Number of samples: NT = 46, PDA = 145. **f–h.** Expression of *PGM2*, *UCK1* and *UCK2* in TCGA (human PDA tumour) and CCLE (human cell line) data separated into *UPP1* low (L) and high (H) subsets. **i.** Western blot for PGM2 in PDA cell lines. Presented in bold are cells that express high UPP1. These samples are the same batch as the data shown in Fig. 1e and the blot was generated during one of the technical replicates of that western blotting. kDa, unit for molecular weight. **j.** qPCR for *PGM2* in ASPC1 cells transfected with siPGM2 compared to non-targeting (siNT) control. On the right: RMA in *PGM2* knockdown cells +/− uridine (1 mM) or glucose (1 mM). **k.** qPCR for *UCK1* in ASPC1 cells transfected with siUCK1 compared to non-targeting (siNT) control. On the right: RMA in *UCK1* knockdown cells +/− uridine (1 mM) or glucose (1 mM). **l.** qPCR for *UCK2* in ASPC1 cells transfected with siUCK2 compared to non-targeting (siNT) control. On the right: RMA in *UCK2* knockdown cells +/− uridine (1 mM) or glucose (1 mM). **Statistics and reproducibility: a–c,** n = 3 biologically independent samples. Statistical significance for data in a-b was measured using one-way ANOVA with Tukey's multiple comparisons test. PATU8988S (comparison between no glucose and no glucose + uridine [0.1 mM]: **$P = 0.0017$; 0.01 mM glucose and 0.01 mM glucose + uridine: ***$P = 0.0008$; 0.1 mM glucose and 0.1 mM glucose + uridine: ***$P = 0.0002$; 1 mM glucose and 1 mM glucose + uridine: ****$P < 0.0001$). CAPAN2 (comparison between no glucose and no glucose + uridine: $P = ns$ (0.5673); 0.01 mM glucose and 0.01 mM glucose + uridine: $P = ns$ (0.0541); 0.1 mM glucose and 0.1 mM glucose + uridine: $P = ns$ (0.092); 1 mM glucose and 1 mM glucose + uridine: ***$P = 0.0007$. #All four group comparisons have significant P: 0.0468, 0.014, 0.0089, 0.0222, respectively, when directly compared using two-tailed unpaired t test). DANG (comparison between no glucose and no glucose + uridine: ****$P < 0.0001$; 0.01 mM glucose and 0.01 mM glucose + uridine: ****$P < 0.0001$; 0.1 mM glucose and 0.1 mM glucose + uridine: **$P = 0.0051$; 1 mM glucose and 1 mM glucose + uridine: **$P = 0.0051$). ASPC1 (comparison between no glucose and no glucose + uridine: *$P = 0.0203$; 0.01 mM glucose and 0.01 mM glucose + uridine: **$P = 0.0031$; 0.1 mM glucose and 0.1 mM glucose + uridine: ***$P = 0.0003$; 1 mM glucose and 1 mM glucose + uridine: ****$P < 0.0001$). Statistical significance for data in c was measured using two-tailed unpaired t test and $P = ns$ (0.0852, 0.3509, 0.3021 and 0.3875 for PATU8988S, CAPAN2, DANG and ASPC1, respectively, in the comparison of no uridine vs 0.1 mM uridine groups in the presence of 10 mM glucose). **d.** Uridine can be channeled

into DNA or RNA synthesis by direct phosphorylation from UCK1/2. Uridine can also be catabolized via UPP1 to produce uracil and ribose 1-phosphate. Ribose 1-phosphate is converted to ribose-5-phosphate by PGM2 and fuel pentose phosphate pathway, nucleotide biosynthesis and glycolysis. **e.** Statistical significance was measured using two-tailed unpaired t test with Welch's correction. Comparison between NT and PDA: *PGM2*, ****$P < 0.0001$; *UCK1*, ****$P < 0.0001$; *UCK2*, *$P = 0.018$. Box plot statistics – *PGM2* (NT: minima = 3.097, maxima = 5.527, median = 4.335, 25th percentile = 3.992, 75th percentile = 4.74; PDA: minima = 2.386, maxima = 6.433, 25th percentile = 4.424, median = 4.961, 75th percentile = 5.457), *UCK1* (NT: minima = 3.7, maxima = 5.1, median = 4.5, 25th percentile = 4.2, 75th percentile = 4.7; PDA: minima = 3.6, maxima = 5.1, median = 4.2, 25th percentile = 4, 75th percentile = 4.4), *UCK2* (NT: minima = 4.034, maxima = 7.615, median = 5.577, 25th percentile = 5.095, 75th percentile = 5.9; PDA: minima = 4.556, maxima = 6.93, median = 5.727, 25th percentile = 5.458, 75th percentile = 6.059). **f–h.** Number of samples: TCGA – *UPP1* low = 75, high = 75; CCLE – *UPP1* low = 22, high = 22. Statistical significance was measured using two-tailed unpaired t test with Welch's correction. Comparison between L and H groups in TCGA (*PGM2*: $P = ns$ (0.1226), *UCK1*: $P = ns$ (0.311); *UCK2*: *$P = 0.0327$). In the CCLE L versus H comparison, *PGM2*, *UCK1* and *UCK2* have $P = ns$ (0.3486, 0.4645, 0.4381, respectively). TCGA – The Cancer Genome Atlas, CCLE – Cancer Cell Line Encyclopaedia. **i.** Vinculin is used as a loading control. **j.** Number of samples: qPCR = 3, RMA = 3 biologically independent samples per group. qPCR – statistical significance was measured using unpaired t test; comparison between siNT and siPGM2: ***$P = 0.0007$. RMA – statistical significance was measured using multiple unpaired t tests with two-stage two-step method; comparison of siNT and siPGM2 in the presence of 1 mM glucose and no uridine: *$P = 0.0452$, and **$P = 0.0014$ in the presence of 1 mM uridine and no glucose. **k.** Number of samples: qPCR = 3, RMA = 3 biologically independent samples per group. qPCR – statistical significance was measured using two-tailed unpaired t test; comparison between siNT and siUCK1: ***$P = 0.0004$. RMA – statistical significance was measured using multiple unpaired t tests with two-stage two-step method. Comparison of siNT and siUCK1 knockdown samples in the presence of 1 mM glucose and no uridine: $P = ns$ (0.8652), and $P = ns$ (0.131) in the presence of 1 mM uridine and no glucose. **l.** Number of samples: qPCR = 3, RMA = 3 biologically independent samples per group. qPCR – statistical significance was measured using unpaired t test; comparison between siNT and siUCK2: ****$P < 0.0001$. RMA – statistical significance was measured using multiple unpaired t tests with two-stage two-step method; comparison of siNT and siUCK1 knockdown cells in the presence of 1 mM glucose and no uridine: *$P = 0.035$, and $P = ns$ (0.8653) in the presence of 1 mM uridine and no glucose. Data (a–c, f–h, j–l) are shown as mean ± s.d. The experiments were performed once (a–c, k), and twice (j,l) with similar results.

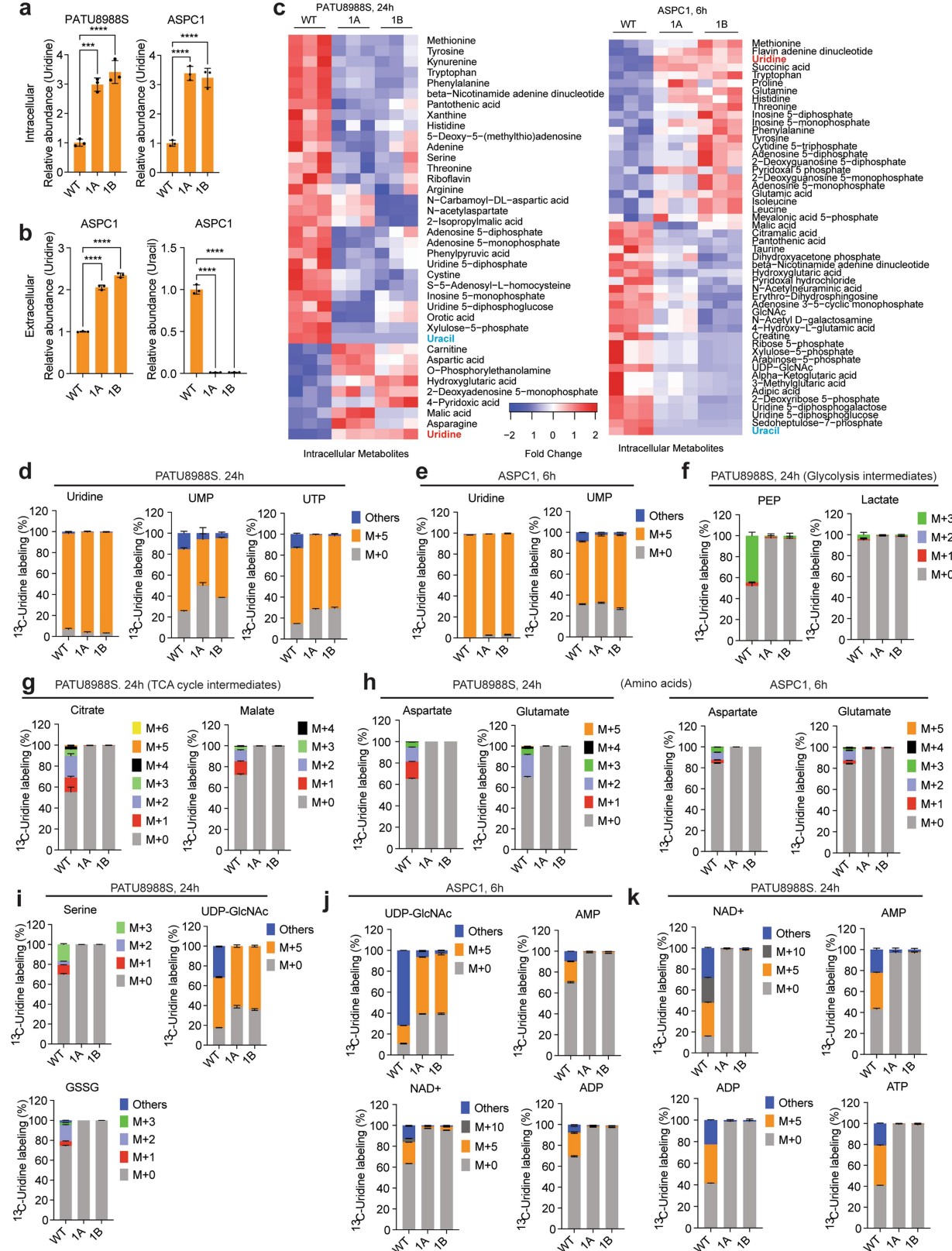

**Extended Data Fig. 5 |** See next page for caption.

**Extended Data Fig. 5 | UPP1 mediates the liberation of uridine-derived ribose for central carbon metabolism. a**. Relative intracellular uridine level in the UPP1 knockout PATU8988S cells (after 24 h) and ASPC1 cells (6 h) compared to the wild type supplemented with 1 mM uridine in medium with 10% dialyzed FBS and no glucose. n = 3 biologically independent samples per group per cell line. Statistical significance was measured using one-way ANOVA with Dunnett's multiple comparisons test. PATU8988S (intracellular uridine) – comparison between WT and 1A, ***P = 0.0002; comparison between WT and 1B, ****P < 0.0001. ASPC1 (intracellular uridine) – comparison between WT and 1A or 1B, ****P < 0.0001. **b**. Relative extracellular uridine and uracil in UPP1 knockout ASPC1 cells compared to the wild type supplemented with 1 mM uridine for 6 h in medium with 10% dialyzed FBS and no glucose. n = 3 biologically independent samples per group. Statistical significance was measured using one-way ANOVA with Dunnett's multiple comparisons test. Extracellular uridine: comparison between WT and 1A or 1B, ****P < 0.0001; extracellular uracil: comparison between WT and 1A or 1B, ****P < 0.0001. **c**. Heatmaps of significantly altered intracellular metabolites (P < 0.05) in PATU8988S cells after 24 h (left) and ASPC1 after 6 h (right), as measured by LC-MS. Data used for the intracellular uridine/uracil plots (a-b) were extracted from this profiling study. **d–k**. Mass isotopologue distribution of 1 mM $^{13}C_5$-uridine ribose-derived carbon into the indicated metabolic pathways in wildtype (WT) or UPP1-KO PATU8988S and ASPC1 cells. M – mass; 'Others' – indicate M other than M+0 or M+5, where applicable. 1A and 1B denote UPP1-KO sgRNA clones. Data (a–b, d–k) are shown as mean ± s.d. Metabolomics experiments were done once. **Statistics and reproducibility:** Abbreviations – ADP, adenosine diphosphate; ATP, adenosine triphosphate; AMP, adenosine monophosphate; GSSG, oxidized glutathione; NAD+ nicotinamide adenine dinucleotide; PEP, phosphoenolpyruvate; TCA cycle, tricarboxylic acid cycle; UDP-GlcNAc, uridine diphosphate N-acetylglucosamine; UMP, uridine monophosphate; UTP, uridine triphosphate.

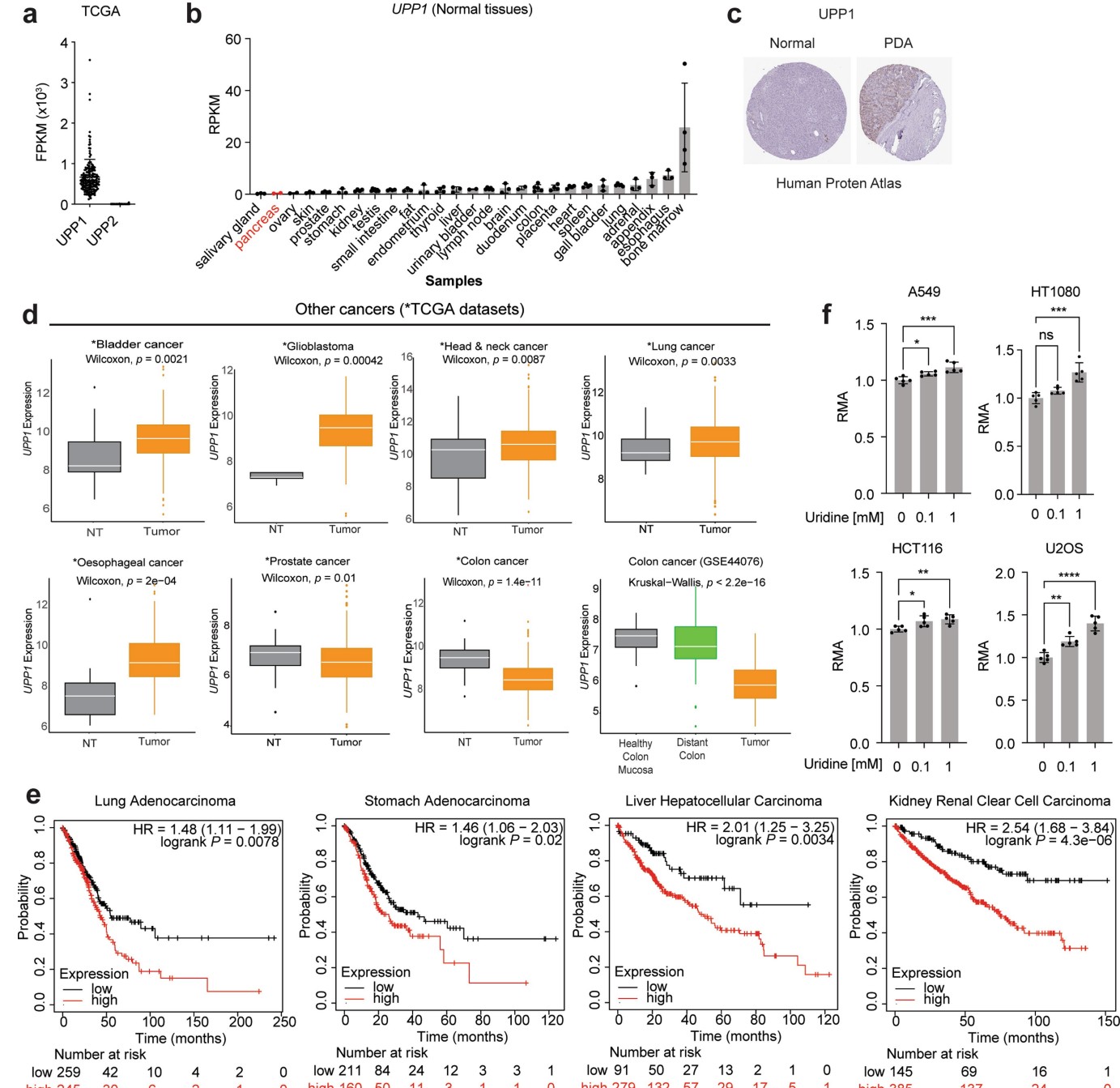

**Extended Data Fig. 6** | See next page for caption.

**Extended Data Fig. 6 | UPP1 expression is elevated in PDA and other cancer types. a**. TCGA RNA seq data showing the expression of *UPP1* and its paralog *UPP2*. FPKM, fragments per kilobase of exon per million mapped fragments. **b**. RNA seq showing UPP1 expression in various normal human tissues (Human Protein Atlas data), as obtained from the National Center for Biotechnology Information (NCBI) portal. **c**. Histological data showing UPP1 protein expression in normal pancreatic tissue compared to PDA. **d**. *UPP1* expression in human non-PDA cancers accessed in publicly accessible datasets. **e**. Kaplan-Meier plot of survival probability (log-rank test) as obtained from KM-plotter (https://kmplot.com/analysis/) using the default parameters. **f**. Relative Metabolic Activity (RMA), reflecting NADH levels, in human non-PDA cancer cell lines supplemented with uridine (as indicated) in 1 mM glucose medium. n = 5 biologically independent samples per group per cell line. Statistical significance was measured using one-way ANOVA with Dunnett's multiple comparisons test. A549 (lung cancer cell line, comparison between no uridine and 0.1 mM uridine, *P = 0.0405 or 1 mM uridine, ***P = 0.0004); HT1080 (fibrosarcoma cell line, comparison between no uridine and 0.1 mM uridine, P = ns (0.1773) or 1 mM uridine, ***P = 0.0001); HCT116 (colon cancer cell line, comparison between no uridine and 0.1 mM uridine, *P = 0.0294 or 1 mM uridine, **P = 0.0081); U2OS (osteosarcoma cell line, comparison between no uridine and 0.1 mM uridine, **P = 0.002 or 1 mM uridine, ****P < 0.0001). **Statistics and reproducibility: a**, n = 150 samples each. **b**. Sample size, n: salivary gland = 3, pancreas = 2, ovary = 2, skin = 3, prostate = 4, stomach = 3, kidney = 4, testis = 7, small intestine = 4, fat = 3, endometrium = 3, thyroid = 4, liver = 3, urinary bladder = 2, lymph node = 5, brain = 3, duodenum = 2, colon = 5, placenta = 4, heart = 4, spleen = 4, gall bladder = 3, lung = 5, adrenal = 3, appendix = 3, esophagus = 3, bone marrow = 4. RPKM, reads per kilobase of exon per million reads mapped. *UPP1* expression in normal pancreas is extremely low (second lowest of the >25 tissues compared). **c**. Data obtained from the Human Protein Atlas (URL for 'Normal' - https://www.proteinatlas.org/ENSG00000183696-UPP1/tissue/pancreas; PDA – https://www.proteinatlas.org/ENSG00000183696-UPP1/pathology/pancreatic+cancer#img). **d**. Sample size, n: NT = 19, tumour = 408 (bladder cancer, TCGA); NT = 5, tumour = 154 (glioblastoma, TCGA); NT = 44, tumour = 520 (head and neck cancer, TCGA); NT = 59, tumour = 551 (lung cancer, TCGA); NT = 11, tumour = 184 (oesophageal cancer, TCGA); NT = 52, tumour = 497 (prostate cancer, TCGA); NT = 41, tumour = 452 (colon cancer); health colon mucosa = 50, distant colon = 98, tumour = 98 (colon cancer, GSE44076). NT – non-tumour/adjacent normal tissue. Data (a-b, f) shown as mean ± s.d. The experiments were performed three times with similar results. Box plot statistics – TCGA, bladder carcinoma (primary: minima = 5.83, maxima = 13.5, median = 9.77, 25th percentile = 9.015, 75th percentile = 10.47; normal: minima = 6.61, maxima = 12.43, median = 8.35, 26th percentile = 8.03, 75th percentile = 9.59); glioblastoma multiforme (primary: minima = 5.71, maxima = 11.84, median = 9.585, 25th percentile = 8.79, 75th percentile = 10.143; normal: minima = 7.04, maxima = 7.63, median = 7.4, 25th percentile = 7.36, 75th percentile = 7.61); head and neck squamous cell carcinoma (primary: minima = 6.59, maxima = 15.64, median = 10.75, 25th percentile = 9.787, 75th percentile = 11.565; normal: minima = 6.38, maxima = 13.73, median = 10.42, 25th percentile = 8.672, 75th percentile = 11.065); lung adenocarcinoma (primary: minima = 6.45, maxima = 13.44, median = 9.8, 25th percentile = 9.13, 75th percentile = 10.49; normal: minima = 8.3, maxima = 11.39, median = 9.3, 25th percentile = 8.945, 75th percentile = 9.93); esophageal carcinoma (primary: minima = 6.7, maxima = 13.08, median = 9.26, 25th percentile = 8.578, 75th percentile = 10.21; normal: minima = 6.17, maxima = 12.39, median = 7.62, 25th percentile = 6.7, 75th percentile = 8.26); prostate adenocarcinoma (primary: minima = 3.96, maxima = 9.69, median = 6.58, 25th percentile = 5.98, 75th percentile = 7.14; normal: minima = 4.56, maxima = 8.62, median = 6.97, 25th percentile = 6.447, 75th percentile = 7.24); colon cancer (primary: minima = 6.41, maxima = 12.96, median = 8.535, 25th percentile = 8.068, 75th percentile = 9.07; normal: minima = 7.76, maxima = 11.29, median = 9.57, 25th percentile = 9.09, 75th percentile = 9.92). Colon cancer (GSE44076, primary: minima = 4.564, maxima = 7.608, median = 5.917, 25th percentile = 5.487, 75th percentile = 6.405; normal: minima = 4.568, maxima = 9.154, median = 7.18, 25th percentile = 6.781, 75th percentile = 7.824; healthy colon mucosal cells: minima = 5.884, maxima = 8.279, median = 7.529, 25th percentile = 7.153, 75th percentile = 7.74). Statistical significance was tested using two-sided Wilcoxon or Kruskal-Wallis tests.

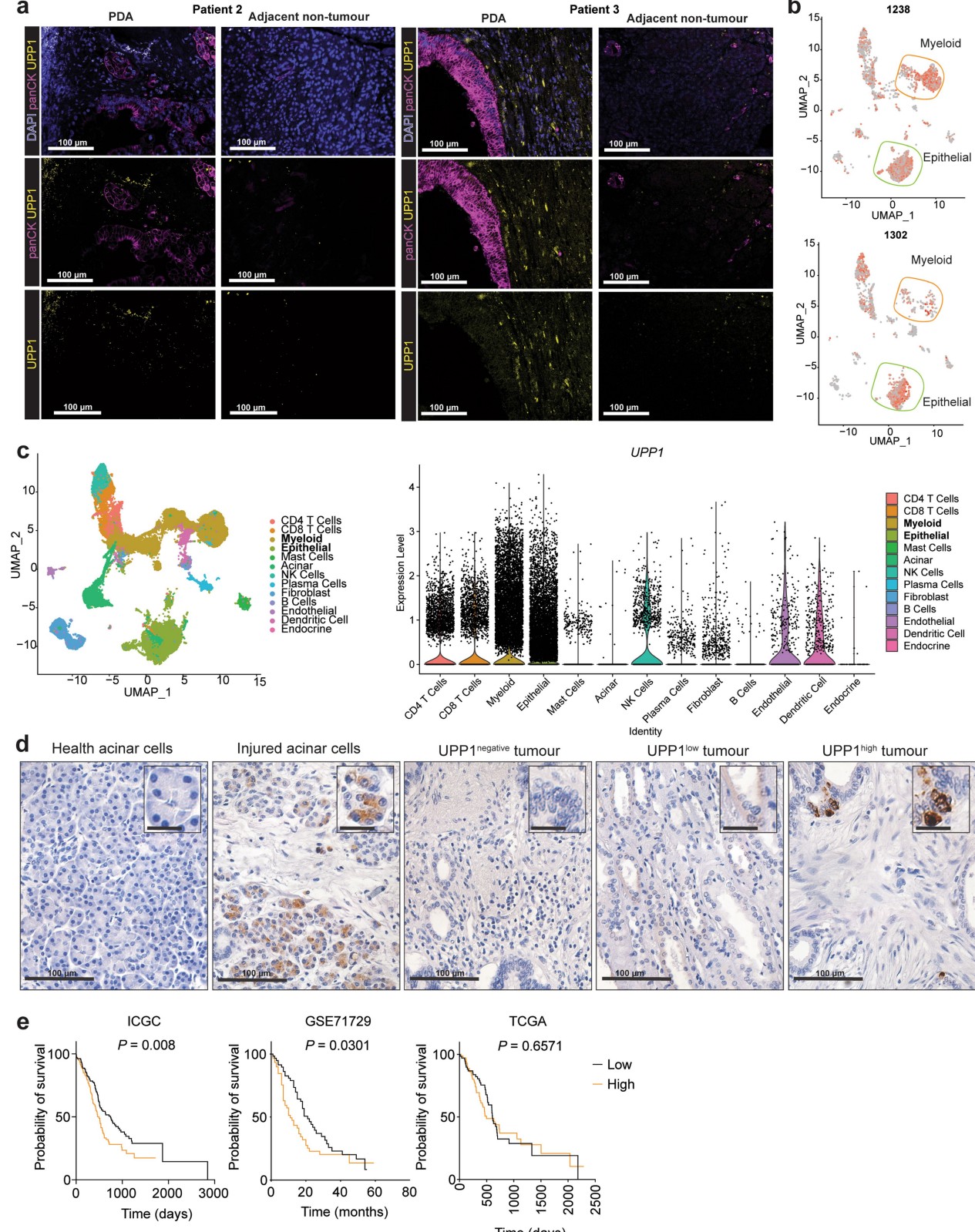

**Extended Data Fig. 7** | See next page for caption.

**Extended Data Fig. 7 | UPP1 is expressed in PDA and TME cells and predicts survival outcome. a**. RNAscope images showing UPP1 expression in tumour (PDA) compared to the adjacent non-tumour tissues. Pan-cytokeratin (PanCK) indicates the tumour cells; DAPI, nuclear stain. The images are representative of three 20x acquisitions per tissue slide, and of two independent experiments. Scale bar indicates 100 µm. **b**. UMAP plot showing the expression of *UPP1* at the transcript level, as determined by single cell RNA seq of PDA tissues from two patients (#1238 and 1302). **c**. Violin plots showing *UPP1* expression in various tumour microenvironment cell types, including myeloid and epithelial cells where *UPP1* is highest. Right, UMAP plot showing the specific cell compartments expressing *UPP1* for all patients' samples combined (n = 16). Data used in plots b-c are from a previously published dataset[55]. **d**. Immunohistochemistry of UPP1 in patient biopsy sections from previously published tissue microarray[48]. Micrographs are representative from 213 patient samples in the microarray and two independent experiments. Large scale bar indicates 100 µm; scale bar on insets indicates 25 µm. **e**. Kaplan-Meier plot showing survival probability (log-rank test) based on *UPP1* expression in three separate datasets. Each dataset was split into two – *UPP1* high and *UPP1* low – based on the ranked *UPP1* expression value. Sample size: low = 133, high = 134 (ICGC); low = 62, high = 63 (GSE71729), low = 73, high = 73 (TCGA). TME – tumour microenvironment.

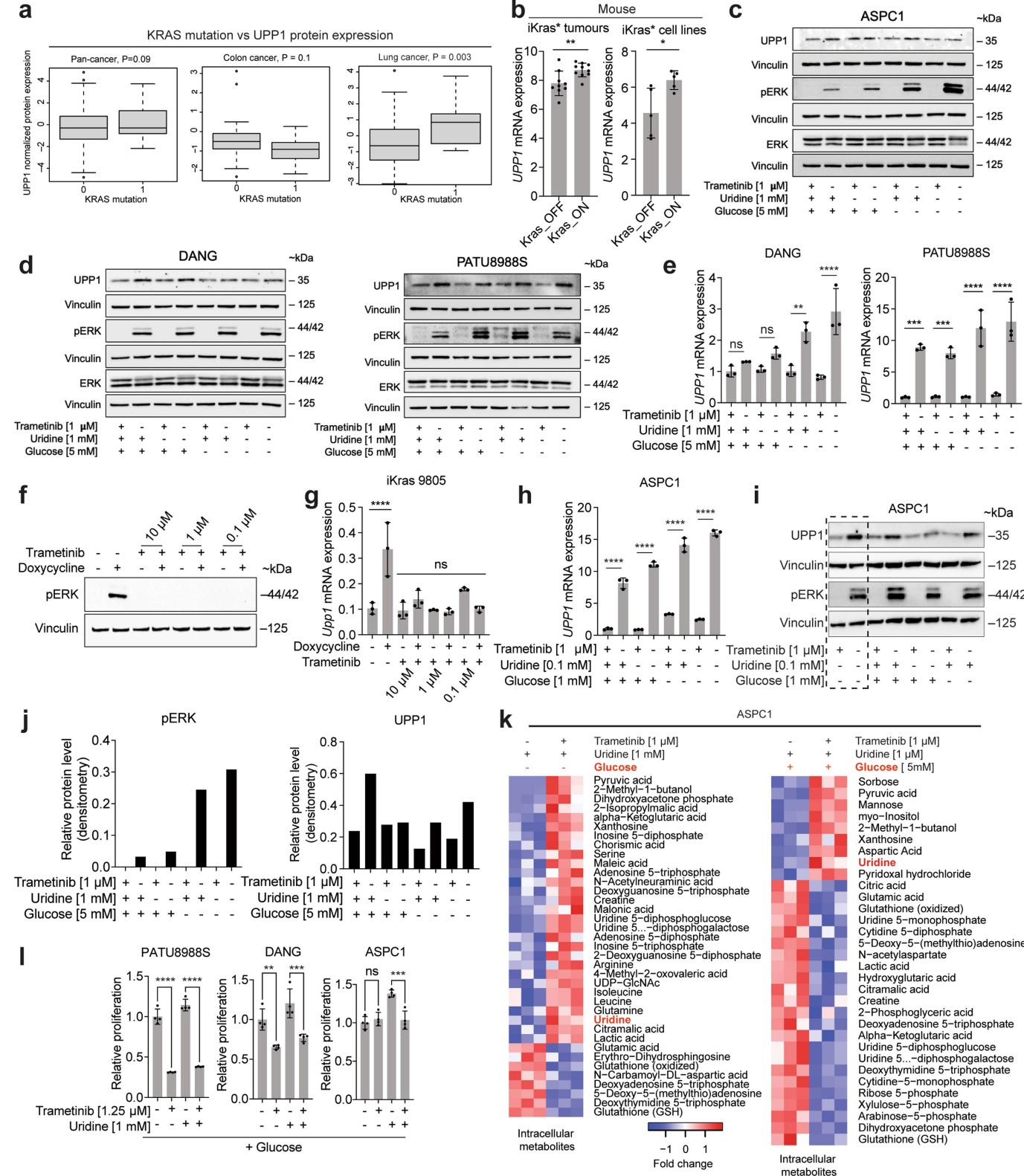

**Extended Data Fig. 8** | See next page for caption.

**Extended Data Fig. 8 | KRAS-MAPK pathway activation and nutrient availability drive UPP1 expression. a**. Normalized UPP1 protein expression in Kras wildtype and mutant cell lines based on CCLE protein data accessed via the DepMap portal. **b**. *Upp1* mRNA expression in iKras* orthotopic tumours and cell lines from dataset GSE32277. **c-d**. Western blot showing UPP1 expression in human PDA cell lines **c**) ASPC1 cells and **d**) DANG and PATU8988S after 24 h culture +/- trametinib [MEK inhibitor], uridine, or glucose. kDa, unit for molecular weight. **e**. qPCR for *UPP1* in human PDA cell lines DANG and PATU8988S treated for 24 h with trametinib (1 μM), uridine (1 mM), or 5 mM glucose. **f**. Western blot showing pERK upon treatment of murine iKras 9805 cells for 24 h with trametinib or doxycycline (1 μg/mL). **g**. qPCR for *Upp1* in iKras* 9805 mouse PDA cells, 24 h after treatment with trametinib or doxycycline (1 μg/mL). **h**. qPCR for *UPP1* in ASPC1 cells treated for 48 h with trametinib and at 0.1 mM uridine or 1 mM glucose concentrations. **i**. Western blot for UPP1 and pERK treated for 48 h with trametinib and low glucose [1 mM] and near physiological uridine concentration [0.1 mM]. **j**. Densitometric quantification of pERK and UPP1 in the ASPC1 blots shown in Fig. 3n. **k**. Metabolomics profiling showing the spectrum of metabolic changes induced in ASPC1 upon pERK inhibition with trametinib [1 μM], 24 h after culture. **l**. MTT assay showing relative proliferation of PDA cell lines treated with 1.25 μM trametinib [MEK inhibitor] +/- 1 mM uridine in the presence of glucose [5 mM] at 96 h. **Statistics and reproducibility: a**, Sample size – wild type 0 and mutation 1: 304 and 69 (pan-cancer), 15 and 15 (colon cancer), 54 and 25 (lung cancer). Box plot statistics – pan-cancer (KRAS = 0: minima = −4.8237, maxima = 4.8004, median = −0.3029, 25th percentile = −1.3212, 75th percentile = 0.7687; minima = −2.1805, maxima = 3.7445, median = −0.3019, 25th percentile = −0.815, 75th percentile = 1.2867), colon (KRAS = 0: minima = −2.3253, maxima = 3.1201, median = -0.5127, 25th percentile = −0.8908, 75th percentile = −0.0958; KRAS = 1: minima = −2.1805, maxima = 0.2672, median = −0.9153, 25th percentile = −1.3926, 75th percentile = −0.5595), lung (KRAS = 0: minima = −2.7979, maxima = 4.0884, median = 0.4201, 25th percentile = -1.305, 75th percentile = 0.6032; KRAS = 1: minima = −0.9357, maxima = 3.7445, median = 0.8446, 25th percentile = −0.4923, 75th percentile = 1.3752). **b**. Sample size: tumours (Kras_OFF = 9, Kras_ON = 10), cell lines (Kras_OFF = 5, Kras_ON = 5). Statistical significance was measured using two-tailed unpaired t test. Comparison of Kras_OFF to Kras_ON: **P = 0.0088 (tumours) and *P = 0.0244 (cell lines). **c-d**. pERK is used as a readout for MAPK

pathway induction/activity. ERK and Vinculin are used as loading controls. Blots are representative of two biological and technical replicates for ASPC1 and one biological replicate for PATU8988S and DANG with similar results. **e**. n = 3 biologically independent samples per group. Statistical significance was measured with one-way ANOVA with Tukey's multiple comparisons test. Comparisons between groups for DANG (from left to right): P = ns (0.9097), P = ns (0.5014), **P = 0.0025 and ****P < 0.0001. Comparisons between groups for PATU8988S (from left to right): ***P = 0.0002, ***P = 0.001, ****P < 0.0001 and ****P < 0.0001. **f**. Vinculin is used as a loading control. **g**. n = 3 biologically independent samples per group. Statistical significance was measured with one-way ANOVA with Dunnett's multiple comparisons test. Comparison between (−) and (+) doxycycline, ****P < 0.0001. Comparison between (−) doxycycline: vs 10 μM trametinib, P = ns (0.9997); 10 μM trametinib + doxycycline, P = ns (0.8226); 1 μM trametinib, P = ns (0.9997); 1 μM trametinib + doxycycline, P = ns (0.9994); 0.1 μM trametinib, P = ns (0.1904); 0.1 μM trametinib + doxycycline, P = ns (>0.9999). **h**. n = 3 biologically independent samples per group. Statistical significance was measured with one-way ANOVA with Tukey's multiple comparisons test. Comparisons between groups (from left to right): ****P < 0.0001, ****P < 0.0001, ****P < 0.0001 and ****P < 0.0001. The experiments (e, g, h) were performed once with similar results on UPP1 displayed by the three cell lines. **i**. n = 3 biologically independent samples per group. This blot was run on the same gel as Fig. 3n hence the first two columns (separated by a box) overlap between the two blots. **j**. Blots (c,i) are representative of two independent experiments; blot e experiment was done once. **k**. n = 3 biologically independent samples per group. The statistical significance (P < 0.05) was determined using *limma* package version 3.38.3 in R. **l**. Statistical significance was measured using one-way ANOVA with Tukey's multiple comparisons test. n = 4 biologically independent samples per group. PATU8988S (comparison between cells cultured with and without trametinib in the absence of uridine: ****P < 0.0001, and with uridine supplementation: ****P < 0.0001); DANG (comparison between cells cultured with and without trametinib in the absence of uridine: **P = 0.0055, and with uridine supplementation: ***P = 0.0009); ASPC1 (comparison between cells cultured with and without trametinib in the absence of uridine: P = not significant, and with uridine supplementation: ****P = 0.0006). Data (b, e, g-h, l) shown as mean ± s.d.

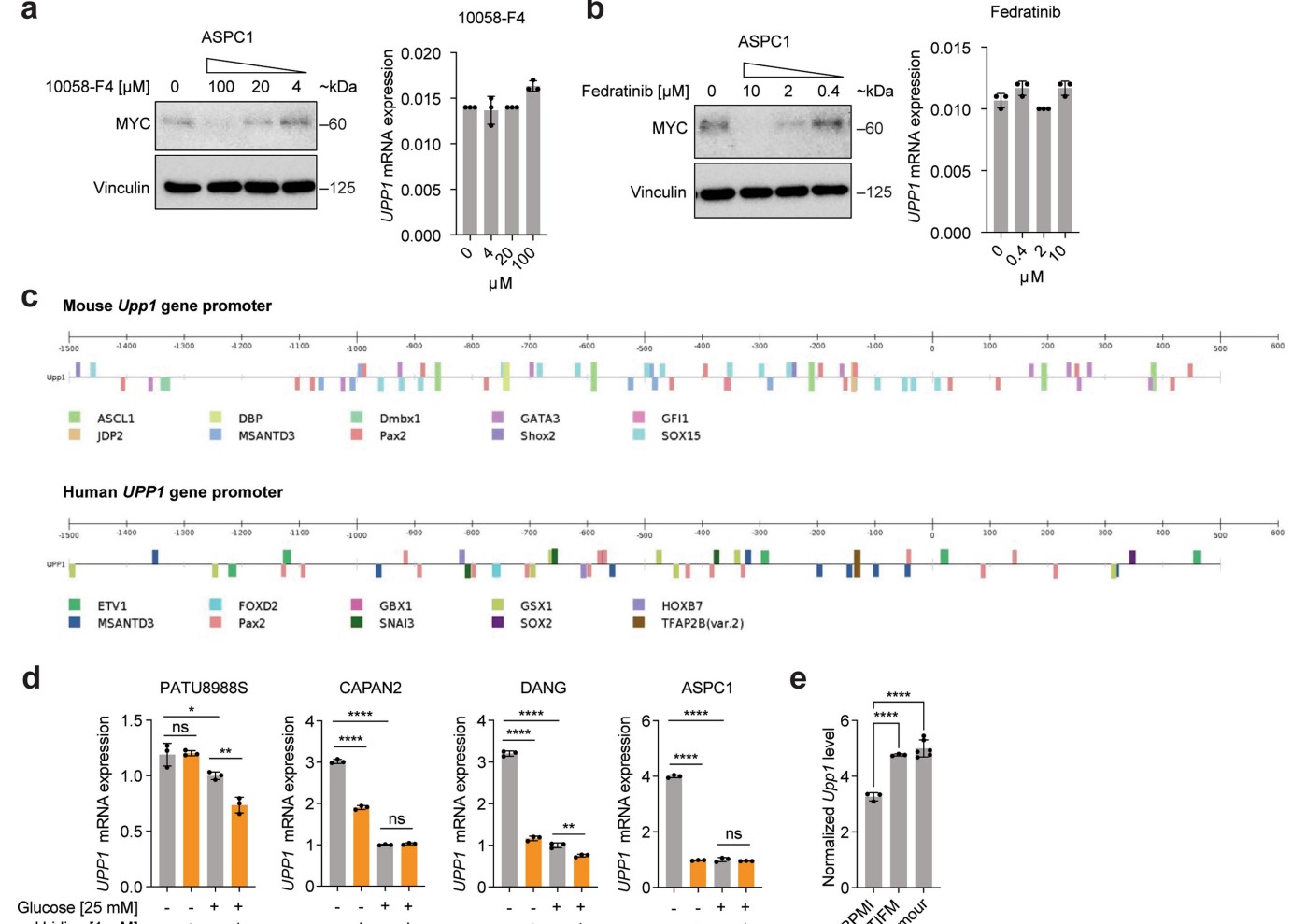

**Extended Data Fig. 9 | Regulation of *UPP1* expression is independent of c‑MYC. a**. Western blot showing Myc inhibition by 10058‑F4 in ASPC1 cells after 24 h of culture. On the right: *UPP1* mRNA expression determined by qPCR. kDa, unit for molecular weight. **b**. Western blot of Myc inhibition by Fedratinib in ASPC1 cells after 24 h of culture. On the right: *UPP1* mRNA expression determined by qPCR. **c**. CiiDER analysis of transcription factor binding sites in the promoters of mouse and human UPP1. Myc binding sites were not detected. Details of analysis is in the "Methods" section. **d**. qPCR showing *UPP1* expression upon uridine supplementation with or without basal glucose concentration in culture medium. **e**. RNA seq data showing the expression of *Upp1* in sorted tumour cells and in KPC cells cultured in vitro in regular RPMI culture medium or tumour interstitial fluid medium (TIFM). Sample sizes: Tumour, n = 6, RPMI, n = 3 biologically independent cell samples, TIFM, n = 3 biologically independent cell samples. Normalized by log transformation [log2 (count +1)]. Statistical significance was measured using one‑way ANOVA with Dunnett's multiple comparisons test. Comparison between RPMI and TIFM group, ****P < 0.0001 or tumour group, ****P < 0.0001. **Statistics and reproducibility: a**, qPCR, n = 3 biologically independent samples per group. **b**. n = 3 biologically

independent samples per group. Blots shown (a‑b) are representative of two biological and technical replicate analyses with similar results. **d**. n = 3 biologically independent samples per group per cell line. Statistical significance was measured using one‑way ANOVA with Tukey's multiple comparisons test. PATU8988S (comparison between no glucose and no glucose +1 mM uridine: P = ns (0.994); comparison between cells cultured in glucose‑containing medium with and without uridine: **P = 0.005); comparison between no glucose and glucose: *P = 0.0316; CAPAN2 (comparison between no glucose and no glucose +1 mM uridine: ****P < 0.0001; comparison between cells cultured in glucose‑containing medium with and without uridine: P = ns (0.8688); DANG (comparison between no glucose and no glucose +1 mM uridine: ****P < 0.0001; comparison between cells cultured in glucose‑containing medium with and without uridine: *P = 0.0021); ASPC1 (comparison between no glucose and no glucose +1 mM uridine: ****P < 0.0001; comparison between cells cultured in glucose‑containing medium with and without uridine: P = ns (0.5339). Comparison between no glucose and glucose for CAPAN2, DANG and ASPC1: ****P < 0.0001. Data (a,b,d,e) shown as mean ± s.d.

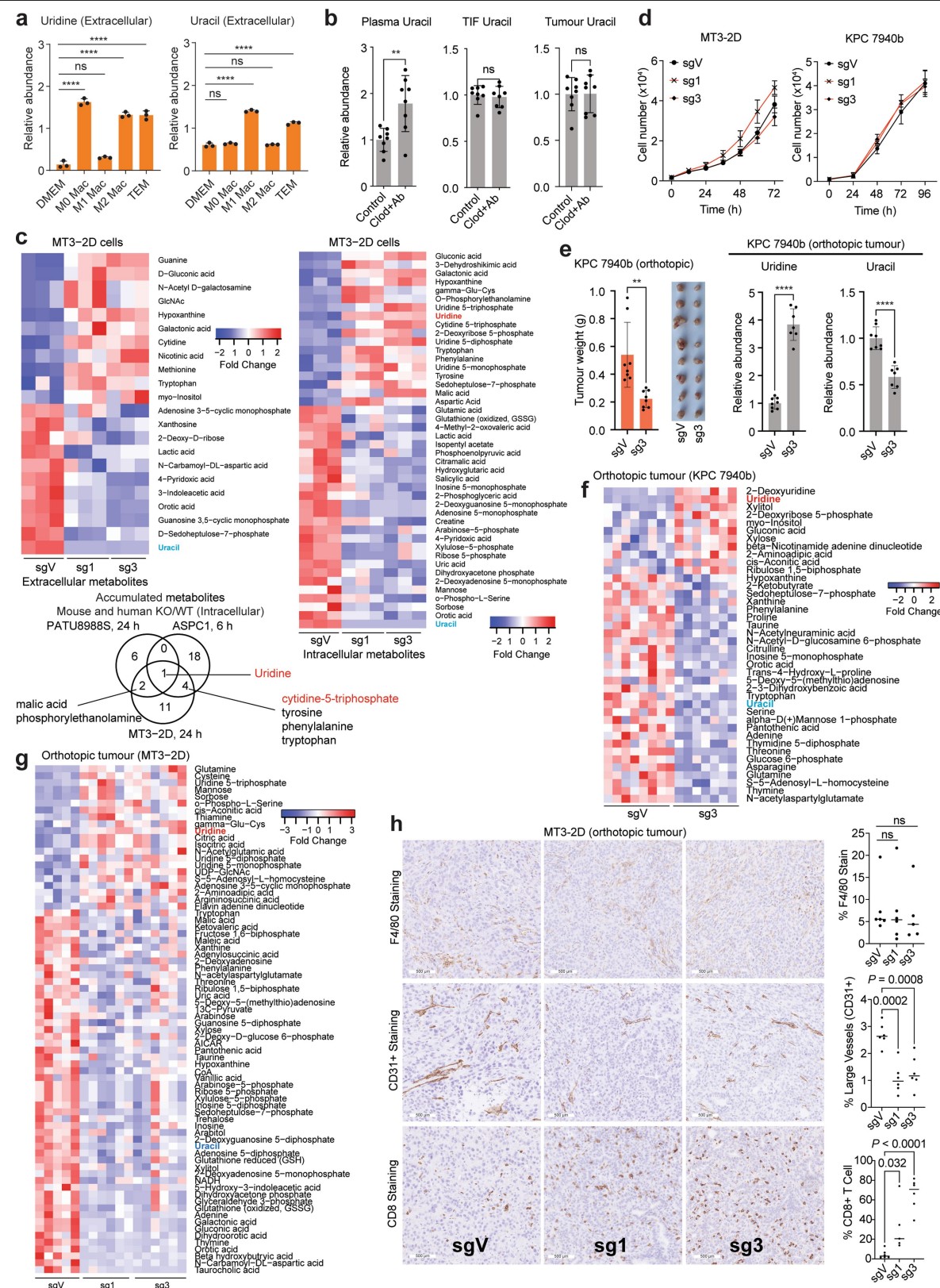

**Extended Data Fig. 10 |** See next page for caption.

**Extended Data Fig. 10 | Knockout of UPP1 suppresses *in vivo* uridine catabolism and tumour growth. a**. LC-MS analysis of extracellular uridine and uracil in unpolarized bone marrow-derived macrophages (M0; 10 ng/mL M-CSF), those polarized toward an M1 fate (10 ng/mL LPS), an M2 fate (10 ng/mL murine IL-4), or a tumour-educated (TEM) phenotype with 75% PDA conditioned medium and compared to growth medium (DMEM). n = 3 biologically independent samples per group. Data was extracted from a previously published metabolomics[14]. **b**. Relative uracil abundance in plasma, tumour interstitial fluid (TIF), and bulk tumour from the experiment described in Fig. 4a, where Clod + Ab indicates the macrophage-depleted group. **c**. Extracellular (left) and intracellular (right) profiles of significantly changed (P < 0.05) metabolites from sgV and UPP1-KO (sg1, sg3) MT3-2D cells. The cells were cultured in medium with 1 mM uridine and no glucose for 24 h, as determined by metabolomics. **d**. Proliferation assay of sgVector (sgV) and UPP1-KO (sg1, sg3) MT3-2D and KPC 7940b mouse PDA cell lines cultured over 72 h and 96 h, respectively, in normal growth medium with 10% dialyzed FBS. **e**. Tumour weight and photograph and bulk tumour uridine and uracil from the orthotopic implantation of sgV and UPP1-KO (sg3) KPC 7940b cell lines into C57BL/6J mice; see Fig. 4i for experimental details. **f,g**. Tumour metabolomics profile indicating statistically significant (*P* < 0.05) metabolites from sgV and UPP1-KO **f**) KPC 7940b and **g**) MT3-2D-derived tumours. **h**. Immunohistochemistry (IHC) staining of macrophages (F4/80), large blood vessels (CD31+), and cytotoxic cells (CD8) using tissue sections of MT3-2D orthotopic tumours. Micrographs are representative of 10 fields per image obtained per experiment group. Scale bar 500 μm. On the right is the respective quantification of each IHC stain. **Statistics and reproducibility: a**, Statistical significance was measured using one-way ANOVA and Dunnett's multiple comparisons test. Uridine – DMEM: comparison to M0 Mac group ****P < 0.0001; to M1 Mac group P = ns (0.0758); to M2 Mac group ****P < 0.0001; to TEM group ****P < 0.0001. Uracil – DMEM: comparison to M0 Mac P=ns (0.5476); to M1 Mac group ****P < 0.0001; to M2 Mac group P=ns (0.9894); to TEM group ****P < 0.0001. **b**. Statistical significance was measured using a two-tailed unpaired t-test – Plasma uracil **P = 0.0045 with n = 8 biologically independent replicates in both groups, TIF uracil P = ns (0.6872) with n = 8 biologically independent replicates in both groups– and using Welch's t-test P=ns (0.9682) with n = 8 biologically independent replicates in both groups. **c**. n = 3 biologically independent samples per group. Color scale denotes fold change. Below: Venn diagram showing overlapping metabolites that accumulated in both human (PATU8988S and ASPC1) and mouse (MT3-2D) cell lines upon UPP1 knockout. **d**. MT3-2D, n = 4 and KPC 7940b, n = 3 biologically independent samples per group. **e**. n = 8 biologically independent samples per group. Statistical significance measured using two-tailed unpaired t-test with Welch's correction, **P = 0.0059. On the right: bulk tumour uridine and uracil as measured using metabolomics. Sample size: sgV = 8, sg3 = 7. Uridine: ****P < 0.0001. Uracil: ****P < 0.0001. **f,g**. Sample size: KPC 7940b sgV = 8 and sg3 = 7, MT3-2D sgV = 5 sg1 = 6 and sg3 = 6 biologically independent samples from mice with orthotopic pancreatic tumours. **h**. Sample size: F4/80 sgV = 7, sg1 = 6, sg3 = 5; CD31 n = 6 per group; CD8+ sgV = 6, sg1 = 5, sg3 = 6, all biologically independent samples per group. Statistical significance was measured using one-way ANOVA with Dunnett's multiple comparisons test. Comparison between %F4/80 sgV and sg1 P = ns (>0.9968), and sgV and sg3 P = ns (0.9583). Data (a, b, d, e) shown as mean ± s.d; horizontal bars in h represent mean value.

# Reporting Summary

## Statistics

For all statistical analyses, confirm that the following items are present in the figure legend, table legend, main text, or Methods section.

| n/a | Confirmed | |
|---|---|---|
| ☐ | ☒ | The exact sample size (*n*) for each experimental group/condition, given as a discrete number and unit of measurement |
| ☐ | ☒ | A statement on whether measurements were taken from distinct samples or whether the same sample was measured repeatedly |
| ☐ | ☒ | The statistical test(s) used AND whether they are one- or two-sided<br>*Only common tests should be described solely by name; describe more complex techniques in the Methods section.* |
| ☒ | ☐ | A description of all covariates tested |
| ☐ | ☒ | A description of any assumptions or corrections, such as tests of normality and adjustment for multiple comparisons |
| ☐ | ☒ | A full description of the statistical parameters including central tendency (e.g. means) or other basic estimates (e.g. regression coefficient) AND variation (e.g. standard deviation) or associated estimates of uncertainty (e.g. confidence intervals) |
| ☐ | ☒ | For null hypothesis testing, the test statistic (e.g. *F*, *t*, *r*) with confidence intervals, effect sizes, degrees of freedom and *P* value noted<br>*Give P values as exact values whenever suitable.* |
| ☒ | ☐ | For Bayesian analysis, information on the choice of priors and Markov chain Monte Carlo settings |
| ☒ | ☐ | For hierarchical and complex designs, identification of the appropriate level for tests and full reporting of outcomes |
| ☒ | ☐ | Estimates of effect sizes (e.g. Cohen's *d*, Pearson's *r*), indicating how they were calculated |

*Our web collection on statistics for biologists contains articles on many of the points above.*

## Software and code

Policy information about availability of computer code

| Data collection | Biolog data was collected with the Omnilog instrument and software. Absorbance, luminescence, and fluorescence data was collected with SoftMax Pro 5.4.2 software. Agilent MassHunter software was used to collect and analyze metabolomics data. QuantStudio 3 and StepOne Plus version 2.3 software was used to collect real-time PCR data. |
|---|---|
| Data analysis | Biolog data was analyzed using the R opm package version 1.3.77. Immunohistochemistry data was analyzed with CellSens standard software and FIJI/Image J (version 1.53c). Antigen expression was scored using Definiens Test Studio Software (Definiens). Metabolomics data were analyzed with Agilent MassHunter Workstation Quantitative Analysis for QQQ Version 10.1, Build 10.1.733.0. Isotope enrichment data were analyzed with Agilent MassHunter Workstation Profinder Version 10.0, Build 10.0.10062.0 and Skyline Daily (version 22.2.1.256). Pathway analyses were performed using DAVID functional annotation platform (https://david.ncifcrf.gov/, version 6.8) or the gene set enrichment analysis (GSEA, version 4.0.3). Statistics were performed either with GraphPad Prism 8 (GraphPad Software Inc.) or using R version 3.5.2. For data analysis and visualization in R, packages (with versions) used include dplyr (0.8.3), ggplot2 (3.3.5), gplots (3.0.1, heatmap.2 function), ComplexHeatmap (2.3.5), tidyverse (1.3.0) and VennDiagram (1.6.20). |

For manuscripts utilizing custom algorithms or software that are central to the research but not yet described in published literature, software must be made available to editors and reviewers. We strongly encourage code deposition in a community repository (e.g. GitHub). See the Nature Portfolio guidelines for submitting code & software for further information.

## Data

Policy information about availability of data

All manuscripts must include a data availability statement. This statement should provide the following information, where applicable:
- Accession codes, unique identifiers, or web links for publicly available datasets
- A description of any restrictions on data availability
- For clinical datasets or third party data, please ensure that the statement adheres to our policy

Human PDA data used in this study are publicly accessible under the accession numbers indicated in the Methods section "PDA dataset analysis", "pan-cancer data analysis" and GSE36133 (for CCLE data). Other experimental data (e.g., the nutrient profiling and metabolomics data) have been summarized and presented in this study. The gene microarrays were obtained from NCBI gene expression omnibus, the ICGC-AU microarray data (release_28) was downloaded from https://dcc.icgc.org/projects/ along with the associated clinical data and had no embargo (March 2020) and the TCGA data was downloaded from CBioPortal (https://www.cbioportal.org/). CCLE protein data was accessed via DepMap portal (https://depmap.org/portal/). Human Protein Atlas data is available from: URL for 'Normal' - https://www.proteinatlas.org/ENSG00000183696-UPP1/tissue/pancreas; PDA – https://www.proteinatlas.org/ENSG00000183696-UPP1/pathology/pancreatic+cancer#img. The accompanying source data are provided as Supplemental Tables. All other data that support the findings of this study are available from the corresponding authors upon request.

## Human research participants

Policy information about studies involving human research participants and Sex and Gender in Research.

| | |
|---|---|
| Reporting on sex and gender | Sex and gender were not provided or used in the selection of slides for histological analysis. |
| Population characteristics | Human samples were from patients diagnosed with pancreatic cancer at the University of Michigan between 2021-2022 that were eligible for resection. No other population characteristics were considered in the analysis. |
| Recruitment | Deidentified samples from the tissue repository supported by IRB (HUM00025339) were used for histological analysis. |
| Ethics oversight | The collection of patient-derived samples was approved by the Institutional Review Board at the University of Michigan (IRB number: HUM00098128). |

Note that full information on the approval of the study protocol must also be provided in the manuscript.

# Field-specific reporting

Please select the one below that is the best fit for your research. If you are not sure, read the appropriate sections before making your selection.

☒ Life sciences    ☐ Behavioural & social sciences    ☐ Ecological, evolutionary & environmental sciences

For a reference copy of the document with all sections, see nature.com/documents/nr-reporting-summary-flat.pdf

# Life sciences study design

All studies must disclose on these points even when the disclosure is negative.

| | |
|---|---|
| Sample size | No sample size calculation was performed. Instead, sample sizes were chosen based on standard experimental group sizes to achieve acceptable power taking into account the increased variability of animal models (3-4 biological replicates for in vitro experiments, and 7-8 biological replicates for in vivo experiments). For metabolomics, isotope tracing, and qPCR experiments, sizes of 3-4 biological replicates were chosen based on the variability observed in pilot studies and in previous publications (citation below). For western blotting, generally one biological replicate representative of >=2 experiments was shown. For in vitro metabolic assays (e.g., MTT, Biolog, CellTiter Glo) generally 4 biological replicates were included in the experimental design based on the variation observed in earlier repetitions of the same experiment and in previous publications (citation below). Citation: Nwosu, Z. C. et al. Severe metabolic alterations in liver cancer lead to ERK pathway activation and drug resistance. EBioMedicine 54, (2020). |
| Data exclusions | No data were excluded from the analyses. |
| Replication | All experiments were successfully reproduced a minimum of two times with at least 3 biological replicates, with the exception of the metabolomics studies which were typically only run once using samples prepared from biological replicates (n=3). |
| Randomization | Experimental groups with mice were assigned randomly. Experimental groups for in vitro studies were not randomized as the cells are taken from a population assumed to be homogeneous and clonally identical. Experimental groups were not randomized for histological analysis of patient-derived biospecimens as the groups being compared (tumor and adjacent normal tissue) are paired. |
| Blinding | Blinding was not performed for in vitro or in vivo analyses as the measurements taken are objective. |

# Reporting for specific materials, systems and methods

We require information from authors about some types of materials, experimental systems and methods used in many studies. Here, indicate whether each material, system or method listed is relevant to your study. If you are not sure if a list item applies to your research, read the appropriate section before selecting a response.

## Materials & experimental systems

| n/a | Involved in the study |
|---|---|
| ☐ | ☒ Antibodies |
| ☐ | ☒ Eukaryotic cell lines |
| ☒ | ☐ Palaeontology and archaeology |
| ☐ | ☒ Animals and other organisms |
| ☒ | ☐ Clinical data |
| ☒ | ☐ Dual use research of concern |

## Methods

| n/a | Involved in the study |
|---|---|
| ☒ | ☐ ChIP-seq |
| ☒ | ☐ Flow cytometry |
| ☒ | ☐ MRI-based neuroimaging |

## Antibodies

**Antibodies used**

For western blotting, the following primary antibodies were used in this study and at 1:1000 dilution: anti-UPP1 (Sigma-Aldrich, # HPA055394), anti-c-MYC (Cell Signaling, # 5605S), anti-pERK (Cell Signaling, # 9106L), anti-ERK (Cell Signaling, # 9102S), anti-PGM2 (Invitrogen, # PA5-31378), and anti-Vinculin (Cell Signaling, # 13901S). The following secondary antibodies were used: anti-rabbit-HRP (Cell Signaling, # 7074S), and anti-mouse-HRP (Cell Signaling, # 7076P2).
For immunohistochemistry, the following primary antibodies were used in this study: Rabbit anti-UPP1 (1:200; Sigma-Aldrich, # HPA055394), Anti-Cd31 monoclonal antibody (1:75, clone SZ31, # DIA-310, Dianova), Anti-Cd8 monoclonal antibody (1:200, clone - 4SM15, # 14-0808, eBioscience), Anti-Cd3 polyclonal antibody (1:400, # ab5690, Abcam), and anti-F4/80 monoclonal antibody (1:100, Clone-A3-1, # MCA497G, Bio-Rad). The following secondary antibodies were used in this study: biotinylated secondary antibody (Horse anti-Rabbit; 1:500; Vector Labs, # BA-1100).
For RNAscope, primary antibody for panCK (Mouse anti-cytokeratin, pan reactive; 1:100; BioLegend; # 628602) was used.

**Validation**

anti-UPP1 (Sigma, HPA055394) - Figure 1e, Figure 3a
anti-c-MYC (Cell Signaling, 5605S) - Extended Figure 9a, 9b
anti-pERK (Cell Signaling, 9106L) - Extended Figure 8c
anti-ERK (Cell Signaling, 9102S) - Extended Figure 8c
anti-PGM2 (Invitrogen, PA5-31378) - Extended Figure 4i
anti-Vinculin (Cell Signaling, 13901S) - Figure 1e, Extended Figure 8c
anti-cytokeratin (BioLegend, 628602) - Figure 3g, Extended Figure 7a

## Eukaryotic cell lines

Policy information about cell lines and Sex and Gender in Research

**Cell line source(s)**

The panel of PDA cell lines, HPNE, A549, HT1080, HCT116, and U2OS, were purchased from the American Type Culture Collection (ATCC) or the German Collection of Microorganisms (DSMZ). The hPSC cell line and the mouse cell lines KPC 7940b and MT3-2D were generously provided under MTA by Rosa Hwang (MD Anderson Cancer Center), Gregory Beatty (University of Pennsylvania), and David Tuveson (Cold Spring Harbor Labs), respectively. iKras cell lines A9993 and iKRAS 9805 were derived as previously described (Collins et al., 2012, JCI, PMID: 22232209).

**Authentication**

All cell lines were authenticated with STR genetic testing.

**Mycoplasma contamination**

Cell lines regularly tested negative for mycoplasma using the MycoAlert kit (Lonza).

**Commonly misidentified lines**
(See ICLAC register)

None used.

## Animals and other research organisms

Policy information about studies involving animals; ARRIVE guidelines recommended for reporting animal research, and Sex and Gender in Research

**Laboratory animals**

6-8 weeks old and 8-12 weeks old C57BL/6J mice (The Jackson Laboratory (Strain # 000664); ~6 weeks old C57BL/6NCrl mice (Charles River Laboratories (Strain Code 027).

**Wild animals**

None used.

**Reporting on sex**

Studies were carried out with a mix of male and female mice or only female mice. No specific method was used for assigning sex and sex-based difference was not observed.

| Field-collected samples | None used. |
| --- | --- |
| Ethics oversight | Animal studies were performed at the University of Michigan (UM), the Institute of Cancer Research (ICR), and the University of Chicago (UChicago) according to approved protocols. UM  Institutional Animal Care and Use Committee (IACUC) PRO00010606; ICR studies conformed to UK Home Office Regulations under the Animals Scientific Procedures Act 1986 and national guidelines (Project licence-P0A54750A protocol 5); UChicago IACUC Protocol #72587. |

Note that full information on the approval of the study protocol must also be provided in the manuscript.

