## [Peer Review File · Nature]

Manuscript Title: Uridine-derived ribose fuels glucose-restricted pancreatic cancer

Reviewer Comments & Author Rebuttals

Reviewer Reports on the Initial Version:

Referee expertise:

Referee #1:

Referee #2:

Referee #3:

Referees' comments:

Referee #1 (Remarks to the Author):

Ward and colleagues report that pancreatic cancer cells metabolize uridine when nutrient availability, particularly glucose, is limiting. A subset of pancreatic cancer cells upregulate the phosphorylase UPP1 to catabolize uridine into uracil and ribose 1-phosphate. The authors suggest that ribose 1-P derived from uridine can be used by the pentose phosphate, glycolysis, and TCA cycle intermediates to maintain pancreatic tumor cell survival.

General comments:

This is an elegant study that provides essential new knowledge on the metabolic dependencies of pancreatic tumor cells. However, the role of uridine catabolism and UPP1 in the context of glucose deprivation appears to have already been documented in the literature (PMID: 18457515, PMID: 16839635, PMID: 6790526). Unfortunately, the aforementioned literature takes away the novelty of the concept of this study. Nevertheless, this work demonstrates, through unbiased methods, the critical role of uridine catabolism for pancreatic cancer cell survival and, therefore, should be reported but probably in a more specialized journal.

Specific comments:

1) The authors report that UPP1 is upregulated in PDA. It will be essential to define the mechanisms by which UPP1 is increased in PDA cells. Do KRAS and MYC correlate with UPP1 expression? How is UPP1 transcriptionally regulated in PDA cells? Do PDA cells express UPP2? Is UPP2 upregulated too?

2) It is unclear which cells produce uridine in the tumor microenvironment? Uridine and uracil levels can reach micromolar levels in the tumor interstitial fluid, and tumor-associated macrophages can release uridine and uracil. Therefore, inhibiting macrophage function in PDA tumors and measuring uridine levels should be performed to determine whether macrophages can produce uridine in the TME for cancer cell survival.

3) Uridine can be metabolized by the pyrimidine salvage pathway through uridine-cytidine kinase enzymes (UCK1/2). UCK converts uridine or cytidine into UMP or CMP. It will be important to knock out or knock down these enzymes (UCK1 and UCK2) to assess whether uridine catabolism can still allow at least a partial cell survival rescue in response to glucose deprivation independently of pyrimidine nucleotide salvage.

4) The authors suggest UPP1 produces ribose 1-P, which can be used by the non-oxidative PPP to replenish glycolytic intermediates and TCA cycle function. However, the PPP enzymes (oxidative and non-oxidative) and nucleotide enzymes are known to use ribose 5-P as a precursor, not ribose 1-P. Therefore, the authors should assess phosphoglucomutase 2 (PGM2) levels and activity in PDA cells.

5) PRPS1/2 enzymes use ribose 5-P to produce PRPP, a substrate for NAD and nucleotide synthesis. Therefore, measurements of PRPP levels derived from uridine catabolism should be measured.

6) Measurements of glucose concentration in the TIF could be helpful. Is glucose limiting in the TME? What is the concentration of glucose that renders PDA cells dependent on uridine? A titration of glucose along with uridine catabolism measurement would be informative.

Referee #2 (Remarks to the Author):

In this manuscript by Ward, Nwosu, Poudel and colleagues, the authors analyzed the capacity of diverse nutrients to support PDAC cell metabolism in the presence of sub-physiologic levels of glutamine and the absence of glucose. Their initial screen, together with subsequent metabolomics and stable isotope tracing experiments, demonstrated that uridine can support PDAC cell metabolism under these nutrient-deprived culture conditions. The authors performed a strong characterization of uridine metabolism in PDAC cells including its metabolic fate, and convincingly demonstrated the requirement for UPP1 for uridine-to-uracil conversion and downstream fates of catabolized uridine. A role for extracellular uridine and for UPP1 in PDAC cell metabolism and viability is entirely novel and potentially of great interest. However, the study at present is descriptive, and lacks compelling evidence to support a functional role for uridine/UPP1 in PDAC. The main weakness of the study is the questionable relevance of uridine and UPP1 for pancreatic cancer cell metabolic fitness within an intact tumor. Numerous published studies, including excellent prior work by these authors, support the notion that PDAC cells employ diverse mechanisms to scavenge nutrients from the extracellular space to support key metabolic processes under nutrient-limiting conditions. While the cell culture studies in the manuscript clearly demonstrate that pancreatic cancer cells can take up and utilize uridine to support diverse metabolic processes, whether they do to a meaningful extent in vivo remains unclear, where diverse additional metabolites are available as are other cell types which may compete for nutrients. The particular cell culture conditions used throughout the study are quite flawed as detailed below, and only a single experiment was performed in vivo which yielded results rather difficult to interpret and was sub-optimally designed. These weaknesses lessen enthusiasm for the study.

Specific comments:

1. In the Biolog assay presented in Figure 1, and in subsequent validation experiments, the authors subject PDAC cells to a total absence of glucose. As glucose concentration in the PDAC microenvironment are greater than 1mM, it seems quite unlikely that cancer cells experience zero glucose in tumors. Where near-physiologic 1mM glucose is used on PDAC cells in Figure 2, it is used on its own, and the ability of uridine to augment relative metabolic activity in this context is not shown. Uridine plus glucose is only shown in Figure 4 in the presence of very high glucose concentrations. Does uridine significantly impact PDAC cell metabolism in the presence of relevant glucose concentrations? If the experiment depicted in Figure 4b is repeated in the presence of lower glucose concentrations, does uridine impact NADH levels?
2. The correlative data presented in Figure 1, ED Figure 3, and the ED Tables with respect to UPP1 in PDAC are interesting, but certainly do not support the conclusion (line 119) that “UPP1 is a critical metabolic enzyme in PDA.” This statement should be amended.
3. Uridine is used at 1mM throughout the manuscript, but its levels in PDAC interstitial fluid are approximately 20-fold lower than that, so it is hard to know whether the metabolic assays performed in vitro are relevant. Key experiments should be repeated in the presence of an appropriate uridine concentration.
4. The tumor growth experiment shown in Figure 4i needs further validation in light of the variability between UPP1 knockout clones, as it is unconvincing at present that UPP1 indeed supports PDAC growth in mice. UPP1 should be restored in clone 1A to confirm specificity, or an additional clone should be tested. Further, in light of the authors’ proposed model in Figure 4j implicating the tumor microenvironment, the role of UPP1 in PDAC growth should be tested in the relevant tissue environment (pancreas) instead of subQ. As Figs 4h and ED Fig 4f,g show UPP1 expression in adaptive immune cells, competition for uridine between cancer cells and tumor-infiltrating leukocytes may indeed be relevant, so UPP1-null PDAC cells should be tested in an immune-competent, syngeneic model.
5. (Minor) It seems the y-axis is mislabeled in the left graph in Figure 4i (presumably tumor weights are shown in grams, not milligrams).

Referee #3 (Remarks to the Author):

Ward et al. performed experiments to determine which of an array of metabolites induce dye reduction in otherwise nutrient-limited pancreatic ductal adenocarcinoma (PDA) cells. They found that uridine could be extensively metabolized in these cells, and that its ribose backbone fuels glycolysis and the TCA cycle. The enzyme UPP1 is required for cells to use uridine to support cell growth when glucose is unavailable. Gene expression profiling indicates that UPP1 is modestly up-regulated in PDA tumors, and that its expression correlates with poor outcomes in most cohorts. Tracing experiments with isotope-labeled uridine demonstrate that the ribose ring enters glycolysis and ultimately feeds bioenergetic and biosynthetic pathways. Knocking out UPP1 in one PDAC line reduces xenograft growth (although inconsistently between two clones). The underlying finding – that uridine can be a meaningful source of carbon from central metabolism

– is interesting and could be relevant in tumors where more conventional nutrients like glucose are scarce. However, there are numerous issues in the paper that need to be addressed.

1. Uridine stands out in the Biolog assay as one of the most commonly metabolized nutrients among the PDA cells, but not by HPNE and HPSC cells. Given the fact that uridine metabolism correlates with glucose metabolism, the authors should examine a few non-PDA cell lines for uridine-dependent NADH production.

2. There are a few concerns about the UPP1 expression data. Although UPP1 expression is discussed throughout the paper, the authors never validate UPP1 levels with a western blot. Protein expression should be correlated with uridine-dependent RMA (as shown for RNA levels in Fig. 1d), and to validate UPP1 loss in CRISPR KO cells.

3. The expression data from human PDA (Fig. 1g) are not convincing. Protein expression or IHC might really help here, because the levels of mRNA levels are barely different between tumor and non-tumor samples. Is nonmalignant pancreas unusually high among the organs in UPP1 expression?

4. Similarly, the evidence that UPP1 is driven by KRAS (Fig. 1f, ED 3d) is not convincing. Further insights into the mechanism of UPP1 expression would be helpful. Does UPP1 expression also correlate with oncogenic KRAS in other types of cancer (public databases like TCGA and CCLE would be helpful here)? Does suppressing KRAS expression also reduce uridine's ability to sustain metabolism in the absence of glucose?

5. The informatics analyses need to be presented more clearly. UPP1-expressing cells display “downregulation of metabolic pathways,” but this is too superficial for a paper about metabolism. It provides no information about which pathways are co-regulated with UPP1. The authors highlight immune-related genes among the DEGs “potentially link UPP1 to TME activities.” This is potentially interesting, but not informative as written. Can the authors better link this finding to the data later in the paper showing that uridine is released by polarized macrophages? The implication is that macrophage-conditioned medium – or at least medium containing the same concentration of uridine as released by macrophages – can sustain PDA cell metabolism in glucose-deprived conditions, in a UPP1-dependent manner.

6. The isotope labeling experiments are generally informative, but how do TU8988S cells manage to label their TCA cycle intermediates so well despite having almost no labeling in pyruvate? This finding does not fit the proposed mechanism.

7. The authors comment that uridine is more abundant in PDA TIF than in blood. But the experiments that generated the dataset queried by the authors used a PDA GEMM, and one of the conclusions of that paper was that the PDA TIF contains millimolar levels of glucose, rather than being severely glucose depleted as commonly thought. This makes it unclear how the data from these TIF experiments relate to the pathway proposed here by the authors, where uridine drives metabolism in the absence of glucose. It would be more relevant to assess glucose and uridine levels in the xenograft models being used in this paper, since the authors argue that UPP1 is required for growth of those models.

8. The evidence that this pathway is important in vivo is limited. In Figure 4i, two clones display disparate growth rates in mice, and we do not know whether residual levels of UPP1 differ between them. The authors need to address this issue and show that UPP1 dependence extends to other PDA tumor models that express UPP1. It would also be good to know whether UPP1 loss increases uridine levels in the tumors.

Minor concerns/questions:

1. It is curious that uridine supplementation reverses UPP1 induction, since UPP1 is required to catabolize uridine. Can the authors explain this?
2. Line 210: Change “uridine-derived glucose” to “uridine-derived ribose”

Author Rebuttals to Initial Comments:

Rebuttal of Ward, Nwosu, et al.

We are truly grateful for the time and effort put forth by the Editors and Reviewers at *Nature*. We have addressed all the experimental and textual concerns raised, and we believe that the helpful comments have allowed us to considerably strengthen the conclusions presented in the accompanying manuscript.

Six key highlights summarizing our updates in this revised study follow:

- **Previously we illustrated that uridine-derived ribose can serve as a nutrient source in glucose-starved pancreatic cancer cells. This supports both bioenergetics and biosynthesis by fueling central carbon metabolism. We extend these findings to demonstrate this *in vitro* (using *in vivo* relevant concentrations) and *in vivo* using isotope tracing and metabolomics. Our data provide the first clear demonstration for these novel functions of uridine-derived ribose.**
- **We conclusively demonstrate that uridine-derived ribose liberation is mediated by UPP1 in pancreatic cancer.**
- **We provide new evidence to illustrate that UPP1 is transcriptionally activated by KRAS-MAPK signaling, and that UPP1 expression is further augmented under nutrient restricted conditions. These results illuminate both a cancer cell specific dependence and physiological rationale for the activation of uridinolysis.**
- **Using public databases, we illustrate that UPP1 expression is elevated in pancreatic and other cancers, and this is predictive of shorter patient survival. We validated this work with single cell sequencing analysis, in situ-RNA hybridization, and IHC in patient samples generated at the University of Michigan.**
- **Using immunocompetent mouse models of pancreatic cancer, we found that UPP1 knockout potently blocked tumor growth, illustrating a novel dependence and potential therapeutic target.**
- **Lastly, we are co-submitting this work alongside a paper from the Mootha lab, which arrived at identical conclusions regarding the role of uridine as a metabolic fuel in glucose limiting conditions (Jourdain et al. *bioRxiv* 2021). Like our study, they demonstrate that uridinolysis provides uridine derived ribose to fuel central carbon metabolism in a UPP1-dependent manner, providing independent support and illustrating the robustness of the mechanism.**

Below is a point-by-point response to the referee comments; referee remarks are presented in plain text, our responses in bold.

REVIEWERS' COMMENTS

Reviewer 1

Ward and colleagues report that pancreatic cancer cells metabolize uridine when nutrient availability, particularly glucose, is limiting. A subset of pancreatic cancer cells upregulate the phosphorylase UPP1 to catabolize uridine into uracil and ribose 1-phosphate. The authors suggest that ribose 1-P derived from uridine can be used by the pentose phosphate, glycolysis, and TCA cycle intermediates to maintain pancreatic tumor cell survival.

General comments:

This is an elegant study that provides essential new knowledge on the metabolic dependencies of pancreatic tumor cells. However, the role of uridine catabolism and UPP1 in the context of glucose deprivation appears to have already been documented in the literature (PMID: 18457515, PMID: 16839635, PMID: 6790526). Unfortunately, the aforementioned literature takes away the novelty of the concept of this study. Nevertheless, this work demonstrates, through unbiased methods, the critical role of uridine catabolism for pancreatic cancer cell survival and, therefore, should be reported but probably in a more specialized journal.

Response: We thank the reviewer for their careful reading of our work and supportive feedback. The previous publications on uridine and glucose cited above concluded that the primary function of uridine was to provide a source of nucleotides and energy during glucose deprivation.

Our work, which was substantially bolstered and extended during the revision, extends well beyond the conclusions from these previous studies. We illustrate how a KRAS-regulated uridine-UPP1 axis provides a nutrient source to glucose-deprived pancreatic cancer cells *in vivo*, while also highlighting a new metabolic vulnerability in this notoriously difficult to treat cancer. These findings are briefly summarized below, as follows:

***First*, our work confirms the role of uridine and UPP1 in the support of nucleotide and energy functions upon glucose deprivation (Figure 2, Extended Data Figures 2,3), as previously reported.**

***Second*, and more importantly, our studies extend these findings by providing conclusive evidence for UPP1-mediated, uridine-derived ribose contributing to the panoply of metabolic fates of glucose, including glycolysis, glycosylation biosynthesis, non-essential amino acid biosynthesis, de novo nucleotide biosynthesis, TCA cycling and mitochondrial metabolism, and redox metabolite biosynthesis (Figure 3, Extended Data Figure 5). We demonstrate this *in vitro* and *in vivo* using isotope tracing and metabolomics (Figure 2, Extended Data Figure 3), and thereby provide the first conclusive evidence for these novel functions of UPP1-mediated, uridine-derived ribose.**

Third, in these revisions we extend and build on our previous work to illustrate an important and previously undescribed role for UPP1 and uridine in pancreatic cancer. We find that pancreatic cancer cells can use uridine in place of glucose in cell based and tumor models (Figure 2, Extended Data Figures 2-4). Further to this point, we also found that UPP1 expression is upregulated in pancreatic cancer in a manner that is dependent upon signaling through the MAPK pathway downstream of mutant Kras signaling (Figure 3l-o, Extended Data Figure 8). We also illustrate that UPP1 expression is further modulated by the energy status of the cell (Figure 3n, Extended Data Figure 8). In the absence of glucose and/or uridine, UPP1 expression is augmented. Together, these results provide a new understanding of the nutrient utilization profile and mechanisms of nutrient metabolism in pancreatic tumors *in vivo*.

Fourth, using immune-competent murine pancreatic tumor models, we found that inhibition of UPP1 expression blocked tumor growth (Figure 4, Extended Data Figure 10). The same cell lines grown in culture did not exhibit proliferative defects, highlighting a potential metabolic compensatory pathway operative in the nutrient deregulated pancreatic tumor microenvironment.

In sum, we believe these results provide the sort of paradigm-shifting thinking in tumor metabolism, oncogenic and nutrient signaling, and therapy that will be of interest to the general readership of *Nature*.

Specific comments:

1) The authors report that UPP1 is upregulated in PDA. It will be essential to define the mechanisms by which UPP1 is increased in PDA cells. Do KRAS and MYC correlate with UPP1 expression? How is UPP1 transcriptionally regulated in PDA cells?

Response: We appreciate this insightful comment from the referee, which has provided the opportunity to identify an important mechanism of UPP1 regulation in pancreatic cancer.

In our initial submission we included transcriptome profiling data from our mouse model with doxycycline-inducible mutant *Kras* expression. These data indicated that mutant *Kras* induction in murine PDA cell lines and tumors correlated with increased *Upp1* (Extended Data Figure 8b). To test this directly, we employed our inducible mutant *Kras* cell lines *in vitro*, confirming that mutant *Kras* activation significantly increases *Upp1* expression (Figure 3l,m; Reviewer Figure 1a). In pancreatic cancer, mutant *Kras* signals through the MAPK pathway to support tumor metabolism (Ying, et al. *Cell* 2012; Perera RM, Bardeesy N. *Can Disc* 2015; Santana-Codina, et al. *Nat Comm* 2018). Thus, we then employed inhibitors of MAPK signaling to determine the impact on UPP1 expression. These studies revealed that the regulation of UPP1 was downstream of MAPK in multiple human and mouse pancreatic cancer cell lines, at both the mRNA and protein expression levels. This closely correlated with pERK expression and response to the MEK inhibitor trametinib (Figure 3n,o, Extended Data Figure 8c-l; Reviewer Figure 1b-h). Lastly, we also now include TCGA data for pancreatic tumors expressing no alterations in KRAS or

Reviewer Figure 1. **a)** *Upp1* expression in murine iKras pancreatic cancer cell lines 9805 and A9993 following the activation of mutant *Kras* expression (with doxycycline, Dox). **b)** Western blot demonstrates activation of pERK with Dox treatment; Vinculin serves as the loading control. **c)** pERK and **d)** *Upp1* expression in iKras cell line 9805 treated with the MEK inhibitor trametinib. Western blot of pERK with doxycycline withdrawal or MEK inhibition; Vinculin serves as the loading control. **e)** *UPP1* expression in human PDA cell line ASPC1 treated with trametinib +/- glucose and uridine and **f)** accompanying Western blot. **g,h)** *UPP1* expression in two additional human PDA cell lines DANG and TU8988S treated with trametinib +/- glucose and uridine. **i)** *UPP1* expression in pancreatic tumors without *KRAS* alterations (No Alt) or *KRAS*-G12D mutation from TCGA database.

G12D mutation, where we also find significantly higher *UPP1* expression in the *KRAS* G12D mutated tumors (Figure 3k; Reviewer Figure 1i).

Further on the mechanistic regulation of *UPP1*, we also now demonstrate that low uridine or low glucose concentrations similarly promote *UPP1* expression (Figure 3n,o, Extended Data Figure 8c-l; Reviewer Figure 1d-h). This makes mechanistic sense, in that when nutrient availability is decreased, *UPP1* expression is up-regulated to support enhanced ribose scavenging from uridine. This nutrient regulation of *UPP1* appears to function as more a rheostat, enhancing the expression primarily mediated by MAPK signaling.

In our previous work, we demonstrated that the mutant *Kras* mediated regulation of anabolic glucose metabolism in pancreatic cancer occurred downstream of MAPK and was transcriptionally regulated by *Myc* (Ying, et al. *Cell* 2012). In other words, we absolutely agree with the referee that *MYC* was a logical starting point for deciphering the regulation of *UPP1* expression. To this end, we first looked in the ICGC dataset,

where we found that MYC is not significantly correlated with UPP1 (Reviewer Figure 2a). Next, we performed an *in silico* prediction of transcription factor binding sites in human and mouse *Upp1* gene using CiiDER (Gearing, et al. *PLOS One* 2019). Here, we did not identify MYC binding sites in the *UPP1* promoter; however, we did identify multiple additional candidate transcription factors (Extended Data Figure 9c; Reviewer Figure 2b). Finally, we also directly tested the impact of MYC on UPP1 expression in pancreatic cancer cells. We employed two independent pharmacological inhibitors of MYC, both of which brought down MYC expression by western. In contrast, we did not observe effects on UPP1 expression (Extended Data Figure 9a,b; Reviewer Figure 2c,d).

In sum, our data illustrate that UPP1 is regulated downstream of mutant KRAS-mediated MAPK signaling in pancreatic cancer, that this signaling axis is further augmented by nutrient (glucose or uridine) deprivation, and that this is independent of MYC.

Reviewer Figure 2. **a)** Correlation of *MYC* and *UPP1* in the International Cancer Genome Consortium (ICGC) PDA datasets. **b)** *In silico* prediction of *UPP1* gene transcription factor binding sites in human and mouse using CiiDER showing that MYC does not bind to *UPP1* promoter region. **c,d)** Immunoblot and qPCR showing that MYC inhibition with 10058-F4 or Fedratinib does not impact UPP1 expression in a UPP1-high expressing PDA cell line, ASPC1. Vinculin serves as the loading control.

Do PDA cells express UPP2? Is UPP2 upregulated too?

Response: Analysis of UPP1 and UPP2 expression in TCGA pancreatic cancer dataset revealed that the fragments per kilobase of transcript per million mapped reads (FPKM) for UPP2 was <1, compared to >500 for UPP1 (Extended Data Figure 6a; Reviewer Figure 3a). By qPCR analysis in our human pancreatic cancer cell lines, we detected UPP1

transcript after ~20-24 cycles; detection of UPP2 required >36 cycles (Reviewer Figure 3b). These data are consistent with a gene that is not expressed.

2) It is unclear which cells produce uridine in the tumor microenvironment? Uridine and uracil levels can reach micromolar levels in the tumor interstitial fluid, and tumor-associated macrophages can release uridine and uracil. Therefore, inhibiting macrophage function in PDA tumors and measuring uridine levels should be performed to determine whether macrophages can produce uridine in the TME for cancer cell survival.

Reviewer Figure 3. a) *UPP1* and *UPP2* expression from the TCGA. b) qPCR analysis of *UPP1* and *UPP2* expression in the human pancreatic cancer cell line ASPC1.

Response: We agree with the referee that this is an important question and thank them for the suggestion. To this end, we first measured the uridine in the tumor interstitial fluid (TIF) from murine pancreatic orthotopic allograft tumors. Uridine was present in the micromolar range (Figure 2h; Reviewer Figure 4a), consistent with previous publications (Sullivan, et al. *eLife* 2019).

Next, we tested the impact of tumor associated macrophages (TAMs) on the production of intratumoral uridine directly by depleting macrophages from murine allograft tumors using the combination of CSF1 antibody and clodronate (liposome) (Figure 4a; Reviewer Figure 4b). As reported previously, this combination depleted TAMs by ~50% and suppressed orthotopic tumor growth (Zhang, et al. *Gut* 2017; Candido, et al. *Cell Reports* 2018) (Figure 4a,b; Reviewer Figure 4c,d). As it relates to uridine and uracil, we observed a reduction in plasma uridine level by ~8 fold upon macrophage depletion. Uracil levels in the serum were not altered (Figure 4c, Extended Data Figure 10b; Reviewer Figure 4e,f). This marked impact on plasma uridine levels following macrophage depletion indicates that macrophages may be major mediators of uridine production/release. However, despite the considerable impact on plasma uridine, tumor and interstitial fluid uridine was not changed (Figure 4c, Reviewer Figure 4g,h).

These *in vivo* tumor data contrast our *in vitro* models, which indicated that anti-inflammatory TAMs release micromolar uridine (Extended Data Figure 10a). However, there is considerable precedent for compensatory activities from other cell types in the TME when one cell type is depleted. For example, an increase in immunosuppressive myeloid subtypes were reported when Tregs are depleted from pancreatic tumors (Zhang, et al. *Cancer Discovery* 2020), and there is competing activity of deoxycytidine release from TAMs and CAFs (Halbrook, et al. *Cell Metabolism* 2019; Dalin, et al. *Cancer Research* 2019). Thus, it is conceivable that uridine levels were maintained from a compensatory cell type in our macrophage depletion model.

Reviewer Figure 4. a) Quantitation of uridine and uracil levels in the tumor interstitial fluid (TIF) from pancreatic orthotopic allograft tumors. b) Schematic of macrophage depletion in C57BL/6J immunocompetent mice. c) Tumor weight following the depletion of tumor associated macrophages (TAMs). d) Quantitation of macrophage depletion from tumors, as assessed by F4/80 staining. Representative images at right. e) Plasma uridine and f) uracil, and g) TIF uridine and h) uracil concentration in the control and macrophage-depleted mice samples.

In a parallel experiment, we administered isotope-labeled uridine to pancreatic tumor bearing animals to study uridine metabolism *in vivo*. Animals were sacrificed one hour after injection, and tumor tissue was collected to measure uridine uptake and utilization (Figure 2g, Extended Data Figure 3a; Reviewer Figure 5). Consistent with our *in vitro* studies, and despite the short experimental duration, we observed contribution of uridine-derived ribose to nucleotides, glycolytic and branching pathways, and TCA cycle metabolites. In addition to demonstrating that exogenous uridine is readily captured and metabolized to fuel central carbon metabolism, these data also indicate that uridine in circulation is used by the tumor. In other words, while we had previously suggested that TAMs were a principal source of uridine, these data indicate that uridine is likely coming from multiple sources, including distant sources through circulation (e.g. liver) as well as locally from the TME.

In sum, based on these new data, we have modified our conclusions to acknowledge that while macrophages may be a source of uridine, other cell types also contribute to the TME and systemic uridine pools.

Reviewer Figure 5. Fractional labeling of uridine-derived ribose carbon indicating uptake and utilization in subcutaneous (SubQ) and pancreatic orthotopic (Ortho) tumors 1 hour after injection. SubQ tumors were directly injected with PBS or 0.2M uridine in 50 μ L PBS. The orthotopic tumor-bearing mice were injected IP with PBS or 0.2M uridine in 200 μ L PBS.

3) Uridine can be metabolized by the pyrimidine salvage pathway through uridine-cytidine kinase enzymes (UCK1/2). UCK converts uridine or cytidine into UMP or CMP. It will be important to knock out or knock down these enzymes (UCK1 and UCK2) to assess whether uridine catabolism can still allow at least a partial cell survival rescue in response to glucose deprivation independently of pyrimidine nucleotide salvage.

Response: We appreciate this insightful suggestion. We performed UCK1 or UCK2 knockdown by siRNA and found that uridine still rescued the bioenergetic defects of glucose restriction (Extended Data Figure 4k,l; Reviewer Figure 6a,b). These data indicate that the catabolism of uridine-derived ribose and its entry into central carbon

metabolism, rather than the pyrimidine salvage pathway, is the mediator of rescue of glucose restriction.

4) The authors suggest UPP1 produces ribose 1-P, which can be used by the non-oxidative PPP to replenish glycolytic intermediates and TCA cycle function. However, the PPP enzymes (oxidative and non-oxidative) and nucleotide enzymes are known to use ribose 5-P as a precursor, not ribose 1-P. Therefore, the authors should assess phosphoglucomutase 2 (PGM2) levels and activity in PDA cells.

Response: This too is an important point required to clarify our model. First, we analyzed *PGM2* expression and found that it is upregulated in PDA (Extended Data Figure 4e; Reviewer Figure 6c). However, *PGM2* is not correlated with *UPP1*: patients and cell lines with high *UPP1* expression do not show differential *PGM2* expression (Extended Data Figure 4f,i; Reviewer Figure 6d,e). Next, we interrogated the activity of *PGM2* by knocking it down and following the uridine rescue of glucose restriction in pancreatic cancer cells, as above. In accordance with the referee suggestion and our model, siRNA-mediated

Reviewer Figure 6. a,b) (left) qPCR data reflecting *UCK* expression with non-targeting (NT) or *UCK*-targeting siRNA in ASPC1 human pancreatic cancer cells. (right) Relative metabolic activity (RMA) in siUCK cells plus or minus 1 mM glucose or 1 mM uridine, compared to siNT-treated cells. c) *PGM2* expression in human PDA dataset GSE71729 (NT=46, PDA=145). d) *PGM2* expression in UPP1 high/low tumors from TCGA (UPP1 high = 75, low = 75) and PDA cell lines (UPP1 high = 22, low = 22). e) Western blot for *PGM2* and UPP1 in a panel of PDA cell lines. Vinculin serves as the loading control. f) (left) *PGM2* expression following siRNA knockdown relative to non-targeting (NT) control. (right) RMA in siPGM2 cells plus or minus 1 mM glucose or 1 mM uridine, compared to siNT-treated cells.

knockdown of PGM2 significantly blocks the uridine-mediated rescue of metabolic activity following glucose deprivation (Extended Data Figure 4j, Reviewer Figure 6f).

5) PRPS1/2 enzymes use ribose 5-P to produce PRPP, a substrate for NAD and nucleotide synthesis. Therefore, measurements of PRPP levels derived from uridine catabolism should be measured.

Response: To address this point, we measured the contribution of ribose-labeled $^{13}\text{C}_5$ -uridine-derived carbon into PRPP and NAD⁺ pools in two pancreatic cancer cell lines by LC/MS (Figure 2i, Reviewer Figure 7). We observed that the M+5 isotopologue of PRPP was predominant, illustrating that uridine-derived ribose was routed through PRPP for de novo nucleotide and NAD biosynthesis. M+5 and M+10 isotopologues of NAD were also a significant fraction of pool sizes.

6) Measurements of glucose concentration in the TIF could be helpful. Is glucose limiting in the TME?

Response: We agree with the referee that knowledge of the glucose concentration in TIF in our tumor models is important to understand and model uridine metabolism. To this end, we measured glucose levels in the TIF from our murine pancreatic cancer models and found it to be ~8-fold lower than that in the plasma, ~0.5 mM versus ~4 mM (Figure 2h; Reviewer Figure 8).

What is the concentration of glucose that renders PDA cells dependent on uridine? A titration of glucose along with uridine catabolism measurement would be informative.

Response: We approached this question using two parallel approaches, in which we applied concentrations of glucose and uridine that approximate those we measured *in vivo* (Figure 2h; Reviewer Figures 4a and 8).

Reviewer Figure 7. $^{13}\text{C}_5$ -Uridine (ribose labeled) tracing at 1 or 0.1 mM showing the fractional labelling of PRPP and NAD⁺ in human PDA cell lines ASPC1 and TU8988S grown in either 5 or 0.1 mM glucose. PRPP, phosphoribosyl-pyrophosphate.

Reviewer Figure 8. Glucose quantification in the plasma and tumor intestinal fluid (TIF) from mice bearing pancreatic orthotopic allograft tumors.

First, we cultured pancreatic cancer cells in 5 mM or 0.1 mM glucose and 1 mM or 0.1 mM ribose-labeled $^{13}\text{C}_5$ -uridine and followed the uridine-derived carbon into central carbon metabolism in two pancreatic cancer cells by LC/MS-based metabolomics. Here, we observed that at low glucose and high uridine, uridine carbon was the predominant contributor to central carbon metabolism (Figure 2i, Extended Data Figure 3b; Reviewer Figures 7 and 9). This was reversed at high glucose and low uridine, with glucose being

Reviewer Figure 9. $^{13}\text{C}_5$ -Uridine tracing showing the fractional labelling of uridine ribose-derived carbon in glycolytic (G6P, 3PG/2PG, pyruvate, lactate) and PPP intermediates (6PG, S7P), uridine, and TCA cycle metabolites in human PDA cell lines ASPC1 and TU8988S grown in 5 or 1 mM glucose and 1 or 0.1 mM uridine. 3PG/2PG, 3/2-phosphoglycerate; G6P, glucose 6-phosphate; 6-PG, 6-phosphogluconate; S7P, sedoheptulose 7-phosphate.

the predominant source of carbon. At the intermediate concentrations, uridine and glucose carbon contributed to central carbon metabolism in roughly equal proportion. These results support our model that glucose and uridine-derived ribose can function interchangeably, where that which is available at a greater concentration dominates.

Second, we assayed cellular reducing potential (denoted as relative metabolic activity, RMA) across a range of glucose concentrations (0-10 mM) in pancreatic cancer cells supplemented with 0.1 mM uridine (Extend Data Figure 4a-c; **Reviewer Figure 10**). We observed that in culture media containing concentrations up to 1 mM glucose, 0.1 mM uridine enhanced RMA. Such increases in RMA were not observed for 0.1 mM uridine supplementation when glucose was set at 10 mM glucose.

Based on these data, we propose that uridine is a conditionally relevant nutrient to support central carbon metabolism in glucose-restricted areas of the tumor. We also believe the observation that uridine catabolism machinery (i.e. UPP1) is transcriptionally upregulated in response to low glucose (**Reviewer Figure 1**) further supports the notion that uridine is conditionally relevant in PDA. Taken together, these results illustrate that uridine can be a major contributor to central carbon metabolism at concentrations relevant in the pancreatic tumor microenvironment.

Reviewer Figure 10. a-c) Relative metabolic activity (RMA) of PDA cells supplemented with 0.1 mM uridine in various glucose concentrations.

Reviewer 2

In this manuscript by Ward, Nwosu, Poudel and colleagues, the authors analyzed the capacity of diverse nutrients to support PDAC cell metabolism in the presence of sub-physiologic levels of glutamine and the absence of glucose. Their initial screen, together with subsequent metabolomics and stable isotope tracing experiments, demonstrated that uridine can support PDAC cell metabolism under these nutrient-deprived culture conditions. The authors performed a strong characterization of uridine metabolism in PDAC cells including its metabolic fate, and convincingly demonstrated the requirement for UPP1 for uridine-to-uracil conversion and downstream fates of catabolized uridine. A role for extracellular uridine and for UPP1 in PDAC cell metabolism and viability is entirely novel and potentially of great interest. However, the study at present is descriptive, and lacks compelling evidence to support a functional role for uridine/UPP1 in PDAC. The main weakness of the study is the questionable relevance of uridine and UPP1 for pancreatic cancer cell metabolic fitness within an intact tumor. Numerous published studies, including excellent prior work by these authors, support the notion that PDAC cells employ diverse mechanisms to scavenge nutrients from the extracellular space to support key metabolic processes under nutrient-limiting conditions. While the cell culture studies in the manuscript clearly demonstrate that pancreatic cancer cells can take up and utilize uridine to support diverse metabolic processes, whether they do to a meaningful extent *in vivo* remains unclear, where diverse additional metabolites are available as are other cell types which may compete for nutrients. The particular cell culture conditions used throughout the study are quite flawed as detailed below, and only a single experiment was performed *in vivo* which yielded results rather difficult to interpret and was sub-optimally designed. These weaknesses lessen enthusiasm for the study.

Response: We thank the reviewer for acknowledging the strengths of our studies while also highlighting the weaknesses in the *in vivo* and *in vitro* studies. Guided by the referee's constructive criticism, we more precisely determined the glucose and uridine concentrations in tumors *in vivo* and used these to better model and study this process *in vitro* with biochemical and functional assays. In addition, we have now also employed stable isotope tracing *in vivo*, the data from which support our *in vitro* mechanistic work. Finally, we have also made syngeneic UPP1 knockout models of pancreatic cancer. With these, we demonstrate that UPP1 inhibition potently blocks orthotopic pancreatic tumor growth in immunocompetent animals.

Specific comments:

1. In the Biolog assay presented in Figure 1, and in subsequent validation experiments, the authors subject PDAC cells to a total absence of glucose. As glucose concentration in the PDAC microenvironment are greater than 1mM, it seems quite unlikely that cancer cells experience zero glucose in tumors. Where near-physiologic 1mM glucose is used on PDAC cells in Figure 2, it is used on its own, and the ability of uridine to augment relative metabolic activity in this context is not shown. Uridine plus glucose is only shown in Figure 4 in the presence of very high glucose concentrations. Does uridine significantly impact PDAC cell metabolism in the presence of relevant glucose concentrations? If the experiment depicted in

Figure 4b is repeated in the presence of lower glucose concentrations, does uridine impact NADH levels?

Response: We agree that a more detailed analysis of the competing activities of glucose and uridine, at physiological concentrations, was necessary to support the physiological relevance of our findings. Concerns around this theme were also communicated by Reviewer-1 above (point 6), and we reiterate partitions of that reply below.

To test the impact of uridine on pancreatic cancer metabolism, we first quantitated the concentration of glucose and uridine in tumor interstitial fluid from our mouse model (Figure 2h; Reviewer Figures 4a and 8) and employed these concentrations *in vitro* (Figure 2i, Extend Data Figure 3b, 4a-c; Reviewer Figures 7, 9, and 10). We then applied two parallel approaches to determine the range of glucose concentrations for which uridine supplementation impacted pancreatic cancer metabolism.

First, we cultured pancreatic cancer cells in 5 mM or 0.1 mM glucose and 1 mM or 0.1 mM ribose-labeled $^{13}\text{C}_5$ -uridine and followed the uridine-derived carbon into central carbon metabolism in two pancreatic cancer cells by LC/MS-based metabolomics. Here, we observed that at low glucose and high uridine, uridine carbon was the predominant contributor to central carbon metabolism (Figure 2i, Extended Data Figure 3b; Reviewer Figures 7 and 9). This was reversed at high glucose and low uridine, with glucose being the predominant source of carbon. At the intermediate concentrations, uridine and glucose carbon contributed to central carbon metabolism in roughly equal proportion. These results support our model that glucose and uridine-derived ribose can function interchangeably, where that which is available at a greater concentration dominates.

Second, we assayed cellular reducing potential (denoted as relative metabolic activity, RMA) across a range of glucose concentrations (0-10 mM) in pancreatic cancer cells supplemented with 0.1 mM uridine (Extend Data Figure 4a-c; Reviewer Figure 10). We observed that in culture media containing concentrations up to 1 mM glucose, 0.1 mM uridine enhanced RMA. Such increases in RMA were not observed for 0.1 mM uridine supplementation when glucose was set at 10 mM glucose.

Third, to directly assess the relevance of uridine metabolism in pancreatic tumors, we used *in vivo* isotope delivery (ribose labeled $^{13}\text{C}_5$ -uridine) and metabolomics. We found that pancreatic allograft tumors are indeed able to capture and metabolize uridine, even with a short 1-hour pulse (Figure 2g, Extended Data Figure 3a; Reviewer Figure 5), in a manner akin to that observed *in vitro* (Figure 2d-f,i and Extended Data Figure 2k, 3b).

Based on these data, we propose that uridine is a conditionally relevant nutrient to support central carbon metabolism in glucose-restricted areas of the tumor. We also believe the observation that uridine catabolism machinery (i.e. UPP1) is transcriptionally upregulated in response to low glucose (Reviewer Figure 1) further supports the notion that uridine is conditionally relevant in PDA. Taken together, these results illustrate that uridine can be a major contributor to central carbon metabolism at concentrations relevant in the pancreatic tumor microenvironment.

2. The correlative data presented in Figure 1, ED Figure 3, and the ED Tables with respect to UPP1 in PDAC are interesting, but certainly do not support the conclusion (line 119) that “UPP1 is a critical metabolic enzyme in PDA.” This statement should be amended.

Response: We agree with the referee that our previous data only provided correlative support for UPP1 in pancreatic cancer metabolism. In this revised study, guided by referee’s suggestions, we have generated two independent *Upp1* knockout syngeneic murine tumor models. Analysis of orthotopic tumor growth of these cells in immunocompetent hosts illustrated that *Upp1* knockout profoundly repressed tumor growth (Figure 4i-k, Extended Data Figure 10e; Reviewer Figure 11 below). In addition, we also now present a wealth of new *in vitro* metabolic data to support the role of UPP1 and uridine catabolism as a major input into central carbon metabolism, particularly under low glucose conditions (Figure 2i, Extended Data Figure 3b; Reviewer Figures 7 and 9). Finally, we also provide *in vivo* isotope tracing metabolomics data, which support our model that uridine can be captured and utilized to support central carbon metabolism in pancreatic tumors (Figure 2g, Extended Data Figure 3a; Reviewer Figure 5).

Together, these new data illustrate the importance of UPP1 and uridine in pancreatic cancer metabolism. That said, we appreciate the essence of this reviewer’s comment, and we have revised the current manuscript to indicate the role of UPP1 as “important”, rather than “critical” in the abstract and discussion.

3. Uridine is used at 1mM throughout the manuscript, but its levels in PDAC interstitial fluid are approximately 20-fold lower than that, so it is hard to know whether the metabolic assays performed *in vitro* are relevant. Key experiments should be repeated in the presence of an appropriate uridine concentration.

Response: We agree with the referee on this extremely important point. In the response to Major Point 1 above, we detail our efforts to quantitate and apply relevant *in vivo* uridine concentrations in our *in vitro* model systems. These new data are presented in Figure 2h,i, Extended Data Figure 3b, 4a-c, and Reviewers Figures 4a, 7-10.

4. The tumor growth experiment shown in Figure 4i needs further validation in light of the variability between UPP1 knockout clones, as it is unconvincing at present that UPP1 indeed supports PDAC growth in mice. UPP1 should be restored in clone 1A to confirm specificity, or an additional clone should be tested. Further, in light of the authors’ proposed model in Figure 4j implicating the tumor microenvironment, the role of UPP1 in PDAC growth should be tested in the relevant tissue environment (pancreas) instead of subQ. As Figs 4h and ED Fig 4f,g show UPP1 expression in adaptive immune cells, competition for uridine between cancer cells and tumor-infiltrating leukocytes may indeed be relevant, so UPP1-null PDAC cells should be tested in an immune-competent, syngeneic model.

Response: We thank the reviewer for highlighting issues with our selection of mouse models, as well as the experimental suggestions. We agree with these important points, and we have refocused our *in vivo* experimental approaches in this revised study.

Namely, we moved our *in vivo* analyses to syngeneic, orthotopic models of pancreatic cancer. These were employed to profile metabolite abundance in TIF (Figure 2h; Reviewer Figures 4a and 8), deplete tumor associated macrophages to determine their impact on intratumoral uridine availability (Figure 4a-c; Reviewer Figure 4), follow uridine metabolism via *in vivo* delivery of isotopically labeled uridine (Reviewer Figure 5), and, most importantly, to inhibit UPP1 and assess the impact on tumor growth (Reviewer Figure 11).

To specifically address the comment on tumor growth, we generated two independent models of *Upp1* knockout in the syngeneic murine pancreatic cancer cell lines MT3-2D and 7940b. Two *sgUpp1* constructs (sg1, sg3) were employed and compared to a vector control (sgV). Both murine pancreatic cancer cells lines are of a B6 background generated from the KPC model (p48-Cre; LSL-Kras^{G12D}; LSL-P53^{R172H}). Similar to our *in vitro* human cell line models, the knockout of *Upp1* in these murine cell lines suppressed their ability to use uridine to support metabolism (Figure 4d-f; Reviewer Figure 11a-c). We implanted these lines into the pancreas of syngeneic hosts (orthotopic allograft) and assessed tumor weight and volume at endpoint, as well as histological features. As illustrated in Reviewer Figure 11d-g, these two cell lines show a consistent and significant reduction in tumor growth upon UPP1 knockout. In fact, the defect in tumor growth was also observed in an immunocompetent, subcutaneous model (Figure 4k; Reviewer Figure 11h).

Reviewer Figure 11. a). Uridine utilization in murine PDA KPC cell lines MT3-2D and 7940b upon CRISPR/Cas9 knockout of UPP1 (sg1 and sg3) relative to sgVector (sgV) controls. b) Metabolomic profiling of uridine and uracil in UPP1 knockout clones c) and other related pathways, as compared with human UPP1 knockout lines TU8988S and ASPC1. d) Schematic of orthotopic implantation of MT3-2D. Tumors are pictured below and e) quantitated. f) MT3-2D UPP1 knockout tumors exhibit increased uridine and decreased uracil, consistent with a defect in uridine metabolism. g) Endpoint tumor weight and pictures for KPC 7940b cells grown in the pancreas of immunocompetent mice. h) Endpoint tumor weight and pictures for subcutaneous, syngeneic MT3-2D tumors grown in immunocompetent mice.

Histological analysis revealed that the UPP1 knockout tumors exhibited a trend toward more anti-tumor T cell infiltration (CD8 T cells), less tumor vascularity, and no changes in macrophage content (Extended Data Figure 10h; Reviewer Figure 12a-c). In addition, metabolomics profiling of tumors at endpoint revealed a >2-4-fold accumulation of uridine and 2-fold drop in uracil in the UPP1 knockouts (Figure 4l, Extended Data Figure 10e; Reviewer Figure 11f). We believe that it is also important to conclude by noting that these *in vivo* tumor data contrast an absence in proliferative defects of the same cell lines *in vitro* (Extended Data Figure 10d; Reviewer Figure 12d,e).

In total, our new data indicate an important role for UPP1 in pancreatic tumor growth. The contrasting data between *in vitro* and *in vivo* phenotypes further highlight the role of nutrient availability, aspects of the tumor microenvironment, and/or immune function influencing the function and necessity of UPP1 *in vivo*.

Reviewer Figure 12. a) Quantitation of histological staining in sections from UPP1 KO (sg1, sg3) and sgVector (sgV) MT3-2D pancreatic tumor allografts for macrophages (F4/80), b) vascularity (CD31), and c) CD8 T cells. d) Proliferation curves for MT3-2D and e) KPC 7940b sgV and UPP1 KO (sg1, sg3) cells.

5. (Minor) It seems the y-axis is mislabeled in the left graph in Figure 4i (presumably tumor weights are shown in grams, not milligrams).

Response: Thank you for the comment. In the revised manuscript, based on the excellent referee feedback, we shifted our focus to immunocompetent murine models that better recapitulate the complex pancreatic TME. As such, the figure referred to in this comment is no longer included in our resubmission. However, we have ensured appropriate labeling of our tumor plots in subsequent experiments.

Reviewer 3

Ward et al. performed experiments to determine which of an array of metabolites induce dye reduction in otherwise nutrient-limited pancreatic ductal adenocarcinoma (PDA) cells. They found that uridine could be extensively metabolized in these cells, and that its ribose backbone fuels glycolysis and the TCA cycle. The enzyme UPP1 is required for cells to use uridine to support cell growth when glucose is unavailable. Gene expression profiling indicates that UPP1 is modestly up-regulated in PDA tumors, and that its expression correlates with poor outcomes in most cohorts. Tracing experiments with isotope-labeled uridine demonstrate that the ribose ring enters glycolysis and ultimately feeds bioenergetic and biosynthetic pathways. Knocking out UPP1 in one PDAC line reduces xenograft growth (although inconsistently between two clones). The underlying finding – that uridine can be a meaningful source of carbon from central metabolism – is interesting and could be relevant in tumors where more conventional nutrients like glucose are scarce. However, there are numerous issues in the paper that need to be addressed.

Response: We thank the referee for their careful reading of our work, enthusiasm, and supportive feedback, including a number of important suggestions that have considerably strengthened our conclusions.

1. Uridine stands out in the Biolog assay as one of the most commonly metabolized nutrients among the PDA cells, but not by HPNE and HPSC cells. Given the fact that uridine metabolism correlates with glucose metabolism, the authors should examine a few non-PDA cell lines for uridine-dependent NADH production.

Response: We thank the reviewer for this suggestion. The data from our initial Biolog screening assay indicated that immortalized pancreatic epithelial cells (HPNE) do metabolize uridine, while human pancreatic fibroblasts (HPSC) are not similarly able (Extended Data Figure 1f; Reviewer Figure 13a). Based on the referee's inquiry, we extended this analysis to a panel of cancer cell lines, including lung cancer, fibrosarcoma, osteosarcoma, and colon cancer. Like pancreatic cancer cells, we find that

Reviewer Figure 13. a). Biolog relative metabolic activity (RMA) screening data in the pancreatic fibroblast line HPSC and the immortalized pancreatic epithelial cell line HPNE plus or minus uridine. b) RMA in A549 (lung), c) HT1080 (fibrosarcoma), d) U2OS (osteosarcoma cell line), and e) HCT116 (colon) in 1 mM glucose-containing media mock-treated or incubated with 0.1 or 1 mM uridine.

these cell lines can all utilize uridine to support bioenergetics in glucose limiting conditions (Extended Data Figure 6f; Reviewer Figure 13b-e). Finally, we would also like to take this opportunity to highlight a co-submitted manuscript from the Mootha lab, which demonstrated varying uridine catabolism in the PRISM library of 500 cancer cell lines, as well as in normal mouse liver (Jourdain, et al. *bioRxiv* 2021). Collectively, we conclude that uridine can be utilized by a wide array of cell lines from varying tissues of origin.

2. There are a few concerns about the UPP1 expression data. Although UPP1 expression is discussed throughout the paper, the authors never validate UPP1 levels with a western blot. Protein expression should be correlated with uridine-dependent RMA (as shown for RNA levels in Fig. 1d), and to validate UPP1 loss in CRISPR KO cells.

Response: We absolutely agree with the referee on this point, and we are pleased to relay that we were able to identify and validate an anti-human UPP1 antibody during this revision period. This is now used throughout the paper, alongside qPCR, to assess 1) UPP1 expression in human pancreatic cancer cell lines (Figure 1e; Reviewer Figure 14a), which we correlate with uridine-dependent RMA (Figure 1c,f; Reviewer Figure 14b); 2) UPP1 knockout in our human pancreatic cancer models (Figure 3a; Reviewer Figure 14c,d); 3) changes in UPP1 expression following MAPK pathway inhibition and following

Reviewer Figure 14. **a.** UPP1 immunoblot in a panel of human PDA cell lines. Vinculin serves as the protein loading control. **b.** Correlation of UPP1 expression and relative metabolic activity (RMA) of PDA cell lines exposed to 1 mM uridine. **c.** UPP1 immunoblot in two TU8988S UPP1 knockout lines and **d.** two ASPC1 UPP1 knockout cell lines (1A, 1B), as compared to wild type lines. Vinculin serves as the protein loading control. **e.** Relative extracellular and intracellular abundance of uridine and uracil in wild type and UPP1 KO human ASPC1 (1A, 1B) and **f.** mouse UPP1 KO (sg1, sg3) MT3-2D cell lines *in vitro* relative to vector control lines (sgVector).

nutrient limitation (Figure 3o, Extended Data Figure 8c,h; Reviewer Figure 1); and 4) UPP1 expression in human pancreatic tumor sections (Figure 3h,i, Extended Data Figure 7; Reviewer Figure 15). However, neither this antibody, nor any other commercially available antibody, has proven effective for lysates or tissues from mice. As such, we have continued to rely on qPCR and metabolic activity +/- uridine as surrogate measurements. For example, similar to the human UPP1 KO cells (Figure 3b,c), our mouse cell lines display an inability to use uridine upon UPP1 knockout (Figure 4d,e; Reviewer Figure 11a). And, again, the like the human UPP1 KO lines (Figure 3d, Extended Data Figure 5a,b; Reviewer Figure 14e), the mouse cell lines also accumulate uridine and have lower levels of uracil both *in vitro* (Figure 4e; Reviewer Figure 14f) and *in vivo* following UPP1 knockout (Figure 4l, Extended Data Figure 10e; Reviewer Figures 11f).

3. The expression data from human PDA (Fig. 1g) are not convincing. Protein expression or IHC might really help here, because the levels of mRNA levels are barely different between tumor and non-tumor samples. Is nonmalignant pancreas unusually high among the organs in UPP1 expression?

Response: We thank the reviewer for the opportunity to address this important inquiry on UPP1 expression in tumors versus normal tissue. First, in an unrelated study from the lab, we comprehensively analyzed the expression of metabolic genes in tumors versus adjacent normal tissue (Nwosu et al. *bioRxiv* 2021). We found *UPP1* to be among the most consistently upregulated genes in PDA and among the best predictors of worse outcome (Reviewer Figure 15a). Next, we turned to the Human Protein Atlas (HPA) data, as extracted from the NCBI portal (<https://www.ncbi.nlm.nih.gov/gene/7378>) and the HPA website (<https://www.proteinatlas.org/ENSG00000183696-UPP1/tissue>), which demonstrate that normal pancreas is one of the lowest UPP1 expressing tissues, at the RNA and

Reviewer Figure 15. a) Top genes whose expression predicts better (green) and worse (red) overall survival outcome in pancreatic cancer from a comprehensive analysis of publicly available pancreatic cancer gene sets (Nwosu Z, et al. *bioRxiv* 2021). b) *UPP1* expression data from the NCBI or c) *UPP1* expression data from the human protein atlas (HPA) in normal human tissues. Normal pancreas expression is highlighted. d) Tissue microarray showing high *UPP1* expression in PDA compared to adjacent normal pancreas (data from HPA). e) Immunofluorescent (IF) analysis of *UPP1* by RNAscope (yellow) in a representative PDA section relative to adjacent (Adj) normal tissue from the same surgical resection. Nuclei are stained in blue; epithelial cells by pan-cytokeratin (panCK) in purple. f) Quantitated *UPP1* expression by RNAscope from three patient samples. g) *UPP1* expression in patient samples. h,i) Uniform Manifold Approximation and Projection (UMAP) showing cell types from the single-cell RNA sequencing data. j) UMAP showing *UPP1* expression in two human PDA tumors (#1238 and #1302) from single-cell RNA sequencing data. The depth of the red color reflects the degree of *UPP1* expression, which is evident in the tumor epithelial and TME populations, notably myeloid cells. k) *UPP1* expression in human PDA relative to normal pancreas (left, center) and tumor relative to liver metastasis (right). Data extracted from the indicated datasets.

protein levels (Extended Data Figure 6b; **Reviewer Figure 15b,c**). We have also included here an immunohistochemistry (IHC) image extracted from the HPA portal, which illustrates high *UPP1* expression in PDA, relative to normal pancreas (Extended Data Figure 6c; **Reviewer Figure 15d**). As independent validation, we assayed *UPP1* by RNAscope, *UPP1* by IHC, and *UPP1* levels by single cell RNA sequencing on patient samples from the University of Michigan (Figure 3h,i, Extended Data Figure 7a-d; **Reviewer Figures 15e-j**). These data are consistent with those in the public datasets, further illustrating high, variable *UPP1* expression in human PDA relative to normal adjacent tissue extracted in surgery (Figure 3g; **Reviewer Figure 15k**).

4. Similarly, the evidence that *UPP1* is driven by *KRAS* (Fig. 1f, ED 3d) is not convincing. Further insights into the mechanism of *UPP1* expression would be helpful.

Response: We agree with the referee that further mechanistic work was required to connect *KRAS* to *UPP1*, and we appreciate the opportunity to build out what we now believe to be an important mechanism of *UPP1* regulation in pancreatic cancer. This concern was also noted by Reviewer 1 (Major Comment 1). The response below reiterates our observations.

In our initial submission we included transcriptome profiling data from our mouse model with doxycycline-inducible mutant *Kras* expression. These data indicated that mutant *Kras* induction in murine PDA cell lines and tumors correlated with increased *Upp1* (Extended Data Figure 8b). To test this directly, we employed our inducible mutant *Kras* cell lines *in vitro*, confirming that mutant *Kras* activation significantly increases *Upp1* expression (Figure 3l,m; **Reviewer Figure 1a**). In pancreatic cancer, mutant *Kras* signals through the MAPK pathway to support tumor metabolism (Ying, et al. *Cell* 2012; Perera RM, Bardeesy N. *Can Disc* 2015; Santana-Codina, et al. *Nat Comm* 2018). Thus, we then employed inhibitors of MAPK signaling to determine the impact on *UPP1* expression. These studies revealed that the regulation of *UPP1* was downstream of MAPK in multiple human and mouse pancreatic cancer cell lines, at both the mRNA and protein expression levels. This closely correlated with pERK expression and response to the MEK inhibitor

trametinib (Figure 3n,o, Extended Data Figure 8c-l; Reviewer Figure 1b-h). Lastly, we also now include TCGA data for pancreatic tumors expressing no alterations in KRAS or G12D mutation, where we also find significantly higher *UPP1* expression in the *KRAS* G12D mutated tumors (Figure 3k; Reviewer Figure 1i).

Further on the mechanistic regulation of *UPP1*, we also now demonstrate that low uridine or low glucose concentrations similarly promote *UPP1* expression (Figure 3n,o, Extended Data Figure 8c-l; Reviewer Figure 1d-h). This makes mechanistic sense, in that when nutrient availability is decreased, *UPP1* expression is up-regulated to support enhanced ribose scavenging from uridine. This nutrient regulation of *UPP1* appears to function as more a rheostat, enhancing the expression primarily mediated by MAPK signaling.

In sum, our data illustrate that *UPP1* is regulated downstream of mutant *KRAS*-mediated MAPK signaling in pancreatic cancer and that this signaling axis is further augmented by nutrient (glucose or uridine) deprivation.

Does *UPP1* expression also correlate with oncogenic *KRAS* in other types of cancer (public databases like TCGA and CCLE would be helpful here)?

Response: As suggested, we first assessed the TCGA data of mutant and wildtype *KRAS*, where we find significantly higher *UPP1* expression in the *KRAS* G12D mutated pancreatic tumors (Reviewer Figure 1i). Next, we performed the analysis of CCLE data using *KRAS* wildtype vs. mutant cell lines. We associated this genotype with *UPP1* protein expression across the entire dataset (pan-cancer), and then in a targeted manner for lung and colorectal cancer. A targeted analysis was not performed for pancreatic cancer cell lines because *Kras* mutations are observed in all but one pancreatic cancer cell line in the CCLE.

From the pan-analysis (n=374), we observed an association between *KRAS* status and *UPP1* expression, with borderline statistical significance (p=0.09) (Extended Data Figure 8a; Reviewer Figure 16a). Similarly, lung cancer cell lines (n=79) with mutant *KRAS* showed significantly (p=0.003) increased *UPP1* expression compared with wild-type *KRAS* lines (Extended Data Figure 8a; Reviewer Figure 16b). On the other hand, colorectal cancer lines (n=30) showed slightly reduced *UPP1* in mutant *KRAS* lines (Extended Data Figure 8a; Reviewer Figure 16c).

Similar to the correlation with *KRAS* status, pancreatic and lung cancers had higher *UPP1* expression than normal tissue, and high *UPP1* expression predicted worse overall survival (Figure 3j, Extended Data Figures 6d,e, and 7e; Reviewer Figure 15k and 16d-f). Again, consistent with the *KRAS* status, colon tumors had lower *UPP1* expression than normal colon (Extended Data Figure 6d; Reviewer Figure 16g). The data for *KRAS* and *UPP1* in colon cancer are not altogether unexpected, as *KRAS* mutations are not driver mutations in this disease, and the impact of *KRAS* on colorectal tumor biology and metabolism is different from that of lung and pancreatic cancer (Kerk, et al. *Nature Reviews Cancer* 2022).

Reviewer Figure 16. a). UPP1 expression stratified by mutant *Kras* status in all (pan-cancer), b) lung cancer, and c) colon cancer cell lines from the cancer cell line encyclopedia (CCLE). d) UPP1 expression in lung tumors relative to normal tissue. e) High UPP1 expression is predictive of worse overall survival in lung and f) in 3 out of 4 pancreatic cancer datasets. g) UPP1 expression in colon tumors relative to normal tissue.

Does suppressing *KRAS* expression also reduce uridine's ability to sustain metabolism in the absence of glucose?

Response: Several of our data suggest that suppressing *KRAS* reduces the ability of uridine to sustain metabolism, including but not exclusively when glucose is lacking. First, as mentioned above, we have demonstrated that extinguishing mutant *Kras* expression in murine pancreatic cancer cells or treatment with the MAPK inhibitor trametinib in murine and human pancreatic cancer cells suppresses *Upp1* expression (Figure 3l-o, Extended Data Figure 8c-i; Reviewer Figure 1). Second, we performed metabolomics on trametinib-treated pancreatic cancer cells and observed an increase in uridine and decrease in uracil, coincident with the decrease in UPP1 expression, reflective of decreased uridine catabolism (Extended Data Figure 8j, Reviewer Figure 17). Third, we also tested the ability of trametinib to block the uridine-mediated rescue of proliferation upon glucose limitation. We find that indeed the ability of the cells to benefit from uridine is restricted following trametinib treatment (Figure 3p, Reviewer Figure 17b,c). Taken together, our data support that *KRAS*-driven MAPK signaling drives uridine metabolism in PDA.

5. The informatics analyses need to be presented more clearly. UPP1-expressing cells display “downregulation of metabolic pathways,” but this is too superficial for a paper about metabolism. It provides no information about which pathways are co-regulated with UPP1.

Response: We appreciate the referee's feedback, and in this revised manuscript we have provided a more detailed informatics analysis of the metabolic pathways co-regulated

Reviewer Figure 17. a) Heatmap of metabolomics profiling for intracellular metabolite changes in ASPC1 cells +/- trametinib treatment plus uridine minus glucose (left) or plus uridine and glucose (right). b) Relative proliferation in TU8988S, DANG, or ASPC1 cells +/- trametinib and uridine minus glucose or c) plus glucose.

with UPP1. First, we have amended the statement ‘downregulation of metabolic pathways’ to read “*UPP1-high cell lines, as well as UPP1-high patient tumors, displayed a profound downregulation of metabolic pathways (Extended Data Fig. 2b,c), notably the downregulation of amino acid-, fatty acid- and glutathione metabolism, altogether indicating widespread metabolic alterations.*” (Reviewer Figure 18a). Second, regarding co-upregulated pathways, we identified a strong correlation between high UPP1 expression and the metabolic signature of glycolysis and highlight this in our revised study (Extended Data Figure 2a, Reviewer Figure 18b,c).

The authors highlight immune-related genes among the DEGs “potentially link UPP1 to TME activities.” This is potentially interesting, but not informative as written.

Reviewer Figure 18. a) Gene set enrichment analysis (GSEA) in UPP1 high vs low tumors. b) Heatmap showing the upregulation of glycolytic genes and c) the GSEA illustrating enriched glycolysis in UPP1 high tumors.

Response: Here too we appreciate the referee's careful reading of our work and constructive criticism. In our revised study, we provide a more informative description of the relationship between UPP1 and immune-related genes. The revised statement now reads: *"The upregulated genes in UPP1-high cell lines included those involved in endocytosis and several inflammation/immune-related pathways, notably NFkB signaling (Supplementary Table 4)..."*

Can the authors better link this finding to the data later in the

paper showing that uridine is released by polarized macrophages? The implication is that macrophage-conditioned medium – or at least medium containing the same concentration of uridine as released by macrophages – can sustain PDA cell metabolism in glucose-deprived conditions, in a UPP1-dependent manner.

Response: We agree with the referee that this is an important point and thank them for the suggestion. Indeed, this was also a comment brought up by Review 1 (Major Point 2), the answer for which we reiterate in part below.

In our previous submission, we provided data illustrating that bone marrow-derived macrophages, polarized *in vitro* with conditioned media from pancreatic cancer cells, release micromolar levels of uridine (Extended Data Figure 10a). Accordingly, we had previously put forth that they may be a major source of tumoral uridine. To test this directly, we depleted tumor associated macrophages (TAMs) from murine allograft tumors using the combination of CSF1 antibody and clodronate (liposome) (Figure 4a, Reviewer Figure 4b). As reported previously, this combination depleted TAMs by ~50% and suppressed orthotopic tumor growth (Zhang, et al. *Gut* 2017; Candido, et al. *Cell Reports* 2018) (Figure 4a,b; Reviewer Figure 4c,d). As it relates to uridine and uracil, we observed a reduction in plasma uridine level by ~8 fold upon macrophage depletion. Uracil levels in the serum were not altered (Figure 4c, Extended Data Figure 10b; Reviewer Figure 4e,f). This marked impact on plasma uridine levels following macrophage depletion indicates that macrophages may be major mediators of uridine production/release. However, despite the considerable impact on plasma uridine, tumor and interstitial fluid uridine was not changed (Figure 4c, Reviewer Figure 4g,h).

These *in vivo* tumor data contrast our *in vitro* models, which indicated that anti-inflammatory TAMs release micromolar uridine (Extended Data Figure 10a). However, there is considerable precedent for compensatory activities from other cell types in the TME when one cell type is depleted. For example, an increase in immunosuppressive myeloid subtypes were reported when Tregs are depleted from pancreatic tumors (Zhang, et al. *Cancer Discovery* 2020), and there is competing activity of deoxycytidine release from TAMs and CAFs (Halbrook, et al. *Cell Metabolism* 2019; Dalin, et al. *Cancer Research* 2019). Thus, it is conceivable that uridine levels were maintained from a compensatory cell type in our macrophage depletion model.

In a parallel experiment, we administered isotope-labeled uridine to pancreatic tumor bearing animals to study uridine metabolism *in vivo*. Animals were sacrificed one hour after injection, and tumor tissue was collected to measure uridine uptake and utilization (Figure 2g, Extended Data Figure 3a; Reviewer Figure 5). Consistent with our *in vitro* studies, and despite the short experimental duration, we observed contribution of uridine-derived ribose to nucleotides, glycolytic and branching pathways, and TCA cycle metabolites. In addition to demonstrating that exogenous uridine is readily captured and metabolized to fuel central carbon metabolism, these data also indicate that uridine in circulation is used by the tumor. In other words, while we had previously suggested that TAMs were a principal source of uridine, these data indicate that uridine is likely coming from multiple sources, including distant sources through circulation (e.g. liver) as well as locally from the TME.

In sum, based on these new data, we have modified our conclusions to acknowledge that while macrophages may be a source of uridine, other cell types also contribute to the TME and systemic uridine pools.

6. The isotope labeling experiments are generally informative, but how do TU8988S cells manage to label their TCA cycle intermediates so well despite having almost no labeling in pyruvate? This finding does not fit the proposed mechanism.

Response: We thank the reviewer for their careful reading of our work. A re-analysis of the data in question revealed that the labeled pyruvate peak was at the level of noise for our instrument, which provided an artificially low value. We have repeated this isotope tracing experiment in TU8988S two additional times. We observe a clear labeling of pyruvate (notably M+3), the fractional labeling for which is strongly enhanced when glucose is restricted, consistent with our model (Extended Data Figure 3b). We also see again a strong labelling of the TCA cycle metabolite citrate supporting the passage of the labeled fractions via glycolysis to TCA cycle (Figure 2i; Extended Data Figure 5g). The accompanying data are displayed in Reviewer Figure 9.

7. The authors comment that uridine is more abundant in PDA TIF than in blood. But the experiments that generated the dataset queried by the authors used a PDA GEMM, and one of the conclusions of that paper was that the PDA TIF contains millimolar levels of glucose, rather than being severely glucose depleted as commonly thought. This makes it unclear how the data

from these TIF experiments relate to the pathway proposed here by the authors, where uridine drives metabolism in the absence of glucose. It would be more relevant to assess glucose and uridine levels in the xenograft models being used in this paper, since the authors argue that UPP1 is required for growth of those models.

Response: We thank the referee for their careful attention to experimental detail and the important suggestions. Guided by these suggestions, we moved our *in vivo* tumor studies to orthotopic, immunocompetent models (more on model selection in Response 8 below). Using this model, we collected tumor interstitial fluid (TIF) and measured the uridine and glucose concentrations. Glucose was found to be in the low mM range (Figure 2h; Reviewer Figure 8), and uridine was present in the micromolar range (Figure 2h; Reviewer Figure 4a), broadly consistent with previous publications (Sullivan, et al. *eLife* 2019). These glucose and uridine concentrations were then employed in many of our *in vitro* studies to more accurately model and study the competition between these nutrient inputs (Figure 2i, Extended Data Figure 3b, 4a-c; Reviewer Figures 9 and 10). For more on this, please also see our response to Referee 1, Major Point 6 (pages 10-12).

8. The evidence that this pathway is important *in vivo* is limited. In Figure 4i, two clones display disparate growth rates in mice, and we do not know whether residual levels of UPP1 differ between them. The authors need to address this issue and show that UPP1 dependence extends to other PDA tumor models that express UPP1. It would also be good to know whether UPP1 loss increases uridine levels in the tumors.

Response: We agree with the referee that this is an important question and thank them for the suggestion. Indeed, this was also a comment brought up by Review 2 (Major Point 4), the answer for which we reiterate in part below.

As noted above, we moved our *in vivo* analyses to syngeneic, orthotopic models of pancreatic cancer. These were employed to profile metabolite abundance in TIF (Figure 2h; Reviewer Figures 4a and 8), deplete tumor associated macrophages to determine their impact on intratumoral uridine availability (Figure 4a-c; Reviewer Figure 4), follow uridine metabolism via *in vivo* delivery of isotopically labeled uridine (Reviewer Figure 5), and, most importantly, to inhibit UPP1 and assess the impact on tumor growth (Reviewer Figure 11).

To specifically address the comment on tumor growth, we generated two independent models of *Upp1* knockout in the syngeneic murine pancreatic cancer cell lines MT3-2D and 7940b. Two *sgUpp1* constructs (sg1, sg3) were employed and compared to a vector control (sgV). Both murine pancreatic cancer cell lines are of a B6 background generated from the KPC model (p48-Cre; LSL-Kras^{G12D}; LSL-P53^{R172H}). Similar to our *in vitro* human cell line models, the knockout of *Upp1* in these murine cell lines suppressed their ability to use uridine to support metabolism (Figure 4d-f; Reviewer Figure 11a-c). We implanted these lines into the pancreas of syngeneic hosts (orthotopic allograft) and assessed tumor weight and volume at endpoint, as well as histological features. As illustrated in Reviewer Figure 11d-g, these two cell lines show a consistent and significant reduction in tumor growth upon UPP1 knockout. In fact, the defect in tumor

growth was also observed in an immunocompetent, subcutaneous model (Figure 4k; Reviewer Figure 11h).

Histological analysis revealed that the UPP1 knockout tumors exhibited a trend toward more anti-tumor T cell infiltration (CD8 T cells), less tumor vascularity, and no changes in macrophage content (Extended Data Figure 10h; Reviewer Figure 12a-c). In addition, metabolomics profiling of tumors at endpoint revealed a >2-4-fold accumulation of uridine and 2-fold drop in uracil in the UPP1 knockouts (Figure 4l, Extended Data Figure 10e; Reviewer Figure 11f). We believe that it is also important to conclude by noting that these *in vivo* tumor data contrast an absence in proliferative defects of the same cell lines *in vitro* (Extended Data Figure 10d; Reviewer Figure 12d,e).

In total, our new data indicate an important role for UPP1 in pancreatic tumor growth. The contrasting data between *in vitro* and *in vivo* phenotypes further highlight the role of nutrient availability, aspects of the tumor microenvironment, and/or immune function influencing the function and necessity of UPP1 *in vivo*.

Minor concerns/questions:

1. It is curious that uridine supplementation reverses UPP1 induction, since UPP1 is required to catabolize uridine. Can the authors explain this?

Response: The reviewer points out an interesting aspect of UPP1 regulation. In this revised work, we have gained a much better understanding of UPP1 regulation in pancreatic cancer. First, we find that UPP1 is under the control of the Kras-MAPK signaling pathway, and our data suggest that this the predominant mechanism by which UPP1 is regulated (Reviewer Figure 1).

More to the question at hand, our data have also revealed that nutrient regulation (i.e. glucose or uridine) serves as a rheostat to fine-tune UPP1 expression (Reviewer Figure 1). With regard to this latter aspect, we propose here that nutrient-regulation acts according to a classical feedback inhibition model. When glucose is readily available, UPP1 expression is down because classical glucose metabolism predominates. When glucose drops, UPP1 expression is activated to catabolize uridine, and this UPP1 expression is further activated when uridine is low to promote the scavenging of uridine-derived ribose to fuel central carbon metabolism. There are undoubtedly important kinetic aspects to this model, which would more accurately address the referee's inquiry. We appreciate the insightful comment and enthusiasm and look forward to a further mechanistic dive into this pathway as part of a future study.

2. Line 210: Change "uridine-derived glucose" to "uridine-derived ribose"

Response: This has been updated.

Reviewer Reports on the First Revision:

Referee expertise:

Referee #1: nucleotide metabolism

Referee #2: cancer metabolism/molecular metabolism

Referee #3: PDAC

Referees' comments:

Referee #1 (Remarks to the Author):

The authors have addressed my concerns. I commend them for the great work they have done. The regulation of UPP1 signaling by KRAS adds an interesting layer of complexity that may be valuable in designing therapeutic strategies for targeting UPP1 and KRAS signaling in PDAC.

The two studies (Nwosu, Ward et al. and Jourdain et al.,) complement one another and come to the same conclusions. However, although Nwosu, Ward et al., show that UPP1 is under the control of KRAS, Jourdain et al., state that UPP1 is constitutively expressed. The authors are expected to clarify this divergence.

This article presents robust data and reveals an unappreciated role of circulating uridine in maintaining metabolism under glucose-limiting conditions in PDAC.

Minor comments:

Extended Data Fig.8i. Densitometric quantification of UPP1 protein signal from ASPC1 cells should accompany p-ERK signal.

Referee #2 (Remarks to the Author):

This revision is improved from the original submission, with more detail and better validation of the importance of UPP1 in PDAC xenograft growth. The paper still does not make a convincing case that uridine is a major carbon source *in vivo*, however. There are a few concerns along these lines:

1. The authors quantify uridine and glucose in TIF from their models, but they don't use these concentrations in the culture experiments. According to the rebuttal (Fig. 4), TIF contains about 25 micromolar uridine and 500 micromolar glucose. But the experiments designed to assess uridine metabolism do not use these conditions; the most relevant concentrations used in the paper are 100 micromolar each of glucose and uridine, but these are skewed to favor uridine metabolism. During culture, the concentration of glucose may decline rapidly, further favoring use of carbon from uridine.

2. With this in mind, the data from the *in vivo* tracing experiment give the impression that the pathway is not very active. The delivery method (ip injection or injection directly into the tumors) labels about 30% of the uridine pool, but most metabolites display much less labeling, more than 10-

fold less in some cases. The authors ascribe this to the short labeling period (1 hour), but it's equally plausible that the pathway is just not very active.

3. The effect of UPP1 knockout on tumor growth is impressive and certainly helps the paper, but given the apparently low activity of uridine catabolism in vivo, how sure can the authors be that the mechanism by which UPP1 supports tumor growth is by providing carbon for central metabolism? UPP1 is a bidirectional enzyme so it seems likely that the mechanism could be different.

Referee #3 (Remarks to the Author):

The authors have thoroughly and meaningfully addressed my comments from the first submission. The current manuscript is much improved and compelling.

Author Rebuttals to First Revision:

Rebuttal of Nwosu, Ward, et al.

We are again truly grateful for the additional time and effort put forth by the Editors and Reviewers at *Nature*. We appreciate the supportive feedback and insightful comments. We address these additional comments in this point-by-point response; referee remarks are presented in plain text, our responses in bold.

REVIEWERS' COMMENTS

R1

The authors have addressed my concerns. I commend them for the great work they have done. The regulation of UPP1 signaling by KRAS adds an interesting layer of complexity that may be valuable in designing therapeutic strategies for targeting UPP1 and KRAS signaling in PDAC.

We sincerely appreciate the referee's thoughtful feedback during this review process. The compendium of excellent suggestions from this and all referees has helped to make for a much stronger manuscript.

The two studies (Nwosu, Ward et al. and Jourdain et al.,) complement one another and come to the same conclusions. However, although Nwosu, Ward et al., show that UPP1 is under the control of KRAS, Jourdain et al., state that UPP1 is constitutively expressed. The authors are expected to clarify this divergence.

In all the pancreatic cancer models examined (human and murine cell lines and tumors), UPP1 is expressed constitutively, albeit with marked variability across lines (Figure 1e,f, Figure 3g; Extended Data Figure 6a). We demonstrate that UPP1 expression is under the control of the KRAS-MAPK pathway, where pathway inhibition potently represses UPP1 expression. Of note, all the models utilized in this study are KRAS mutant (Figure 3l-o; Extended Data Figure 8b-i).

Furthermore, and in line with the work of Jourdain, et al., the baseline expression of UPP1 can be augmented considerably by external stimuli. We found that UPP1 expression is increased by glucose and/or uridine withdrawal (Figure 3n,o). These results are consistent with the findings from Jourdain, et al., in terms of both baseline and inducible regulation of UPP1. For example, Jourdain, et al. illustrate that UPP1 is baseline expressed in THP1, PBMCs, and BMDMs, and that this is markedly increased upon exposure to LPS or other stimuli.

Guided by the referee's astute observation, in our revised study, we discuss that UPP1 is expressed at baseline in the models examined, and that it is subject to layered regulation.

This article presents robust data and reveals an unappreciated role of circulating uridine in maintaining metabolism under glucose-limiting conditions in PDAC.

Minor comments:

Extended Data Fig.8i. Densitometric quantification of UPP1 protein signal from ASPC1 cells should accompany p-ERK signal.

We appreciate this suggestion and have now updated Extended Data Fig.8i to also include densitometry for the UPP1 blot in Figure 3o.

R2

This revision is improved from the original submission, with more detail and better validation of the importance of UPP1 in PDAC xenograft growth. The paper still does not make a convincing case that uridine is a major carbon source in vivo, however.

We thank the referee for their time, supportive feedback, and careful attention to experimental detail. Like the referee, we agree that a clear demonstration of the physiological relevance of uridine-derived ribose as a carbon source is paramount. In our response below, we discuss both the experimental rationale for the in vivo uridine tracing studies, from both the Lyssiotis and Mootha labs, and points for consideration when interpreting the data. We have included aspects of this discussion and new associated experimental data in our revised submission.

There are a few concerns along these lines:

1. The authors quantify uridine and glucose in TIF from their models, but they don't use these concentrations in the culture experiments. According to the rebuttal (Fig. 4), TIF contains about 25 micromolar uridine and 500 micromolar glucose. But the experiments designed to assess uridine metabolism do not use these conditions; the most relevant concentrations used in the paper are 100 micromolar each of glucose and uridine, but these are skewed to favor uridine metabolism. During culture, the concentration of glucose may decline rapidly, further favoring use of carbon from uridine.

We appreciate this comment from the referee and agree that quantifying metabolite concentrations in TIF and applying these in vitro is an appropriate approximation to study fuel utilization at physiological concentrations. In our previous experimental studies, we observed that glucose and uridine competitively enter central carbon metabolism, dependent on relative abundance (mole to mole), and we used TIF-related concentration estimates to model this competition in vitro. When glucose is higher, glucose carbon dominates metabolic incorporation, as read out by less uridine carbon enrichment in downstream metabolites; when uridine is higher, uridine carbon is preferentially utilized (Figure 2i; Extended Data Figure 3b). Further, as illustrated in Extended Data Figure 4a-c, and mirroring the isotope enrichment analysis, uridine significantly influences cellular metabolic activity when the uridine to glucose ratio is 1:10 but not 1:100.

These points notwithstanding, we appreciate the referee's opinion, and in the spirit of uniformity and transparency, we performed isotope tracing metabolomics with $^{13}\text{C}5$ -uridine using the formal concentrations abstracted from the TIF studies (i.e. 25 μM uridine, 650 μM glucose; Figure 2h), and we include these data in this revised study (Extended Data Figure 3c,d). Our new data are consistent with previous observations that ribose carbon is abstracted from uridine, in mouse and human PDAC cells, and that this enters into the PPP, glycolysis, glycolytic branch pathways, nucleotide metabolism, and the TCA cycle.

Finally, we would also like to briefly mention that TIF measurements, while a reasonable approximation, are representative of "bulk". There are undoubtedly areas of the tumor that see more/less glucose and uridine. We believe that it is in these areas/instances of fluctuation in glucose availability that uridine is a particularly important input into central carbon metabolism. We have updated the discussion to highlight the additional points motivated by this excellent referee suggestion.

2. With this in mind, the data from the in vivo tracing experiment give the impression that the pathway is not very active. The delivery method (ip injection or injection directly into the tumors) labels about 30% of the uridine pool, but most metabolites display much less labeling, more than 10-fold less in some cases. The authors ascribe this to the short labeling period (1 hour), but it's equally plausible that the pathway is just not very active.

We again appreciate the referee's careful attention to our experimental parameters, being particularly cognizant of the labeling duration. We argue that the labeling is actually substantial. After only 1 hour of uridine exposure, the tumor uridine pools were labeled by 30% (Figure 2g). Pentose phosphate pathway and upstream glycolytic intermediates (e.g. R5P, F6P, FBP) were 10% labeled. Normalizing this to the tumor uridine labeling, as is common for in vivo isotope tracing analyses, uridine-derived ribose carbon contributed to a third of these metabolite pools (i.e. 10% from 30%). Following the same logic downstream, uridine labeled 15% of the lactate and citrate pools (~5% from ~30%). Put another way, each labeled metabolite fraction is normalized relative to the input mixture of labeled/unlabeled uridine to account for flux of unlabeled, endogenous uridine into each metabolite pool (Figure 2g; Extended Data Figure 3a). We utilize this new presentation of the data to clarify the referee's discerning question. Substantiating our point, we have observed in vitro that the duration of label exposure is directly proportional to the degree of label incorporation, with upstream metabolites (uridine, pentose phosphate pathway) being labeled faster and more thoroughly than downstream metabolites (e.g. TCA cycle intermediates). In fact, it takes >2 hours of ¹³C5-ribose-labeled uridine exposure to saturate the labeling of TCA cycle intermediates.

In either case, it is also important to note that the objective of this experiment was not to label tumors to steady-state with uridine-derived ribose – a technique for comparing relative activity between two conditions – but rather to determine if uridine was used in vivo. Perhaps more relevantly, the labeling of central carbon metabolism with uridine-derived ribose is on par with that of a similar 1h injection/analysis paradigm from our lab using ¹³C-glucose (Yuan, et al. Nature Protocols 2019); glucose being unquestionably a critical fuel for pancreatic tumors. We believe that the considerable uptake/metabolism in this bolus injection at a pre-steady state labeling time lends to the interpretation that uridine-derived ribose can be readily used in pancreatic tumors in vivo.

Finally, in a related experiment, Jourdain, et al. demonstrated that fasted mice injected with a bolus of labeled uridine for 30min exhibited similar magnitudes and patterns of label enrichment in a number of central carbon metabolites (e.g. lactate, ribose and blood glucose) in the liver and serum. Collectively, we believe that these data provide support for a physiological role of uridine-derived ribose as a carbon source. Based on this important concern from the referee, we have updated our revised manuscript to highlight these experimental considerations and limitations.

3. The effect of UPP1 knockout on tumor growth is impressive and certainly helps the paper, but given the apparently low activity of uridine catabolism in vivo, how sure can the authors be that the mechanism by which UPP1 supports tumor growth is by providing carbon for central metabolism? UPP1 is a bidirectional enzyme so it seems likely that the mechanism could be different.

Based on the response above, we propose that our method to follow uridine catabolism in vivo is underestimating the actual activity of this pathway due to technical reasons; i.e. limited label exposure and incomplete saturation of uridine pools. Namely, we would also like to emphasize that the uridine we deliver is being competitively diluted by serum- and TME-derived uridine, which meaningfully decrease fractional enrichment. Furthermore, and perhaps more compelling, Jourdain, et al. provide data in their parallel study that intracellular RNA turnover is an additional source of uridine that can be utilized similarly, again diluting enrichment, where the recycling of ribosomes through ribophagy plays an important role in supporting viability during nutrient limitation (Kraft C, et al. Nature Cell Biology 2008). These caveats, together with our current data showing biologically significant labeling, suggest that the true in vivo catabolism of uridine is not low.

In either case, as the referee expertly recognizes, the tumor growth phenotype provides a more proximal readout of UPP1 activity that is less subject to these experimental shortcomings. We believe that the effect of this knockout in vivo is on-mechanism and not due to ablation of the reverse activity of UPP1 (i.e. blocking uracil + R1P -> uridine) because we demonstrate that in UPP1- functional tumors (like the experiment referenced above), there is considerable flux of uridine-derived carbon into central carbon metabolism. This result would be unlikely if UPP1 was primarily functioning in reverse in vivo.

An initially plausible counterargument to this conclusion could be that delivering a bolus of uridine to the tumor elevated concentrations enough to push UPP1 in the forward (i.e. ribose-producing) direction, in a manner not present in the tumor knockout studies where we do not inject uridine. However, knockout of UPP1 in tumors leads to an accumulation of uridine and a drop in not uracil, indicating that the predominant direction of flux of UPP1 is the breaking of uridine (Figure 4l; Extended Data Figure 10e). Furthermore, we recall to the referee our response to their first point and Extended Data Figure 3d in which, even at physiological glucose and uridine levels, UPP1 is functioning in the ribose-producing direction.

We heartily appreciate the referee's suggestions encouraging us to thoroughly check our assumptions regarding UPP1 activity as the resulting data has undoubtedly strengthened our manuscript.

R3

The authors have thoroughly and meaningfully addressed my comments from the first submission. The current manuscript is much improved and compelling.

We are grateful for the referee's thoughtful feedback and support during this review process. The collective feedback resulted in meaningful improvements in depth and presentation of our manuscript.

Reviewer Reports on the Second Revision:

Referees' comments:

Referee #2 (Remarks to the Author):

The authors addressed my critiques and I appreciate the response.